# ZFP148 is a transcriptional repressor of cytolytic effector CD8+ T cell differentiation

Tong Xiao[1], Xingyu Chen [2,3], No-Joon Song[1], Ryan J. Brown[4,5,6], Anjun Ma [1,7], Jay K. Mandula[1], Amir Yousif[1,8], Yi Wang[1], Minh Quynh May Le[1], Jianying Li[1,8], Fengxia Gao[1,8], Bella Weaver[1,8], Heng-Yi Chen [1,8], Fang-Yun Lay[1,8], Debasish Sundi[1,9], Maria Velegraki [1], Payton Weltge[1], Juanita L. Merchant[10], Mark P. Rubinstein[1], Kenneth J. Oestreich [1,8], Chan-Wang Jerry Lio[1,8], Hazem E. Ghoneim [1,8], Xue Li[11], Dan Theodorescu [10], Gang Xin[1,8], Qin Ma [1,7], Weiguo Cui [4,5] & Zihai Li [1,12] ✉

Progenitor CD8+ T cells differentiate into effector and exhausted progenies during chronic antigen stimulation; however, mechanisms that restrain exhaustion and sustain effector differentiation remain incompletely defined. Here we identified the transcription factor ZFP148 as a repressor of CD8+ T cell effector differentiation. ZFP148-deficient CD8+ T cells displayed increased frequency of cytolytic effector cells and reduced frequency of exhausted cells compared with *Zfp148*fl/fl controls during chronic viral infection. Mechanistically, ZFP148 limited the chromatin accessibility of effector-driving transcription factor motifs and directly repressed expression of the transcription factor KLF2. Furthermore, conditional ZFP148 ablation in CD8+ T cells synergized with programmed cell death-1 blockade to improve tumor control in syngeneic mouse models. Consistently, cancer patients with lower *ZNF148* expression in tumor-infiltrating CD8+ T cells showed improved responsiveness to immunotherapies. Collectively, our study identifies ZFP148 as a transcriptional repressor of CD8+ T cell effector differentiation and highlights its therapeutic potential for enhancing antitumor immunity.

During persistent viral infection and cancer, antigen-specific CD8+ T cells undergo T cell exhaustion, marked by progressive loss of effector function and proliferative capacity[1,2]. Within this heterogeneous compartment, a small subset of self-renewing progenitor CD8+ T cells ($T_{PRO}$ cells), defined by high expression of stemness-associated transcription factor TCF1 and chemokine receptor CXCR5, sustains antigen-driven responses by generating more differentiated progenies, including terminally exhausted ($T_{EX}$) cells[3,4]. Although the exhaustion program limits immunopathology, it also compromises viral clearance and tumor control[5,6]. Notably, CD8+ $T_{PRO}$ cells can alternatively give rise to cytolytic effectors that highly express the effector marker CX3CR1 and cytolytic molecule granzyme B (GZMB) (hereafter $T_{EFF}$ cells), which are critical for viral and tumor control and response to programmed cell death 1

(PD-1) blockade[7–9]. Several transcriptional and epigenetic programs regulating these fate decisions have been described. Sustained TCF1 expression maintains the CD8+ $T_{PRO}$ cell pool, whereas TCF1 downregulation is associated with terminal exhaustion[10,11]. Transcription factors reinforcing exhaustion include TOX[12–14], IRF2[15], IRF4[16], the NR4A family[17] and NFAT proteins[18], whereas differentiation toward CX3CR1+CD8+ $T_{EFF}$ cells is driven by transcription factors BATF[19], KLF2[20] and ZEB2[21]. Thus, defining mechanisms that enhance cytolytic CD8+ $T_{EFF}$ cell differentiation while limiting exhaustion is key to improving immune control of chronic infections and cancer.

ZFP148 (encoded by *Zfp148* in mouse and *ZNF148* in human) is a Krüppel family transcription factor that regulates proliferation and differentiation in nonimmune cells[22] and promotes T helper 2 function

in CD4[+] T cells[23]. However, the role of ZFP148 in CD8[+] T cells remains unknown. Here we identify ZFP148 as a negative regulator that restrains CD8[+] $T_{EFF}$ cell differentiation and promotes exhaustion through repressing KLF2. Conditional deletion of ZFP148 in CD8[+] T cells enhanced their cytolytic capacity, costimulatory signaling and proliferative potential, synergizing with PD-1 blockade to control syngeneic tumor growth; concordantly, lower *ZNF148* expression was associated with improved immunotherapy responses in cancer patients, implicating a conserved ZFP148–KLF2 axis. Together, these findings establish ZFP148 as a transcriptional repressor of cytolytic CD8[+] $T_{EFF}$ cell differentiation and a potential target to enhance responses to immunotherapies.

## Results

### ZFP148 is enriched in CD8[+] $T_{PRO}$ cells

Published computational inferences identified ZFP148 as a potential transcriptional regulator in CD8[+] T cells differentiation during chronic viral infection and cancer[10,24]. We analyzed published single-cell (sc) RNA sequencing (RNA-seq) datasets of adoptively transferred P14 CD8[+] T cells from spleens of C57BL/6 wild-type (WT) recipient mice infected with acute (Armstrong) or chronic (Cl13) lymphocytic choriomeningitis virus (LCMV) at days 8, 15 and 30 postinfection (p.i.)[21]. *Zfp148* expression was reduced in P14 CD8[+] T cells during chronic compared to acute infection (Fig. 1a), suggesting potential antigen load-dependent regulation. Flow cytometry showed that ZFP148 mean fluorescence intensity (MFI) was upregulated rapidly in splenic CD44[hi]GP$_{33-41}$ tetramer (Tet)[+]CD8[+] T cells of C57BL/6 WT mice at day 8 post-LCMV Cl13 infection versus naive CD44[−]CD62L[+]CD8[+] T cells (hereafter CD8[+] $T_N$ cells; Fig. 1b and Extended Data Fig. 1a). In vitro TCR engagement through anti-CD3 (aCD3) antibody (Ab) stimulation also induced ZFP148 protein expression in splenic CD8[+] $T_N$ cells with minimal dependence on CD28 costimulatory signaling (Fig. 1c,d). Furthermore, cyclosporin A (CsA) abrogated this induction (Extended Data Fig. 1b), indicating calcineurin–NFAT-dependent regulation.

ZFP148 protein expression declined by ~50% between days 8 and 16 and stabilized through day 30 in splenic CD44[hi]GP$_{33-41}$ Tet[+]CD8[+] T cells p.i. (Fig. 1b). Using established effector markers (CX3CR1, GZMB) and progenitor markers (Ly108, TCF1)[19], we detected higher expression of ZFP148 protein in CX3CR1[−]Ly108[+] or GZMB[−]TCF1[+]CD8[+] $T_{PRO}$ cells versus their CD8[+] $T_{EFF}$ or CD8[+] $T_{EX}$ counterparts in both splenic CD44[hi]GP$_{33-41}$ Tet[+]CD8[+] T cells and CD44[hi]GP$_{276-286}$ Tet[+]CD8[+] T cells at day 22 p.i. (Fig. 1e and Extended Data Fig. 1c,d). Similar enrichment of ZFP148 in CD8[+] $T_{PRO}$ cells was also observed in CD8[+] tumor-infiltrating lymphocytes (CD8[+] TILs) in C57BL/6 WT mice at day 14 postsubcutaneous injection with the colon adenocarcinoma MC38 and human muscle-invasive bladder tumors (Extended Data Fig. 1e,f). Collectively, these data indicated that ZFP148 was enriched in CD8[+] $T_{PRO}$ cells and declined during differentiation into CD8[+] $T_{EFF}$ and CD8[+] $T_{EX}$ cells.

### ZFP148 inhibits CD8[+] $T_{EFF}$ cell differentiation

To assess the role of ZFP148 in antigen-specific CD8[+] T cell differentiation during chronic viral infection, we crossed E8i[Cre] mice with *Zfp148*[fl/fl] mice to generate CD8[+] T cell-specific ZFP148-knockout (KO) mice (hereafter ZFP148 cKO mice). Flow cytometry confirmed efficient deletion of ZFP148 in splenic CD8[+] T cells and comparable frequencies of principal immune cell lineages in spleens of ZFP148 cKO mice compared to *Zfp148*[fl/fl] mice (Extended Data Fig. 2a–c).

Longitudinal flow cytometric analysis showed similar numbers and frequencies of CD44[hi]GP$_{33-41}$ Tet[+]CD8[+] T cells in spleens of *Zfp148*[fl/fl] and ZFP148 cKO mice at days 16, 22 and 30 post-LCMV Cl13 infection, albeit a reduction in number at day 8 p.i. in ZFP148 cKO mice compared to *Zfp148*[fl/fl] mice (Extended Data Fig. 3a,b). ZFP148 cKO mice exhibited increased frequencies of CX3CR1[+]Ly108[−]CD8[+] $T_{EFF}$ cells compared to *Zfp148*[fl/fl] mice in spleens at days 16, 22 and 30 p.i. (Fig. 2a and Extended Data Fig. 3c), accompanied by reduced frequency and number of CX3CR1[−]Ly108[+]CD8[+] $T_{PRO}$ cells at days 8 and

16 p.i. (Extended Data Fig. 3d) and CX3CR1[−]Ly108[−]CD8[+] $T_{EX}$ cells and Ly108[−]CD69[+]CD8[+] $T_{EX}$ cells at day 22 p.i. (Extended Data Fig. 3e,f). Expression of inhibitory receptors (PD-1, TIM-3, LAG-3, CTLA-4, CD39 and CD101) in total CD44[hi]GP$_{33-41}$ Tet[+]CD8[+] T cells or CD8[+] $T_{EX}$ cells in spleens at days 22 and 30 p.i. was largely comparable between genotypes (Fig. 2b and Extended Data Fig. 3g). We observed increased production of GZMB but not pro-inflammatory cytokines (interferon gamma (IFNγ), tumor necrosis factor (TNF) and interleukin 2 (IL-2)) in ZFP148 cKO versus *Zfp148*[fl/fl] CD44[hi]GP$_{33-41}$ Tet[+]CD8[+] T cells in spleens at days 8, 16 and 22 p.i. (Fig. 2c and Extended Data Fig. 3h). Increased frequencies of CX3CR1[+]Ly108[−]CD8[+] $T_{EFF}$ cells were also detected in livers and lungs of ZFP148 cKO versus *Zfp148*[fl/fl] mice at day 22 p.i. (Extended Data Fig. 3i).

ZFP148 cKO mice displayed increased frequencies of subset expressing both CX3CR1 and the proliferation marker Ki-67 in splenic CD44[hi]GP$_{33-41}$ Tet[+]CD8[+] T cells compared to *Zfp148*[fl/fl] mice (Fig. 2d), together with elevated Ki-67 MFI in splenic CD8[+] $T_{PRO}$ cells, but not CD8[+] $T_{EX}$ cells (Extended Data Fig. 3j). Despite enhanced proliferation, neither the frequency nor number of splenic CD44[hi]GP$_{33-41}$ Tet[+]CD8[+] T cells (Extended Data Fig. 3b), nor their apoptosis, as assessed by Annexin V and propidium iodide (PI) staining, differed in ZFP148 cKO versus *Zfp148*[fl/fl] mice (Extended Data Fig. 4a,b). Instead, increased frequency and number of CD44[hi]GP$_{33-41}$ Tet[+]CD8[+] T cells were detected in the blood of ZFP148 cKO mice (Extended Data Fig. 4c), accompanied by increased lymphoid-homing marker CD62L and reduced tissue-resident marker CD69 expression in splenic CD44[hi]GP$_{33-41}$ Tet[+]CD8[+] T cells, compared to *Zfp148*[fl/fl] mice (Extended Data Fig. 4d). These features were consistent with increased circulation and reduced tissue retention of CX3CR1[+]Ly108[−]CD8[+] $T_{EFF}$ cells[25–28].

To assess cytolytic function, mixed splenic CD44[hi]GP$_{33-41}$ Tet[+] and CD44[hi]GP$_{276-286}$ Tet[+]CD8[+] T cells sorted from ZFP148 cKO or *Zfp148*[fl/fl] mice at day 22 post-LCMV Cl13 infection were cocultured with B16F10 cells expressing LCMV glycoprotein (hereafter B16-GP cells). Sorted CD8[+] T cells from ZFP148 cKO mice exhibited enhanced killing of B16-GP cells compared to those from *Zfp148*[fl/fl] mice, as indicated by increased Annexin V staining (Fig. 2e). Similarly, increased cytotoxicity against B16-GP cells was observed in activated P14 CD8[+] T cells CRISPR-edited with *Zfp148*-targeting sgRNAs (hereafter P14 CD8[+] $T_{ZFP148-null}$ cells) versus a nontargeting (NT) control sgRNA (hereafter P14 CD8[+] $T_{NT}$ cells) (Fig. 2f). Serum LCMV Cl13 titers were comparable between ZFP148 cKO and *Zfp148*[fl/fl] mice at days 8, 21 and 28 p.i. (Extended Data Fig. 4e).

During acute LCMV Armstrong infection, both frequencies and numbers of CD44[hi]GP$_{33-41}$ Tet[+]CD8[+] T cells in inguinal lymph nodes (iLNs) and spleens were comparable between ZFP148 cKO and *Zfp148*[fl/fl] mice at day 9 p.i. (Extended Data Fig. 5a). We observed increased frequency of short-lived effector CD8[+] T cells ($T_{SLEC}$) expressing the effector marker KLRG1 but not the memory marker CD127 and reduced frequency and number of memory precursor KLRG1[−]CD127[+]CD8[+] T cells ($T_{MPEC}$) (Extended Data Fig. 5b–d), accompanied by increased expression of cytolytic molecule GZMA and GZMB and unchanged IFNγ and TNF production, in CD44[hi]GP$_{33-41}$ Tet[+]CD8[+] T cells in iLNs and spleens of ZFP148 cKO versus *Zfp148*[fl/fl] mice at day 9 p.i. (Extended Data Fig. 5e). Together, these results indicated that CD8[+] T cell-intrinsic loss of ZFP148 promoted cytolytic CD8[+] $T_{EFF}$ differentiation during both chronic and acute viral infection.

### ZFP148 loss promotes CD8[+] $T_{EFF}$ cell programming

To define transcriptional and epigenetic reprogramming induced by ZFP148 deficiency, we performed paired scRNA-seq and scATAC-seq (assay for transposase-accessible chromatin using sequencing) on CD44[hi]GP$_{33-41}$ Tet[+]CD8[+] T cells sorted from spleens of ZFP148 cKO or *Zfp148*[fl/fl] mice at day 21 post-LCMV Cl13 infection (Extended Data Fig. 6a). Joint analysis identified six clusters (Fig. 3a). After examining expression of key CD8[+] T cell marker genes (Fig. 3b)

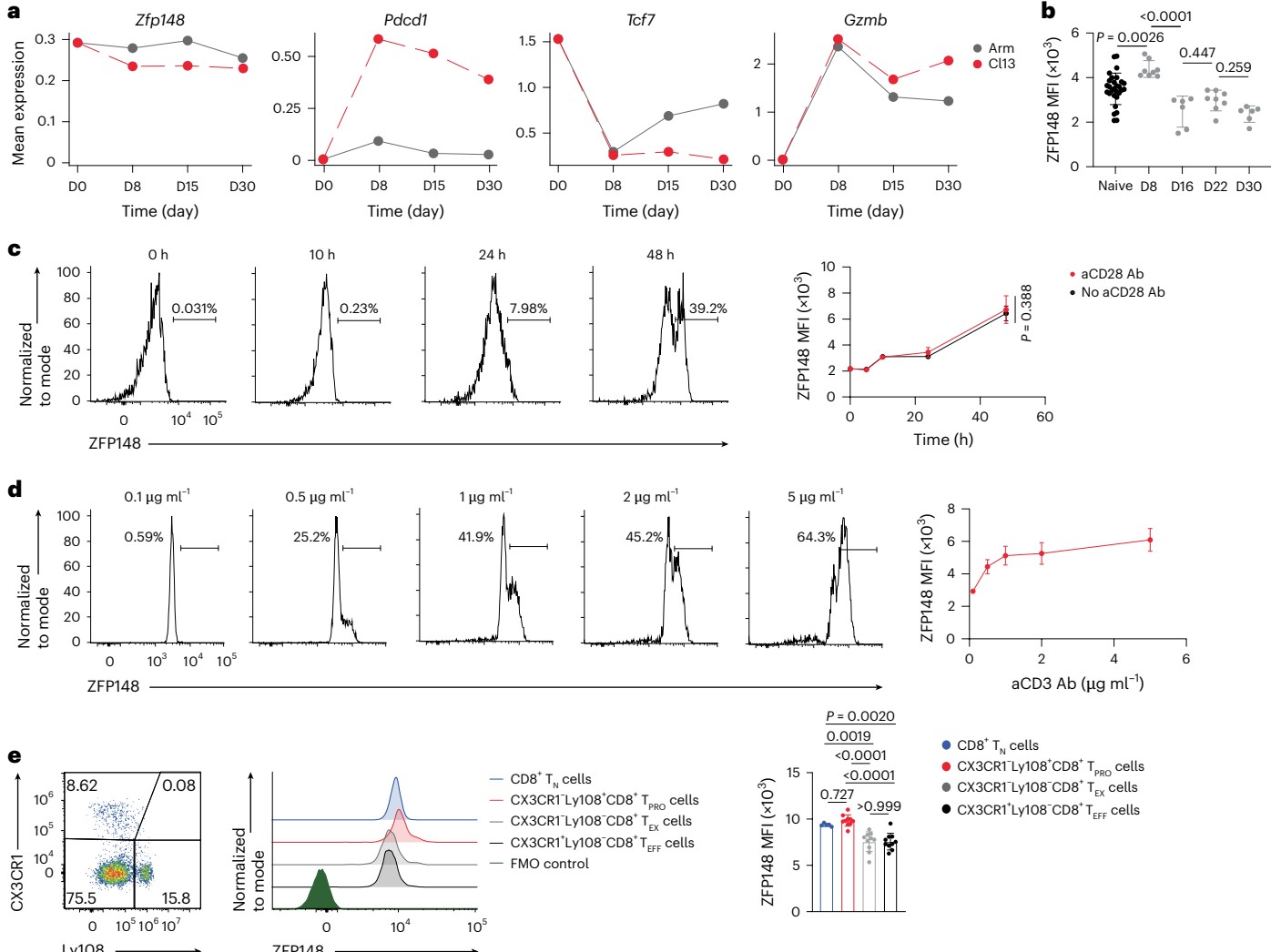

**Fig. 1 | ZFP148 is enriched in CD8⁺ T_PRO cells and induced by TCR signaling.**
**a**, Expression of *Zfp148*, *Pdcd1*, *Tcf7* and *Gzmb* mRNA in splenic P14 CD8⁺ T cells transferred to C57BL/6 WT mice infected with either LCMV Armstrong (Arm) or Cl13 at day 1 post-T cell transfer and analyzed at day (D) 0, 8, 15 and 30 p.i., generated by reanalysis of published scRNA-seq datasets[21]. **b**, MFI of ZFP148 in splenic CD44^hi GP_{33–41} Tet⁺CD8⁺ T cells in C57BL/6 WT mice at day 8 (*n* = 8), 16 (*n* = 6), 22 (*n* = 8) and 30 (*n* = 6) post-LCMV Cl13 infection and splenic CD8⁺ T_N cells in noninfected C57BL/6 WT mice as a control (*n* = 29). **c**,**d**, Density plots showing ZFP148 protein expression (left) and MFI of ZFP148 (right) in splenic CD8⁺ T_N cells of C57BL/6 WT mice stimulated with 5 μg ml⁻¹ aCD3 Ab with or without 2 μg ml⁻¹ aCD28 Ab for 0, 10, 24 and 48 h (**c**) or with 0.1, 0.5, 1, 2 or 5 μg ml⁻¹ of aCD3 Ab and

2 μg ml⁻¹ aCD28 Ab for 48 h (**d**) (*n* = 4). **e**, Scatter plots showing CX3CR1 versus Ly108 protein expression in splenic CD44^hi GP_{33–41} Tet⁺CD8⁺ T cells (left) and density plots showing expression of ZFP148 protein (middle) and MFI of ZFP148 (right) in splenic CX3CR1⁻Ly108⁺CD8⁺ T_PRO cells, CX3CR1⁻Ly108⁻CD8⁺ T_EX cells and CX3CR1⁺Ly108⁻CD8⁺ T_EFF cells in C57BL/6 WT mice at day 22 post-LCMV Cl13 infection (*n* = 10) and splenic CD8⁺ T_N cells in noninfected C57BL/6 WT mice as a control (*n* = 5). FMO, fluorescence minus one. Data are representative of two to three independent experiments (**b**–**e**). Data are presented as mean ± s.d. Statistical analysis was performed using one-way ANOVA followed by Šídák's multiple comparisons test (**b**); two-way ANOVA (**c**); one-way ANOVA followed by Tukey's multiple comparisons test (**e**).

and gene signatures from a published LCMV Cl13 scRNA-seq dataset[29] (Fig. 3c), we annotated five distinct CD8⁺ T cell subsets, including T_P1 (progenitor CD8⁺ T cells group 1, expressing *Sell*, *Tcf7*, *Slamf6* and *Pdcd1*), T_P2 (progenitor CD8⁺ T cells group 2; *Tcf7*, *Slamf6* and *Pdcd1*), T_EFF (effector CD8⁺ T cells, *Cx3cr1*, *Gzmb*), prolif T (proliferative CD8⁺ T cells; *mKi67*) and T_EX (exhausted CD8⁺ T cells; *Pdcd1*, *Havcr2*, *Cd244a*) (Fig. 3d and Extended Data Fig. 6b). Compared to splenic CD44^hi GP_{33–41} Tet⁺CD8⁺ T cells in *Zfp148*^fl/fl mice, the same cells in ZFP148 cKO mice contained increased frequency of CD8⁺ T_EFF and CD8⁺ prolif T cells (35.2% and 13.5% versus 13.8% and 6.5%, respectively), accompanied by reduced frequency of CD8⁺ T_P2 and CD8⁺ T_EX cells and minimal changes in CD8⁺ T_P1 cells (Fig. 3d). Slingshot pseudotime analysis identified two trajectories: an 'effector lineage' progressing from CD8⁺ T_P1 through CD8⁺ T_P2 to CD8⁺ T_EFF and CD8⁺ prolif T cells, and an 'exhausted lineage' progressing from CD8⁺ T_P1

through CD8⁺ T_P2 and CD8⁺ T_EFF to CD8⁺ T_EX cells (Fig. 3e). Consistently, CD44^hi GP_{33–41} Tet⁺CD8⁺ T cells in ZFP148 cKO mice were enriched in the 'effector lineage' compared to same cells in *Zfp148*^fl/fl mice (38.2% versus 23.8%) (Fig. 3e).

Differential gene expression analysis revealed reduced expression of stemness genes (*Il7r*, *Tcf7* and *Slamf6*) in ZFP148 cKO versus *Zfp148*^fl/fl CD8⁺ T_P2 cells, increased expression of effector genes (*Klrk1*, *Klrd1*, *Cx3cr1*, *Zeb2* and *Klf2*) in ZFP148 cKO versus *Zfp148*^fl/fl CD8⁺ T_EFF cells, increased expression of proliferation-associated genes (*Cdca8*, *Cdc20* and *E2f7*) in ZFP148 cKO versus *Zfp148*^fl/fl CD8⁺ prolif T cells and reduced expression of inhibitory receptor genes (*Cd160* and *Cd244a*) in ZFP148 cKO versus *Zfp148*^fl/fl CD8⁺ T_EX cells (Fig. 3f). Subset-specific transcriptional differences were also pronounced in CD8⁺ T_EFF cells, with ZFP148 cKO CD8⁺ T_EFF cells showing increased expression of effector genes (*Zeb2*, *Klrc2*, *Klf2* and *S1pr5*) compared to *Zfp148*^fl/fl CD8⁺ T_EFF cells

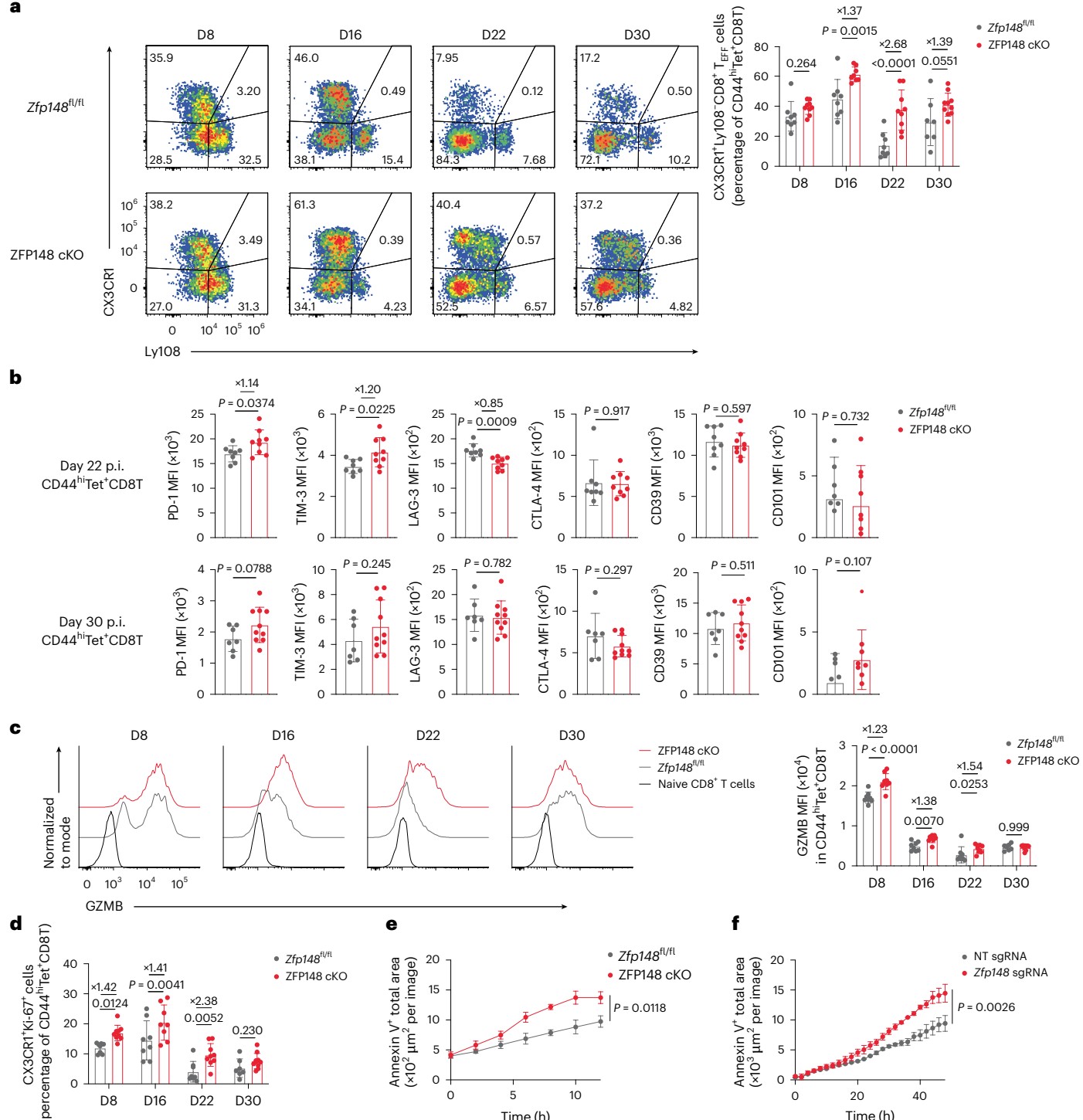

**Fig. 2 | ZFP148 restrains cytolytic effector differentiation of antigen-specific CD8⁺ T cells during chronic LCMV infection. a**, Scatter plots showing CX3CR1 versus Ly108 protein expression (left) and frequency of CX3CR1⁺Ly108⁻CD8⁺ $T_{EFF}$ cells (right) in splenic CD44^hi GP$_{33-41}$ Tet⁺CD8⁺ T cells in *Zfp148*^fl/fl or ZFP148 cKO mice at day 8 (*Zfp148*^fl/fl, $n = 8$; ZFP148 cKO, $n = 9$), 16 (*Zfp148*^fl/fl, $n = 8$; ZFP148 cKO, $n = 8$), 22 (*Zfp148*^fl/fl, $n = 8$; ZFP148 cKO, $n = 9$) and 30 (*Zfp148*^fl/fl, $n = 7$; ZFP148 cKO, $n = 10$) post-LCMV Cl13 infection. **b**, MFI of PD-1, TIM-3, LAG-3, CTLA-4, CD39 and CD101 in splenic CD44^hi GP$_{33-41}$ Tet⁺CD8⁺ T cells in *Zfp148*^fl/fl or ZFP148 cKO mice at day 22 (*Zfp148*^fl/fl, $n = 8$; ZFP148 cKO, $n = 9$) and 30 (*Zfp148*^fl/fl, $n = 7$; ZFP148 cKO, $n = 10$) post-LCMV Cl13 infection. **c**, Density plots showing expression of GZMB protein (left) and MFI of GZMB (right) in splenic CD44^hi GP$_{33-41}$ Tet⁺CD8⁺ T cells

in *Zfp148*^fl/fl and ZFP148 cKO mice as in **a**. **d**, Frequency of Ki-67⁺CX3CR1⁺CD8⁺ T cells in splenic CD44^hi GP$_{33-41}$ Tet⁺CD8⁺ T cells in *Zfp148*^fl/fl or ZFP148 cKO mice as in **a**. **e**,**f**, Kinetics of Annexin V⁺ total area in B16-GP cells cocultured with mixed splenic CD44^hi GP$_{33-41}$ Tet⁺ and CD44^hi GP$_{276-286}$ Tet⁺CD8⁺ T cells sorted from ZFP148 cKO or *Zfp148*^fl/fl mice at day 22 post-LCMV Cl13 infection ($n = 4$) (**e**) or P14 CD8⁺ T cells activated and CRISPR-edited with either an NT control or *Zfp148*-targeting sgRNAs ($n = 3$) (**f**). Fold changes in **a**–**d** were calculated as mean (ZFP148 cKO) divided by mean (*Zfp148*^fl/fl). Data are representative of two to three independent experiments (**a**–**f**). Data are presented as mean ± s.d. (**a**–**d**) or ± s.e.m. (**e**,**f**). Statistical analysis was performed using multiple unpaired two-sided *t*-tests (**a**, **c** and **d**); an unpaired two-sided *t*-test (**b**) and two-way ANOVA (**e** and **f**).

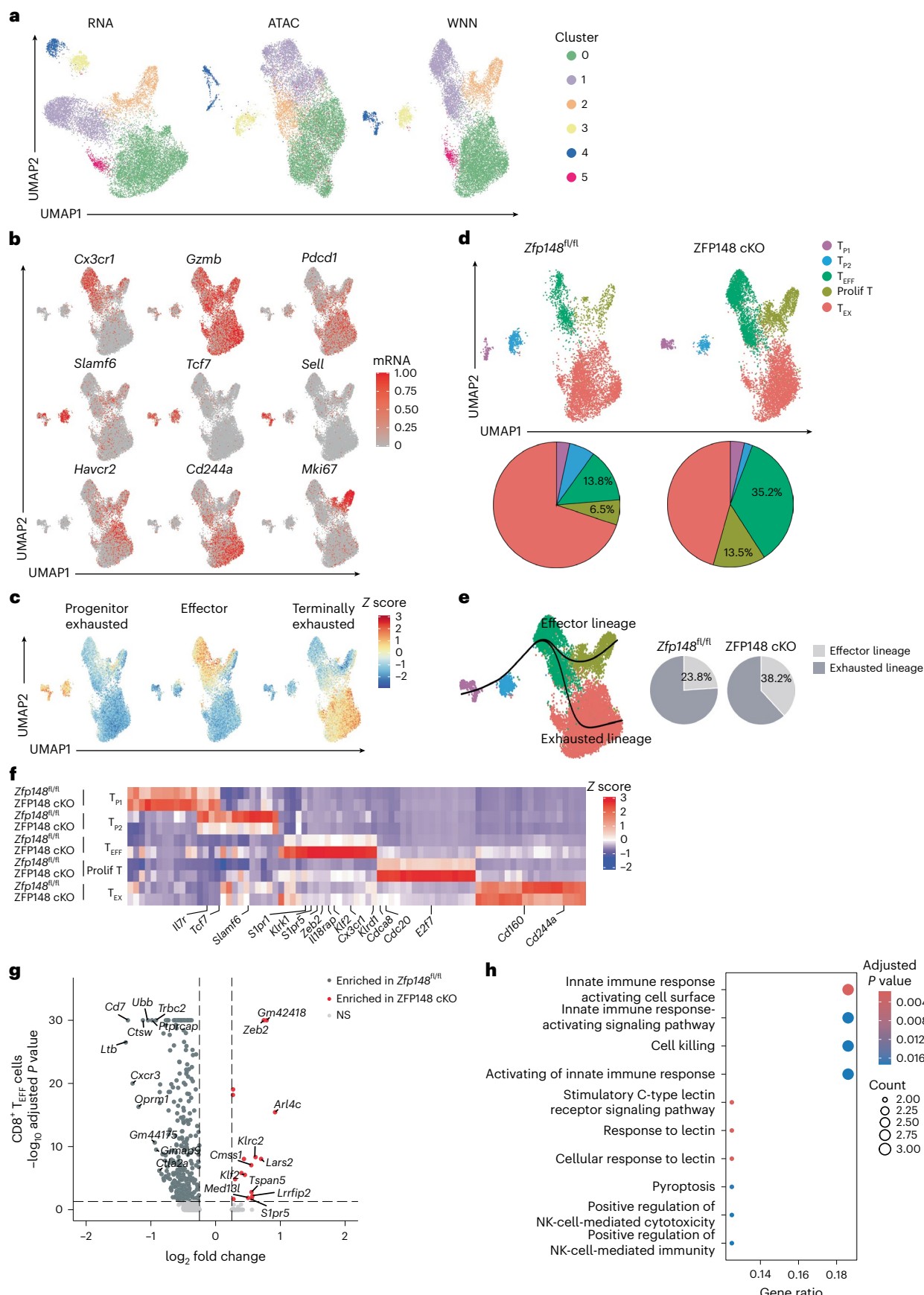

**Fig. 3 | ZFP148 deficiency transcriptionally reprograms antigen-specific CD8⁺ T cells toward cytolytic effectors. a**, RNA, ATAC and weighted-nearest neighbor (WNN) Uniform Manifold Approximation and Projection (UMAP) visualization of splenic CD44$^{hi}$GP$_{33-41}$ Tet⁺CD8⁺ T cells in *Zfp148*$^{fl/fl}$ (pooled from 10 mice; 5,669 cells) or ZFP148 cKO mice (pooled from 12 mice; 8,567 cells) at day 21 post-LCMV Cl13 infection, annotated by clusters identified by the Seurat R package. **b,c**, UMAP visualization of *Cx3cr1*, *Gzmb*, *Pdcd1*, *Slamf6*, *Sell*, *Havcr2*, *Cd244a* and *Mki67* mRNA expression (**b**) and published 'Progenitor exhausted,' 'Effector,' and 'Terminally exhausted' gene signatures[29] (**c**) in splenic CD44$^{hi}$GP$_{33-41}$ Tet⁺CD8⁺ T cells in *Zfp148*$^{fl/fl}$ or ZFP148 cKO mice as in **a**. **d**, UMAP visualization of splenic CD44$^{hi}$GP$_{33-41}$ Tet⁺CD8⁺ T cells annotated by subsets (CD8⁺ T$_{P1}$, CD8⁺ T$_{P2}$, CD8⁺ T$_{EFF}$, CD8⁺ prolif T and CD8⁺ T$_{EX}$) (top) and pie charts showing frequency of subsets

in splenic CD44$^{hi}$GP$_{33-41}$ Tet⁺CD8⁺ T cells in *Zfp148*$^{fl/fl}$ or ZFP148 cKO mice as in **a**. **e**, 'Effector' and 'exhausted' cell-state trajectories inferred using the Slingshot R package in splenic CD44$^{hi}$GP$_{33-41}$ Tet⁺CD8⁺ T cells in *Zfp148*$^{fl/fl}$ or ZFP148 cKO mice as in **a** (top) and pie charts showing proportion of *Zfp148*$^{fl/fl}$ or ZFP148 cKO CD44$^{hi}$GP$_{33-41}$ Tet⁺CD8⁺ T cells assigned to each lineage (bottom). **f**, Heatmap showing subset-specific DEGs in *Zfp148*$^{fl/fl}$ versus ZFP148 cKO splenic CD44$^{hi}$GP$_{33-41}$ Tet⁺CD8⁺ T cells as in **a**. **g,h**, Volcano plot showing DEGs (**g**) and GO enrichment analysis of DEGs (**h**) in splenic CD8⁺ T$_{EFF}$ cells of *Zfp148*$^{fl/fl}$ versus ZFP148 cKO mice as in **a**. NS, nonsignificant. Statistical analysis was performed using a two-sided Wilcoxon rank-sum test with Benjamini–Hochberg correction for multiple comparisons (**g**) or a one-sided hypergeometric test with Benjamini–Hochberg correction for multiple comparisons (**h**).

(Fig. 3g and Extended Data Fig. 6c). Gene ontology (GO) enrichment analysis revealed enrichment of 'Cell killing' and 'Stimulatory C-type lectin receptor signaling' pathways and natural killer (NK) cell-related cell identities in ZFP148 cKO versus *Zfp148*$^{fl/fl}$ CD8⁺ T$_{EFF}$ cells (Fig. 3h and Extended Data Fig. 6d). Collectively, these data indicated that ZFP148 deficiency redirected CD44$^{hi}$GP$_{33-41}$ Tet⁺CD8⁺ T cells differentiation toward CD8⁺ T$_{EFF}$ cells with NK cell-like transcriptional features over CD8⁺ T$_{EX}$ cells.

## ZFP148 loss remodels transcriptional regulatory programs

We analyzed differentially accessible chromatin regions (DACRs) and observed widespread changes in chromatin accessibility in ZFP148 cKO versus *Zfp148*$^{fl/fl}$ splenic CD44$^{hi}$GP$_{33-41}$ Tet⁺CD8⁺ T cells at day 21 post-LCMV Cl13 infection in scATAC-seq data described above (Fig. 4a). The highest proportion of gene-linked DACRs was observed in the CD8⁺ T$_{EFF}$ cells (Extended Data Fig. 7a) and expression of DACR-linked effector genes (*Klf2*, *Rap1gap2*, *S1pr5*, *Zeb2* and *Klrg1*) was increased in ZFP148 cKO versus *Zfp148*$^{fl/fl}$ CD8⁺ T$_{EFF}$ cells (Extended Data Fig. 7b). Consistently, *Zeb2* and *S1pr5* loci displayed increased promoter accessibility in ZFP148 cKO versus *Zfp148*$^{fl/fl}$ CD8⁺ T$_{P2}$ and CD8⁺ T$_{EFF}$ cells (Fig. 4b), indicating enhanced accessibility at key CD8⁺ T$_{EFF}$-associated gene loci.

ZFP148 deficiency altered accessibility of several lineage-determining transcription factor motifs across subsets of CD44$^{hi}$GP$_{33-41}$ Tet⁺CD8⁺ T cells (Fig. 4c). We observed reduced accessibility of progenitor-associated TCF1 and LEF1 motifs in ZFP148 cKO CD8⁺ T$_{P2}$ cells, increased accessibility of effector-associated KLF2, T-bet, STAT5A and RUNX3 motifs in ZFP148 cKO CD8⁺ T$_{EFF}$ cells and reduced accessibility of exhaustion-associated IRF2, IRF4, IKZF1, FLI1 and NFAT family motifs in ZFP148 cKO CD8⁺ T$_{EX}$ cells, compared to their *Zfp148*$^{fl/fl}$ counterparts (Fig. 4c).

Through subset-specific differential transcription factor motif accessibility analysis, we observed increased accessibility of effector-driving T-bet motif and reduced accessibility of LEF1 and TCF1 motifs in ZFP148 cKO versus *Zfp148*$^{fl/fl}$ CD8⁺ T$_{P2}$ cells (Fig. 4d). Concordantly, we also detected increased accessibility of effector-driving AP-1 family (FOS, JUND) motifs and reduced accessibility of exhaustion-associated NFATc2, NFATc4, IKZF1 and NR4A1 motifs in ZFP148 cKO versus *Zfp148*$^{fl/fl}$ CD8⁺ T$_{EFF}$ cells (Fig. 4d). These changes were initiated early in ZFP148 cKO CD8⁺ T$_{P2}$ cells and were sustained in CD8⁺ T$_{EFF}$, CD8⁺ prolif T and CD8⁺ T$_{EX}$ progenies compared to *Zfp148*$^{fl/fl}$ counterparts (Fig. 4e and Extended Data Fig. 7c). Collectively, ZFP148

deficiency reduced accessibility of CD8⁺ T$_{PRO}$ or CD8⁺ T$_{EX}$ cell-associated transcription factor motifs, while enhancing accessibility of CD8⁺ T$_{EFF}$ cell-associated transcription factor motifs.

## ZFP148 restrains KLF2-driving effector differentiation

To identify ZFP148 target genes, we focused on genes exhibiting both differential expression and altered promoter chromatin accessibility in ZFP148 cKO versus *Zfp148*$^{fl/fl}$ splenic CD44$^{hi}$GP$_{33-41}$ Tet⁺CD8⁺ T cells in scRNA-seq data described above and identified *Klf2* (encoding KLF2) among the top three upregulated genes in the absence of ZFP148 (Fig. 5a). Increased *Klf2* mRNA expression, promoter chromatin accessibility and KLF2 motif accessibility were detected across subsets of ZFP148 cKO versus *Zfp148*$^{fl/fl}$ splenic CD44$^{hi}$GP$_{33-41}$ Tet⁺CD8⁺ T cells (Fig. 5b,c and Extended Data Fig. 8a). To assess KLF2 protein expression, we used P14 mice expressing a KLF2–EGFP fusion protein (hereafter P14 KLF2–EGFP mice). Increased *Klf2* mRNA and KLF2–EGFP protein were detected in P14 KLF2–EGFP CD8⁺ T cells activated in vitro and CRISPR-edited with *Zfp148*-targeting sgRNAs versus an NT sgRNA (Fig. 5d,e). Concordantly, increased expression of the KLF2 gene signatures were observed in ZFP148 cKO versus *Zfp148*$^{fl/fl}$ splenic CD44$^{hi}$GP$_{33-41}$ Tet⁺CD8⁺ T cell (Fig. 5f and Extended Data Fig. 8b), indicating enhanced KLF2 downstream effector-driving transcriptional programs[20,30].

To test whether KLF2 mediated ZFP148-dependent CD8⁺ T$_{EFF}$ differentiation, we CRISPR-edited splenic P14 CD8⁺ T cells activated in vitro with an NT sgRNA (P14 CD8⁺ T$_{NT}$ cells), *Zfp148*-targeting sgRNAs (P14 CD8⁺ T$_{ZFP148-null}$ cells), *Klf2*-targeting sgRNAs (hereafter P14 CD8⁺ T$_{KLF2-null}$ cells) and mixed *Zfp148*-targeting and *Klf2*-targeting sgRNAs (hereafter P14 CD8⁺ T$_{ZFP148+KLF2-null}$ cells) and transferred equal numbers of CRISPR-edited cells into C57BL/6 WT recipient mice followed by LCMV Cl13 infection on the same day. Splenic CX3CR1⁺Ly108⁻CD8⁺ T$_{EFF}$ cells were enriched in transferred P14 CD8⁺ T$_{ZFP148-null}$ cells but nearly absent in P14 CD8⁺ T$_{KLF2-null}$ cells and P14 CD8⁺ T$_{ZFP148+KLF2-null}$ cells compared to P14 CD8⁺ T$_{NT}$ cells at day 22 p.i. (Fig. 5g,h). To enable direct comparison and priming of transferred P14 CD8⁺ T cells in vivo, we CRISPR knocked out both *Zfp148* and *Klf2* in naive splenic P14 CD8⁺ T cells and transferred mixtures containing equal numbers of naive P14 CD8⁺ T$_{NT}$ cells and P14 CD8⁺ T$_{ZFP148+KLF2-null}$ cells into C57BL/6 WT recipient mice followed by LCMV Cl13 infection on the same day. We observed decreased frequency of CX3CR1⁺Ly108⁻CD8⁺ T$_{EFF}$ cells, increased frequency of CD8⁺ T$_{EX}$ cells expressing exhaustion marker

**Fig. 4 | ZFP148 limits effector-driving transcription factor motif accessibility in antigen-specific CD8⁺ T cells. a**, Heatmap showing subset-specific DACRs in *Zfp148*$^{fl/fl}$ versus ZFP148 cKO splenic CD44$^{hi}$GP$_{33-41}$ Tet⁺CD8⁺ T cells at day 21 post-LCMV Cl13 infection. Genes linked to specific peaks identified by the Link() function of the Signac R package are displayed at the bottom of the heatmap. **b**, Genomic tracks showing subset-specific open chromatin regions at *Zeb2* and *S1pr5* loci in *Zfp148*$^{fl/fl}$ or ZFP148 cKO CD44$^{hi}$GP$_{33-41}$ Tet⁺CD8⁺ T cells as in **a**. DACRs (adjusted *P* < 0.05) are highlighted in orange. mRNA expression of *Zeb2* or *S1pr5* are displayed on the right. Gene-peak links, identified using the LinkPeaks() function of the Signac R package, are displayed at the bottom. **c**, Heatmap

showing subset-specific differentially accessible transcription factor motifs between *Zfp148*$^{fl/fl}$ or ZFP148 cKO CD44$^{hi}$GP$_{33-41}$ Tet⁺CD8⁺ T cells as in **a**. **d**, Volcano plot showing differentially accessible transcription factor motifs in *Zfp148*$^{fl/fl}$ versus ZFP148 cKO splenic CD8⁺ T$_{P1}$ cells (top) and CD8⁺ T$_{EFF}$ cells (bottom) as in **a**. **e**, Violin plots showing motif accessibility of TCF1, T-bet and NFATc4 in *Zfp148*$^{fl/fl}$ and ZFP148 cKO CD8⁺ T$_{P1}$, CD8⁺ T$_{P2}$, CD8⁺ T$_{EFF}$, CD8⁺ prolif T and CD8⁺ T$_{EX}$ cells as in **a**. Data are pooled from 10 mice for *Zfp148*$^{fl/fl}$ and 12 mice for ZFP148 cKO. Statistical analysis was performed using a two-sided Wilcoxon rank-sum test with Benjamini–Hochberg correction for multiple comparisons (**d**) or a two-sided Wilcoxon rank-sum test (**e**).

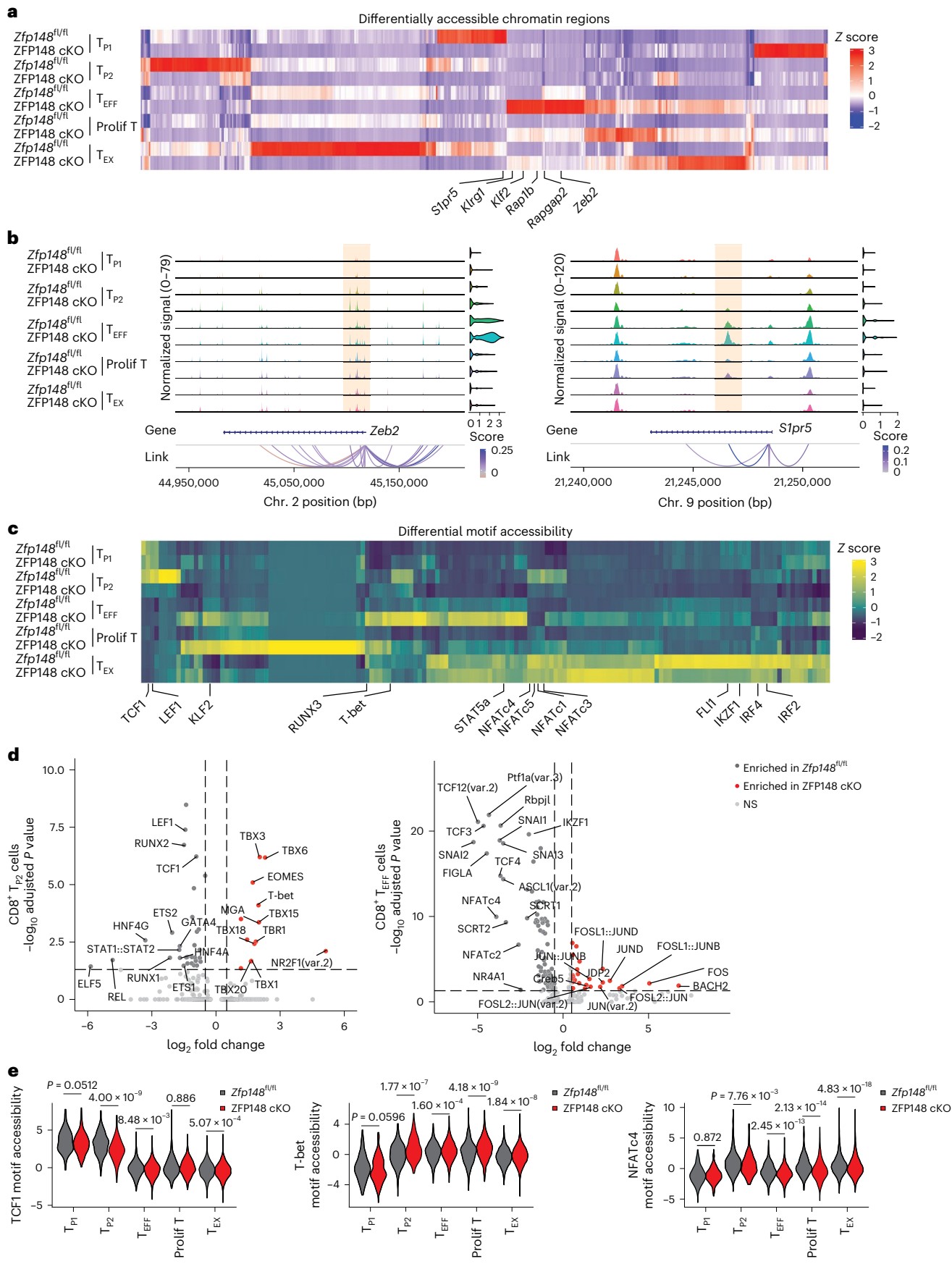

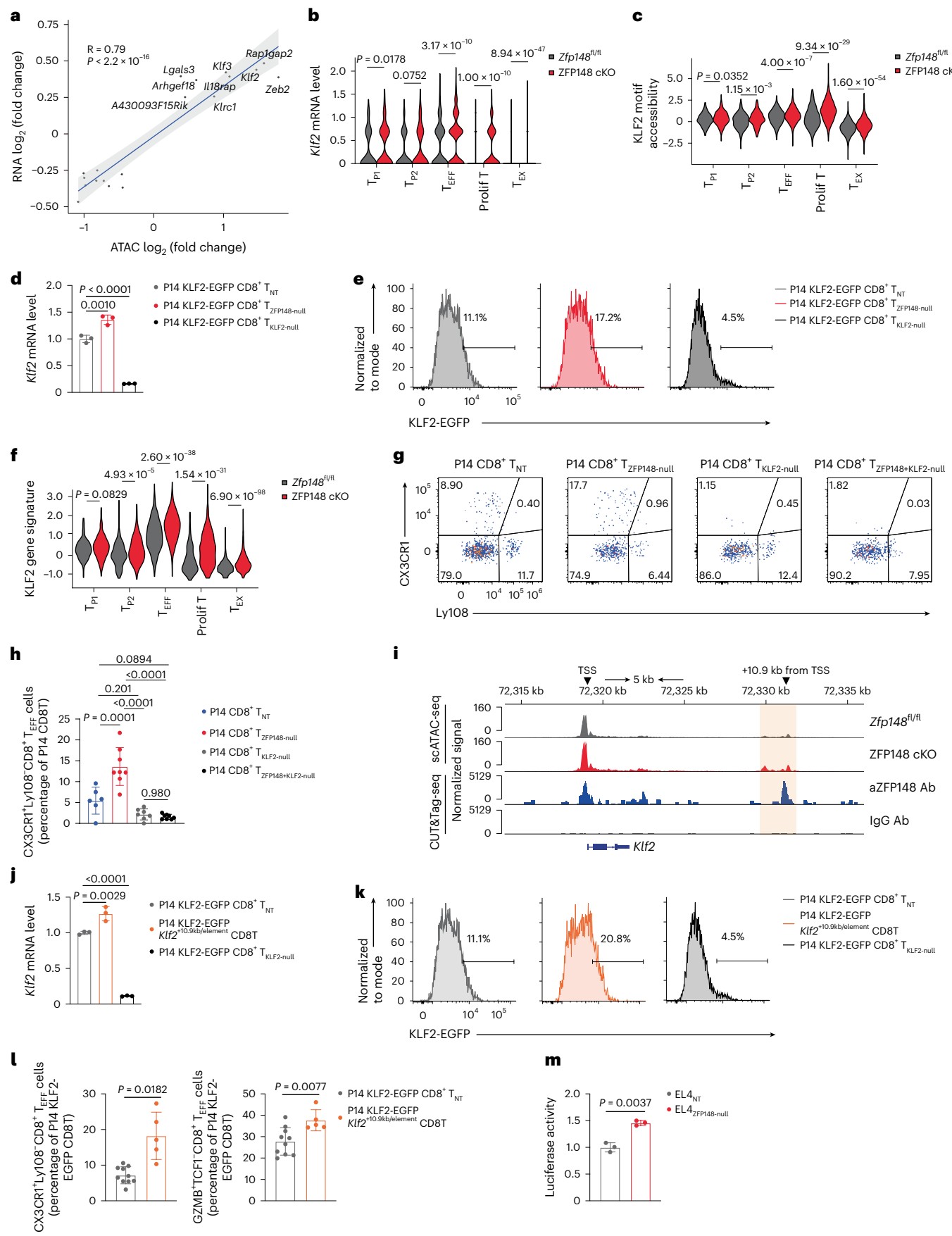

**Fig. 5 | ZFP148 represses KLF2 to inhibit CD8⁺ T_EFF differentiation. a**, log₂ fold changes of mRNA expression versus promoter chromatin accessibility in genes showing both differential expression and differential promoter chromatin accessibility in *Zfp148*^fl/fl versus ZFP148 cKO splenic CD44^hiGP_{33–41} Tet⁺CD8⁺ T cells at day 21 post-LCMV Cl13 infection. *R* and *P* values are calculated by Spearman correlation. **b,c**, Violin plots showing *Klf2* mRNA expression (**b**) and KLF2 motif accessibility (**c**) in *Zfp148*^fl/fl and ZFP148 cKO CD8⁺ T_P1, CD8⁺ T_P2, CD8⁺ T_EFF, CD8⁺ prolif T and CD8⁺ T_EX cells as in **a**. **d,e**, expression of *Klf2* mRNA (*n* = 3) (**d**) and density plots showing expression of KLF2–EGFP fusion protein (**e**) in P14 KLF2–EGFP CD8⁺ T_NT, P14 KLF2–EGFP CD8⁺ T_ZFP148-null and P14 KLF2–EGFP CD8⁺ T_KLF2-null cells activated in vitro. **f**, Violin plots showing KLF2 motif accessibility in *Zfp148*^fl/fl and ZFP148 cKO CD8⁺ T_P1, CD8⁺ T_P2, CD8⁺ T_EFF, CD8⁺ prolif T and CD8⁺ T_EX cells as in **a**. **g,h**, Scatter plots showing CX3CR1 versus Ly108 protein expression (**g**) and frequency of CX3CR1⁺Ly108⁻CD8⁺ T_EFF cells (**h**) in transferred P14 CD8⁺ T_NT cells (*n* = 6), P14 CD8⁺ T_ZFP148-null cells (*n* = 8), P14 CD8⁺ T_KLF2-null cells (*n* = 7) and P14 CD8⁺ T_ZFP148+KLF2-null cells (*n* = 8) in spleens of C57BL/6 WT recipient mice at day 22 post-LCMV Cl13 infection. **i**, Overlaid genome tracks showing chromatin accessibility of the *Klf2* loci in *Zfp148*^fl/fl and ZFP148 cKO CD44^hiGP_{33–41} Tet⁺CD8⁺

T cells as in **a** (top) and binding activity of ZFP148 measured by CUT&Tag-seq using splenic CD8⁺ T cells from C57BL/6 WT mice activated for 24 h in vitro with IgG Ab as a negative control (bottom). **j,k**, Expression of *Klf2* mRNA (*n* = 3) (**j**) and density plots showing expression of KLF2–EGFP fusion protein (**k**) in P14 KLF2–EGFP CD8⁺ T_NT, P14 KLF2–EGFP *Klf2*^+10.9kb/element CD8⁺ T and P14 KLF2–EGFP CD8⁺ T_KLF2-null cells activated in vitro. **l**, Frequency of CX3CR1⁺Ly108⁻CD8⁺ T_EFF cells and GZMB⁺TCF1⁻CD8⁺ T_EFF cells in transferred P14 KLF2–EGFP CD8⁺ T_NT (*n* = 10) and P14 KLF2–EGFP *Klf2*^+10.9kb/element CD8⁺ T cells (*n* = 5) in spleens of C57BL/6 WT recipient mice at day 22 post-LCMV Cl13 infection. **m**, Relative luciferase activity in EL4_ZFP148-null versus EL4_NT cells transfected with the pGL3-*Klf2*^+10.9kb/element luciferase vector (*n* = 3). Data are pooled from 10 mice for *Zfp148*^fl/fl and 12 mice for ZFP148 cKO (**a**–**c**, **f** and **i**); pooled from 5 mice for CUT&Tag-seq (**i**); representative of two to three independent experiments (**d**–**g**, **h**, **j** and **k**); pooled from two to three independent experiments (**l** and **m**). Data are presented as mean ± s.d. Statistical analysis was performed using a two-sided Wilcoxon rank-sum test (**b**, **c** and **f**); one-way ANOVA followed by Šídák's multiple comparisons test (**d** and **j**); one-way ANOVA followed by Tukey's multiple comparisons test (**h**) or an unpaired two-sided *t*-test (**l** and **m**).

CD39 and 2B4 and increased MFI of exhaustion marker PD-1 and TOX in transferred P14 CD8⁺ T_ZFP148+KLF2-null cells compared to P14 CD8⁺ T_NT cells at day 22 p.i. (Extended Data Fig. 8c,d). Overall, these data indicated that KLF2 is required for the augmented CD8⁺ T_EFF differentiation post-ZFP148 deletion.

To determine whether ZFP148 bound the *Klf2* locus directly, we performed ZFP148 CUT&Tag-seq in splenic CD8⁺ T cells from C57BL/6 WT mice activated for 24 h in vitro. Overlay of scATAC-seq and CUT&Tag-seq revealed increased chromatin accessibility in ZFP148 cKO versus *Zfp148*^fl/fl splenic CD44^hiGP_{33–41} Tet⁺CD8⁺ T cell and direct ZFP148 binding at the same region 10.9 kb downstream of the *Klf2* transcription start site (hereafter *Klf2*^+10.9kb/element) previously reported in the CD8⁺ T_PRO-to-T_EFF cell transition during LCMV Cl13 infection[19] (Fig. 5i). To assess the role of *Klf2*^+10.9kb/element in CD8⁺ T_EFF cell differentiation, we used CRISPR ribonucleoprotein (RNP) to delete this region in splenic P14 KLF2–EGFP CD8⁺ T cells (hereafter P14 KLF2–EGFP *Klf2*^+10.9kb/element CD8⁺ T cells) activated in vitro and transferred equal numbers of P14 CD8⁺ T_NT cells, P14 CD8⁺ T_ZFP148-null cells and P14 KLF2–EGFP *Klf2*^+10.9kb/element CD8⁺ T cells separately into C57BL/6 WT recipient mice followed by LCMV Cl13 infection on the same day. We detected increased expression of *Klf2* mRNA expression and KLF2–EGFP protein in P14 KLF2–EGFP *Klf2*^+10.9kb/element CD8⁺ T cells versus P14 CD8⁺ T_NT cells before adoptive transferring (Fig. 5j,k) and increased frequencies of CX3CR1⁺Ly108⁻ and GZMB⁺TCF1⁻CD8⁺ T_EFF cells in transferred P14 KLF2–EGFP *Klf2*^+10.9kb/element CD8⁺ T cells versus P14 CD8⁺ T_NT cells at day 22 p.i. (Fig. 5l), phenocopying P14 CD8⁺ T_ZFP148-null cells (Extended Data Fig. 8e). To test *Klf2*^+10.9kb/element-mediated transcriptional repression of *Klf2* directly, EL4 cells were CRISPR-edited with *Zfp148*-targeting (hereafter EL4_ZFP148-null) or NT (hereafter EL4_NT) sgRNAs and transduced with a pGL3 luciferase vector containing the *Klf2*^+10.9kb/element. Both endogenous ZFP148 and KLF2 were detected by flow cytometry in WT EL4 and EL4_NT cells (Extended Data Fig. 8f,g). Through efficient ZFP148 deletion

in EL4_ZFP148-null cells (Extended Data Fig. 8g), we observed increased luciferase activity in EL4_ZFP148-null cells versus EL4_NT cells (Fig. 5m), demonstrating direct *trans*-repression of ZFP148 on *Klf2* through the *Klf2*^+10.9kb/element. In summary, ZFP148 repressed *Klf2* through a distal regulatory element (*Klf2*^+10.9kb/element) and restrained KLF2-driven CD8⁺ T_EFF cell differentiation.

## ZFP148 loss synergizes with PD-1 blockade

Next, we analyzed CD8⁺ TILs in ZFP148 cKO versus *Zfp148*^fl/fl mice at day 16 postsubcutaneous injection of MC38 tumors and observed increased expression of activation marker CD44, costimulatory marker ICOS, GZMB and inhibitory receptors (PD-1, TIM-3, LAG-3, TIGIT and CD39) in ZFP148 cKO CD8⁺ TILs and a comparable frequency and number of CD8⁺ TILs in ZFP148 cKO versus *Zfp148*^fl/fl mice (Fig. 6a and Extended Data Fig. 9a,b). We also detected an increased frequency of GZMB⁺TCF1⁻CD8⁺ T_EFF cells and unchanged frequency of GZMB⁻TCF1⁺CD8⁺ T_PRO cells in ZFP148 cKO versus *Zfp148*^fl/fl mice at days 10, 14, 16 and 18 postsubcutaneous injection of MC38 tumors (Extended Data Fig. 9c). These data suggest increases in both CD8⁺ T_EX and CD8⁺ T_EFF cell differentiation in ZFP148 cKO versus *Zfp148*^fl/fl CD8⁺ TILs.

Because PD-1 signaling constrains CD8⁺ T_EFF cell functions[31], we examined the relationship between *ZNF148* and *PDCD1* mRNA expression in TCGA colorectal adenocarcinoma patients and revealed higher *PDCD1* mRNA in *ZNF148*^lo versus *ZNF148*^hi patients (Fig. 6b). To test whether ZFP148 loss sensitized CD8⁺ TILs to anti-PD-1 (aPD-1) Ab, ZFP148 cKO and *Zfp148*^fl/fl mice were treated with aPD-1 or isotype Abs at days 9, 12 and 15 postsubcutaneous injection of MC38 tumors (hereafter Iso-*Zfp148*^fl/fl, aPD-1-*Zfp148*^fl/fl, Iso-cKO and aPD-1-cKO mice). Improved tumor control and prolonged overall survival (OS) were detected in aPD-1-cKO versus aPD-1-*Zfp148*^fl/fl mice (median survival >48 versus 29 days) (Fig. 6c,d).

**Fig. 6 | ZFP148 deficiency in CD8⁺ TILs enhance PD-1 blockade efficacy. a**, MFI of CD44, ICOS, GZMB and PD-1 in CD8⁺ TILs in *Zfp148*^fl/fl (*n* = 16) and ZFP148 cKO (*n* = 11) mice at day 16 postsubcutaneous injection of MC38 tumors. **b**, *PDCD1* mRNA expression in *ZNF148*^hi (*n* = 51) versus *ZNF148*^lo (*n* = 123) colorectal adenocarcinoma patients from the TCGA database. **c**, Tumor growth curves in Iso-*Zfp148*^fl/fl (*n* = 10), aPD-1-*Zfp148*^fl/fl (*n* = 11), Iso-cKO (*n* = 5) and aPD-1-cKO (*n* = 8) mice injected subcutaneously with MC38 tumors. Mice received intraperitoneal injection of aPD-1 or IgG Ab at days 9, 12 and 15 after tumor implantation. **d**, OS of Iso-*Zfp148*^fl/fl (*n* = 10), aPD-1-*Zfp148*^fl/fl (*n* = 10), Iso-cKO (*n* = 6) and aPD-1-cKO (*n* = 8) mice as in **c**. **e,f**, UMAP visualization of clustering (**e**) and CX3CR1, GZMB, Ki-67, TCF1, ICOS, CD27, CD25 and TIM-3 protein expression (**f**) in all CD8⁺ TILs in Iso-*Zfp148*^fl/fl (*n* = 13), aPD-1-*Zfp148*^fl/fl (*n* = 18), Iso-cKO (*n* = 20) and aPD-1-cKO (*n* = 17) mice combined together at day 13 postsubcutaneous

injection of MC38 tumors. **g**, Contour plots showing distribution of CD8⁺ TILs in Iso-*Zfp148*^fl/fl, aPD-1-*Zfp148*^fl/fl, Iso-cKO and aPD-1-cKO mice as in **e**. **h,i**, Scatter plots showing GZMB versus CX3CR1 (**h**, left) or GZMB versus Perforin protein expression (**i**, left) and frequency of CX3CR1⁺GZMB⁺CD8⁺ T_EFF cells (**h**, right) or GZMB⁺Perforin⁺CD8⁺ T_EFF cells (**i**, right) in CD8⁺ TILs as in **e**. **j,k**, MFI of GZMB (left) and Perforin (right) (**j**) and frequency of proliferating BCL2⁻Ki-67⁺CD8⁺ T cells in CD8⁺ TILs (**k**) as in **e**. Data are pooled from two to three independent experiments (**a**, and **c**–**k**). Data are presented as mean ± s.d (**a**, and **h**–**k**); mean tumor areas ± s.e.m (**c**). Statistical analysis was performed using an unpaired two-sided *t*-test (**a**); a two-sided Wilcoxon rank-sum test (**b**); two-way ANOVA with Bonferroni correction for multiple comparisons (**c**); a log-rank test with Bonferroni correction for multiple comparisons (**d**) or a one-way ANOVA followed by Holm–Šídák's multiple comparisons test (**h**–**k**).

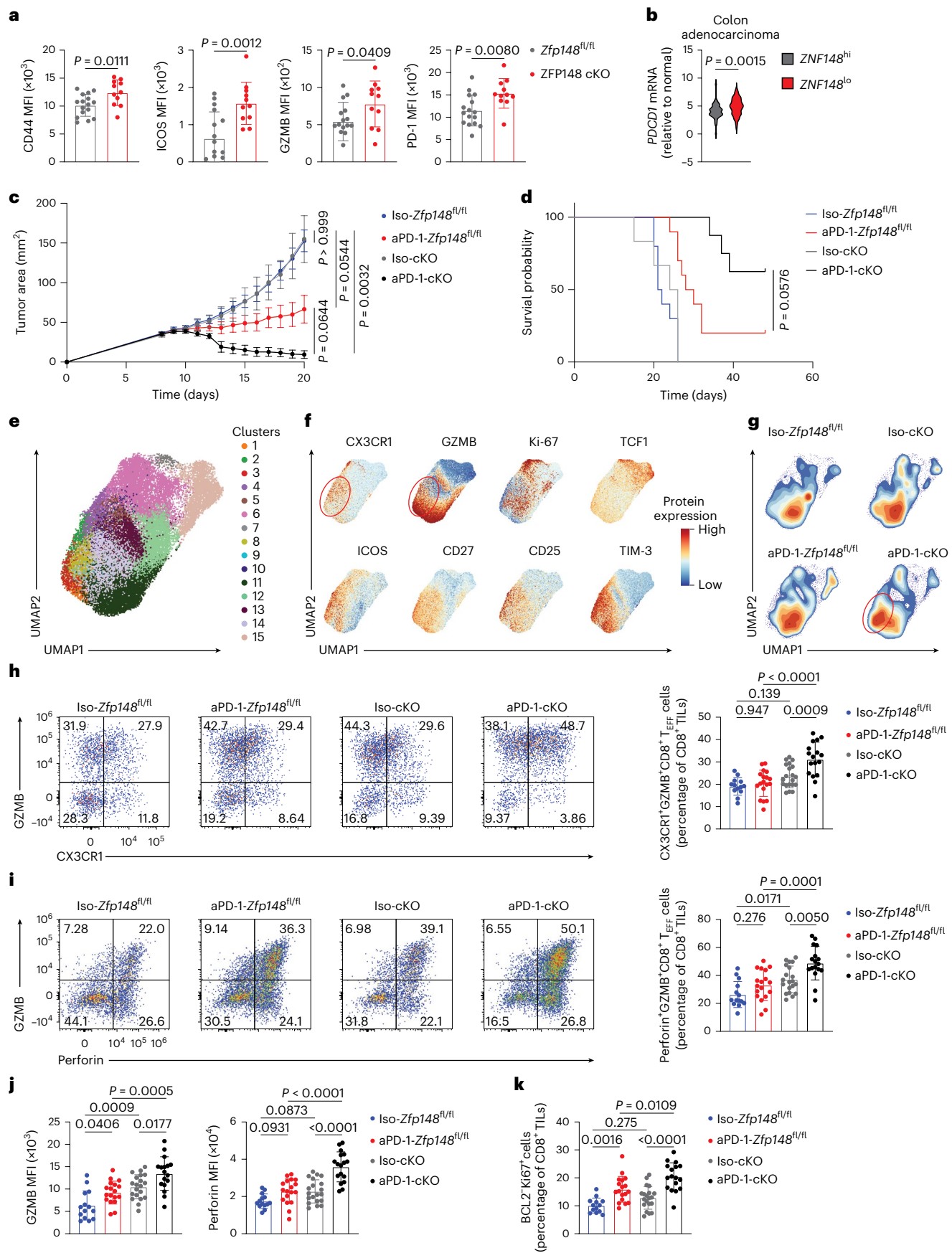

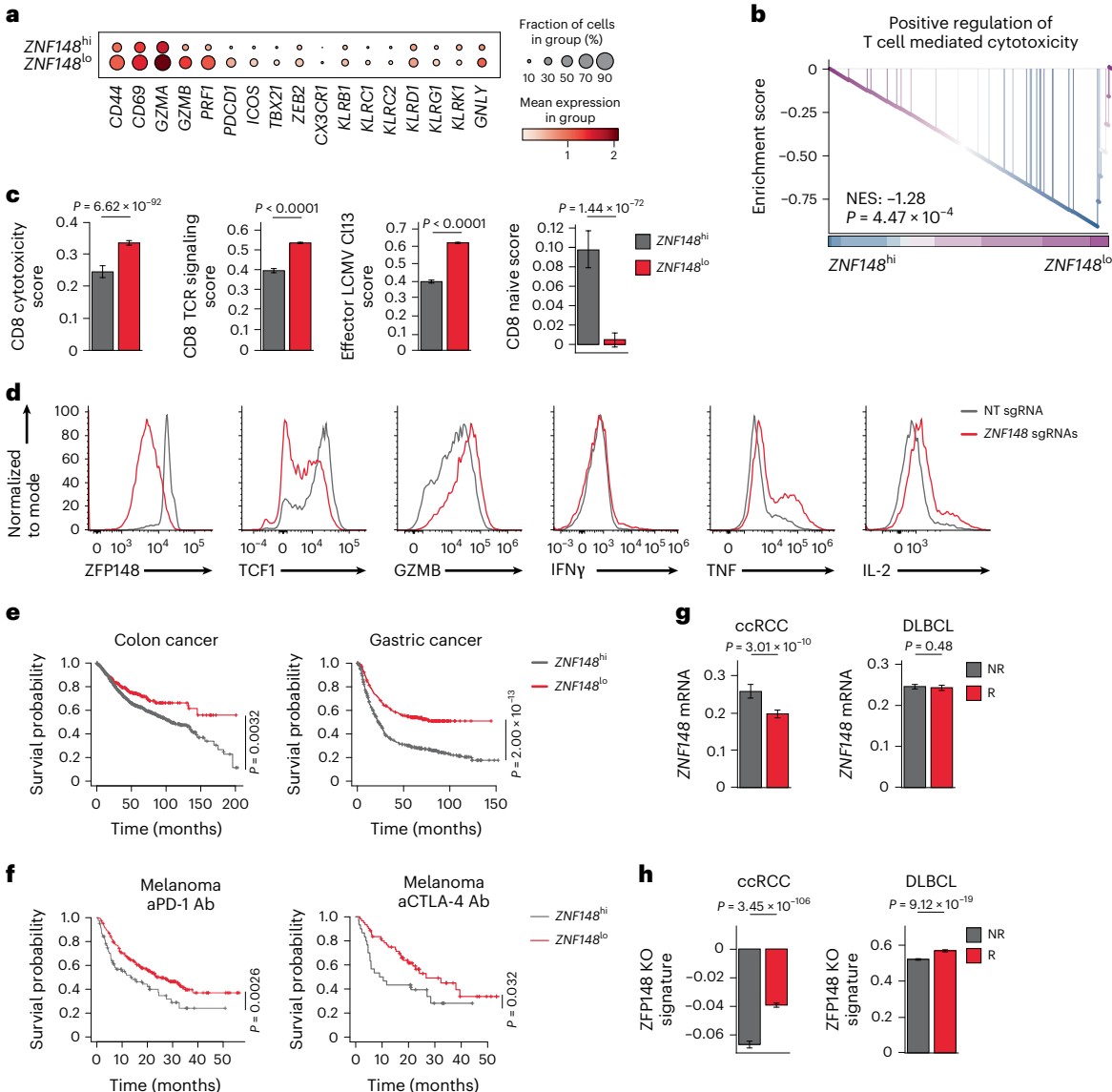

**Fig. 7 | Low *ZNF148* in human CD8⁺ TILs associates with improved immunotherapy response. a**, Dotplots showing mean expression of genes in *ZNF148*hi and *ZNF148*lo CD8⁺ TILs in the integrated human cancer scRNA-seq dataset. **b**, Gene set enrichment analysis plot showing 'Positive regulation of T cell mediated cytotoxicity' pathway in *ZNF148*hi (*n* = 895 cells) and *ZNF148*lo CD8⁺ TILs (*n* = 7,415 cells) in the integrated human cancer scRNA-seq dataset. NES, normalized enrichment score. **c**, Functional signature scores of 'CD8 cytotoxicity,' 'CD8 TCR signaling,' 'Effector LCMV Cl13' and 'CD8 naive' pathways in *ZNF148*hi and *ZNF148*lo CD8⁺ TILs in the integrated human cancer scRNA-seq dataset. **d**, Density plots showing expression of ZFP148, TCF1, GZMB, IFNγ, TNF and IL-2 protein in healthy donor PBMCs-derived human CD8⁺ T cells activated and CRISPR-edited with an NT control or *ZNF148*-targeting sgRNAs in vitro. Cytokine production was assessed following restimulation. **e**, Kaplan–Meier survival curves showing OS of colon cancer (left) or gastric

cancer (right) patients, stratified into *ZNF148*hi (colon, *n* = 787; gastric, *n* = 577) and *ZNF148*lo groups (colon, *n* = 274; gastric, *n* = 298). **f**, Kaplan–Meier survival curves showing OS of melanoma patients treated with aPD-1 or aCTLA-4 Ab, stratified into *ZNF148*hi (aPD-1, *n* = 83; aCTLA-4, *n* = 30) and *ZNF148*lo groups (aPD-1, *n* = 274; aCTLA-4, *n* = 61). **g,h**, Expression of *ZNF148* mRNA (**g**) or the ZFP148 KO signature (**h**) in CD8⁺ TILs in responders (R) versus nonresponders (NR) of advanced ccRCC patients treated with aPD-1 + aCTLAT-4 Abs[37] (left, NR: 5,150 cells from five patients, R: 12,561 cells from eight patients) and DLBCL patients treated with anti-CD19 CAR-T cells[38] (right, NR: *n* = 63,482 cells from 57 patients; R: *n* = 59,351 cells from 52 patients). Data are representative of two independent experiments (**d**). Data are presented as mean ± s.d. Statistical analysis was performed using a two-sided permutation-based enrichment test with false discovery rate (FDR) adjustment (**b**); two-sided Wilcoxon rank-sum test (**c**, **g** and **h**) or a log-rank test (**e** and **f**).

At day 13 postsubcutaneous injection of MC38 tumors, flow cytometry detected increased frequency and number of total CD8⁺ TILs in aPD-1-cKO versus Iso-cKO mice (Extended Data Fig. 9d,e). Both uniform manifold approximation and projection (UMAP) analysis and conventional gating further identified increased frequencies of CX3CR1⁺GZMB⁺CD8⁺ T_EFF and GZMB⁺Perforin⁺ CD8⁺ T_EFF cells and higher production of cytolytic molecule GZMB and Perforin in aPD-1-cKO versus aPD-1-*Zfp148*fl/fl CD8⁺ TILs (Fig. 6e–j). Production of IFNγ and TNF was unchanged, whereas IL-2 production was reduced in aPD-1-cKO

versus aPD-1-*Zfp148*fl/fl CD8⁺ TILs (Extended Data Fig. 9f). aPD-1-cKO CD8⁺ TILs also showed increased expression of costimulatory marker CD27, ICOS and IL-2 receptor α chain CD25, indicating reinforced costimulation and IL-2 signaling (Fig. 6f,g and Extended Data Fig. 9g,h), and increased frequency of proliferating BCL2⁻Ki-67⁺CD8⁺ T cells (Fig. 6k), whereas minor and nondirectional differences in the expression of inhibitory receptors (TIM-3, CTLA-4, CD39 and VISTA) (Extended Data Fig. 9i), compared with aPD-1-*Zfp148*fl/fl CD8⁺ TILs. Collectively, ZFP148 deficiency synergized with aPD-1 Ab to preferentially

expand a proliferative, cytolytic CD8[+] T$_{EFF}$ population, resulting in superior tumor control.

## Low *ZNF148* associates with immunotherapy response

To examine the role of ZFP148 in human CD8[+] T cells, we annotated CD8[+] TILs from published scRNA-seq datasets of treatment-naive patients across 17 cancer types (hereafter integrated human cancer scRNA-seq dataset) with a published classification scheme[32] and stratified them into *ZNF148*[lo], *ZNF148*[int] and *ZNF148*[hi] groups (Extended Data Fig. 10a,b). We detected increased expression of effector genes (*GZMA*, *GZMB*, *PRF1*) and NK cell receptors (*KLRD1*, *GNLY*) in *ZNF148*[lo] versus *ZNF148*[hi] total CD8[+] TILs (Fig. 7a) and CD8[+] T$_{EFF}$ subsets ('CD8_c2_Teff' and 'CD8_c8_Teff_KLRG1') (Extended Data Fig. 10c). We also observed enhanced expression of 'Positive regulation of T cell-mediated cytotoxicity,' 'CD8 cytotoxicity,' 'CD8 TCR signaling' and 'Effector LCMV Cl13' signatures, and reduced expression of a 'CD8 naive' signature, in *ZNF148*[lo] versus *ZNF148*[hi] CD8[+] TILs (Fig. 7b,c), indicating an inverse relationship between *ZNF148* expression and human CD8[+] T$_{EFF}$ cell differentiation.

We next CRISPR-edited peripheral blood mononuclear cells (PBMCs)-derived human CD8[+] T cells activated in vitro with *ZNF148*-targeting or NT sgRNAs. We detected reduced expression of TCF1 and increased expression of GZMB, TNF and IL-2 in *ZNF148*-targeting sgRNA-edited versus NT sgRNA-edited human CD8[+] T cells (Fig. 7d). By using the Kaplan–Meier plotter webserver, we observed improved OS in *ZNF148*[lo] versus *ZNF148*[hi] cancer patients[33] (Fig. 7e and Supplementary Table 1). We also generated a ZFP148 KO signature from genes upregulated in ZFP148 cKO versus *Zfp148*[fl/fl] CD44[hi]GP$_{33-41}$ Tet[+]CD8[+] T cells in our in-house scRNA-seq dataset and stratified a published liver cancer patient cohort[34] based on ZFP148 KO signature enrichment in the CD8[+] T$_{EFF}$ subset of CD8[+] TILs ('CD8T_04_GNLY'). Consistently, we detected prolonged OS in ZFP148 KO signature[hi] versus ZFP148 KO signature[lo] patients (Extended Data Fig. 10d). Relevant to immunotherapies, we observed prolonged OS in *ZNF148*[lo] versus *ZNF148*[hi] melanoma patients stratified by *ZNF148* expression pre-aPD-1 or pre-aCTLA-4 Ab treatments[35] (Fig. 7f). In separate cohorts, we also observed prolonged OS in ZFP148 KO signature[hi] versus ZFP148 KO signature[lo] melanoma patients treated with aPD-1 or aCTLA-4 Ab[36] (Extended Data Fig. 10e). Consistently, by reanalyzing scRNA-seq datasets of patients with clear cell renal cell carcinoma (ccRCC) treated with aPD-1 plus aCTLA-4 Abs[37] or diffuse large B-cell lymphoma (DLBCL) treated with anti-CD19 CAR-T cells[38], we found lower *ZNF148* expression or higher ZFP148 KO signature scores in responder (R) versus nonresponder (NR) CD8[+] TILs (Fig. 7g,h). Thus, reduced *ZNF148* or elevated ZFP148 KO signature might predict improved responsiveness to immunotherapies.

Finally, higher expression of *KLF2* mRNA was found in *ZNF148*[lo] versus *ZNF148*[hi] colorectal adenocarcinoma patients from the TCGA database (Extended Data Fig. 10f). Analysis of the integrated human cancer scRNA-seq dataset showed higher expression of *KLF2* mRNA and KLF2 gene signature[30] in *ZNF148*[lo] versus *ZNF148*[hi] CD8[+] TILs (Extended Data Fig. 10g). We also detected improved OS in *KLF2*[hi] or KLF2 gene signature[hi] versus to *KLF2*[lo] or KLF2 signature[lo] patients, respectively, in melanoma patients treated with aPD-1 or aCTLA-4 Abs[36] (Extended Data Fig. 10h,i). Collectively, these results establish ZFP148 as a conserved negative checkpoint that restrains CD8[+] T$_{EFF}$ differentiation in humans, with direct implications for cancer immunotherapy.

## Discussion

Here we showed that ZFP148 restrained antigen-specific CD8[+] T cell cytolytic effector differentiation by repressing KLF2. Conditional deletion of ZFP148 enriched frequency of CD8[+] T$_{EFF}$ cells and synergized with PD-1 blockade, resulting in superior tumor control.

We found that ZFP148 modulated chromatin accessibility at binding motifs for several lineage-determining transcription factors (TCF1, T-bet, KLF2, BATF, IRFs and IKZF1) in a subset-specific manner and

repressed KLF2 directly through a distal regulatory element within the *Klf2* loci in CD8[+] T cells. In mouse embryonic fibroblasts, ZFP148 deficiency induces p53-dependent proliferative arrest partially through regulating other transcription factors[39]. Furthermore, ZFP148 can act as a binding partner for transcription factors NF-1[40], GATA1[41] and ZFP281[23] or compete with SP1[42] in various types of cells other than CD8[+] T cells. Thus, future studies are needed to define the binding motifs and cofactors of ZFP148 to elucidate its precise downstream regulations in CD8[+] T cells.

Our data did not exclude the possibility that ZFP148 may alternatively promote the formation or maintenance of KLF2[-]CD8[+] T$_{PRO}$ and CD8[+] T$_{EX}$ cells instead of driving KLF2[+]CD8[+] T$_{EFF}$ cell differentiation. In the absence of ZFP148, both the frequency and number of CD8[+] T$_{PRO}$ and CD8[+] T$_{EX}$ cells were reduced, potentially shifting the differentiation of CD8[+] T cells toward CD8[+] T$_{EFF}$ cells. This model requires further validation using lineage-tracing approaches and temporal deletion of ZFP148.

Despite the transformative impact of PD-1/programmed death ligand 1 (PD-L1) blockade, durable response in cancer patients is limited[43]. Here we found synergy between conditional ZFP148 deletion and PD-1 blockade, paralleling observations from dual checkpoint blockade strategies[44,45]. These findings suggest complementary roles for ZFP148 and PD-1 that ZFP148 restrains primarily cytolytic effector differentiation, whereas PD-1 signaling limits proliferation[46,47], of CD8[+] T cells.

The effect of ZFP148 conditional deletion seems to be context dependent. ZFP148 deficiency led to reduction in CD8[+] T$_{PRO}$ cells during LCMV Cl13 infection but not in the MC38 tumor model, probably reflecting distinct antigenic landscapes, as chronic viral infection imposes uniform, high antigen load[48], whereas tumors present heterogeneous and reduced antigen levels due to MHC-I downregulation[49]. As ZFP148 induction depends on TCR signal strength, these contextual differences probably shape the expression and function of ZFP148. Future studies are needed to examine how ZFP148 regulates CD8[+] T cell differentiation across diverse human disease settings to determine the clinical application of targeting this pathway.

## Online content

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

[1]Pelotonia Institute for Immuno-Oncology, The Ohio State University Comprehensive Cancer Center—James Cancer Center and Solove Research Institute, Columbus, OH, USA. [2]Department of Urology, Cedars–Sinai Medical Center, Los Angeles, CA, USA. [3]Department of Biomedical Sciences, Cedars–Sinai Medical Center, Los Angeles, CA, USA. [4]Department of Pathology, Feinberg School of Medicine, Northwestern University, Chicago, IL, USA. [5]Center for Human Immunobiology, Feinberg School of Medicine, Northwestern University, Chicago, IL, USA. [6]Department of Microbiology and Immunology, Medical College of Wisconsin, Milwaukee, WI, USA. [7]Department of Biomedical Informatics, The Ohio State University College of Medicine, Columbus, OH, USA. [8]Department of Microbial Infection and Immunity, The Ohio State University College of Medicine, Columbus, OH, USA. [9]Department of Urology, Division of Urologic Oncology, The Ohio State University Comprehensive Cancer Center—James Cancer Center and Solove Research Institute, Columbus, OH, USA. [10]University of Arizona Cancer Center, Tucson, AZ, USA. [11]Departments of Medicine and Biomedical Sciences, Samuel Oschin Comprehensive Cancer Institute, Cedars–Sinai Medical Center, Los Angeles, CA, USA. [12]Department of Internal Medicine, Division of Medical Oncology, The Ohio State University, Columbus, OH, USA. ✉e-mail: Zihai.li@osumc.edu

## Methods

### Ethics

All research conducted in this study complied with ethical regulations and was approved by The Ohio State University Institutional Animal Care and Use Committee (IACUC; protocol 2019A00000075), Institutional Biosafety Committee (IBC; protocol 2019R00000046), and Institutional Review Board (Buck-IRB; protocol 2018C0181).

### Mice

C57BL/6J (strain 000664) mice were obtained from the Jackson Laboratory. CD8[+] T cell-specific ZFP148-deficient mice were generated by crossing E8i[Cre] (the Jackson Laboratory, strain 008766) mice with *Zfp148*[fl/fl] mice kindly provided by J. L. Merchant at University of Arizona. P14 mice were kindly provided by S. M. Kaech at The Salk Institute for Biological Studies. KLF2[GFP] reporter mice were kindly provided by S. C. Jameson at the University of Minnesota through W. Cui at Northwestern University and crossed with P14 mice at Northwestern University to generate P14 KLF2–EGFP mice. Mice were maintained in the animal facility at The Ohio State University under standard conditions (ambient temperature 20–24 °C, relative humidity 30–70%, 12 h dark–light cycle (lights on from 6:00 to 18:00)). Both male and female mice aged 8–10 weeks were used. All procedures were performed in strict accordance with the NIH Guide for the Care and Use of Laboratory Animals and approved by the Committee on the Ethics of Animal Experiments of The Ohio State University.

### Cell lines

B16-GP cell line was kindly provided by A. Wieland at The Ohio State University. B16-GP cells were cultured in RPMI-1640 medium (Gibco, cat. no. 11875-093) with 10% heat-inactivated fetal bovine serum (FBS) (Gibco, cat. no. 10082-147) and 1% penicillin/streptomycin (Pen-Strep; Gibco, cat. no. 15140-122). The MC38 cell line was purchased from Kerafast (cat. no. ENH204-FP). MC38 cells were cultured in Dulbecco's modified Eagle's medium (DMEM; Gibco, cat. no. 11965-092) with 10% FBS and 1% Pen-Strep. EL4 cell line was kindly provided by K. Oestreich at The Ohio State University. EL4 cells were cultured in RPMI-1640 with 10% FBS and 1% Pen-Strep. Cell lines were tested regularly for mycoplasma contamination.

### Tumor challenge

MC38 cells ($1.5 \times 10^6$) were resuspended in 100 μl of ice-cold phosphate-buffered saline (PBS) for subcutaneous injection into the right flanks of shaved mice. For experiments involving PD-1 blockade, mice were treated with aPD-1 Ab (BioXCell, 200 μg, clone 29F.1A12X) on days 9, 12 and 15 after tumor implantation. Tumors were monitored and measured daily using an electronic caliper starting on day 8 after tumor injection until day 20. Tumor surface area was calculated using the formula (width × length in square millimeters). Maximal tumor size cutoff was 16 mm in diameter for non-endpoint studies. For survival analysis, tumor-bearing mice were monitored daily and euthanized as nonsurvivors when the tumors sizes reached 16 mm in diameter.

### LCMV infection models

For acute LCMV infection, 8- to 10-week-old mice were injected intraperitoneally with $2 \times 10^5$ plaque-forming units of LCMV Armstrong. For chronic LCMV infection, 8- to 10-week-old mice were injected intravenously with $2 \times 10^6$ plaque-forming units of LCMV Cl13. Viruses were prepared by a single passage on BHK21 cells and viral titers were determined by plaque formation assay on Vero cells. Serum virus titers were determined by the plaque assay performed using Vero cells as described previously[50].

### Tissue processing and single-cell suspension preparation

Mouse spleens were disrupted mechanically, washed once with ice-cold PBS, subjected to red blood cell lysis (BioLegend, cat. no. 420302),

passed through 70-μm cell strainers, and resuspended as single-cell suspensions.

For liver lymphocyte isolation, mice were perfused with ice-cold PBS via the hepatic portal vein. Livers were dissociated mechanically through 100-μm strainers, followed by lymphocyte isolation using Percoll gradient centrifugation (44% Percoll in RPMI over 67% Percoll in PBS; 450*g* for 20 min at room temperature) and red blood cell lysis.

For lung lymphocyte isolation, mice were perfused with ice-cold PBS. Lungs were minced and digested with Collagenase Type III (Worthington Biochemical Corporation, cat. no. LS004182) for 90 min at 37 °C, passed through 70-μm strainers, and lymphocytes were isolated by Percoll gradient centrifugation (44% Percoll in RPMI over 67% Percoll in PBS; 500*g* for 20 min at room temperature).

Tumors were dissociated mechanically and digested with Collagenase Type I and Collagenase Type IV (Worthington Biochemical Corporation, cat. nos. LS004196 and LS004188) for 30 min at 37 °C with shaking at 125 rpm. Digestion was quenched with ice-cold PBS containing 2% bovine serum albumin (BSA), and red blood cell lysis (BioLegend) was performed as needed before filtration through 70-μm strainers.

### Human bladder tumor and peripheral blood sample processing

Muscle-invasive bladder tumor samples were obtained from consenting treatment-naive patients (the patient cohort had a median age of 68 years) enrolled prospectively in The Ohio State University Comprehensive Cancer Center bladder cancer tissue registry (Buck-IRB 2018C0181). Clinical specimens were processed on the day of radical cystectomy. Tumor tissue was dissociated manually, centrifuged (600*g* for 5 min at 4 °C), resuspended in human tumor dissociation enzyme solution (Miltenyi Biotec, cat. no. 130-095-929) and homogenized using a gentleMACS semi-automated dissociator (Miltenyi Biotec). Homogenized tissue was incubated at 37 °C for 20 min under continuous rotation on a MACSmix Tube Rotator (Miltenyi Biotec). Following addition of 2% BSA, single-cell suspensions were filtered twice through a 70-μm cell strainer, washed with PBS, subjected to red blood cell lysis (BioLegend, cat. no. 420302), and resuspended in RPMI medium.

Peripheral blood was collected in heparin–EDTA tubes and processed by Ficoll–Hypaque (Sigma) density gradient centrifugation to isolate PBMCs.

### Flow cytometry

Single-cell suspensions were stained at 4 °C with LIVE/DEAD Fixable Blue viability dye (Invitrogen, cat. no. L23105) for 15 min, followed by simultaneous Fc receptor (FcR) blocking and surface marker staining for 30 min at 4 °C. Intracellular staining was performed using the Foxp3 Transcription Factor Staining Kit (Invitrogen, cat. no. 00-5523-00) according to the manufacturer's instructions. Data were acquired on a five-laser Cytek Aurora high-dimensional flow cytometer.

For cytokine detection in CD8[+] TILs and human CD8[+] T cells, cells were restimulated with Cell Stimulation Cocktail (Invitrogen, cat. no. 00-4970-03) containing PMA and ionomycin in the presence of brefeldin A (BioLegend, cat. no. 420601) for 2 h at 37 °C. For detection of cytokine production by $GP_{33–41}$-specific CD8[+] T cells from spleens of LCMV-infected mice, splenocytes were restimulated with $GP_{33–41}$ peptide in the presence of brefeldin A for 5 h at 37 °C.

Apoptosis was assessed by Annexin V and PI staining using the fluorescein isothiocyanate Annexin V Apoptosis Detection Kit with PI (BioLegend, cat. no. 640914), followed by FcR blocking and surface marker staining. Data were analyzed using FlowJo software (v.10.7.1, Tree Star) or OMIQ Flow Cytometry software (Dotmatics). Antibodies used for multispectral flow cytometry are listed in Supplementary Table 3.

### CD8[+] T cell isolation and stimulation in vitro

Spleens from C57BL/6J WT mice were processed into single-cell suspensions, and untouched CD8[+] T cells were purified by negative selection

(STEMCELL, cat. no. 19853). For time-course TCR stimulation assays, CD8[+] T cells were cultured in complete T cell medium (cTCM) (RPMI-1640 (Gibco, cat. no. 11875-093) with 10% FBS, 1% Pen-Strep, 1 mM sodium pyruvate (Gibco, cat. no. 11360-070), 1× MEM NEAA (Gibco, cat. no. 11140-050), 10 mM HEPES (Gibco, cat. no. 15630-080) and 50 μM 2-mercaptoethanol (Gibco, cat. no. 21985-023)) supplemented with 100 U ml[−1] recombinant human IL-2 (rhIL-2, acquired from the Biological Resources Branch at the NIH) and stimulated with 5 μg ml[−1] plate-bound aCD3 Ab (BioLegend, cat. no. 100359), with or without 2 μg ml[−1] soluble aCD28 Ab (BioLegend, cat. no. 102121), at $1 \times 10^6$ cells per well in 24-well plates for 0, 10, 24 or 48 h at 37 °C and 5% $CO_2$. For dose–response assays, cells were stimulated with 0.1–5 μg ml[−1] plate-bound aCD3 Ab plus 2 μg ml[−1] soluble aCD28 Ab for 48 h under identical culture conditions. For CsA inhibition experiments, cells were stimulated with 5 μg ml[−1] plate-bound aCD3 Ab and 2 μg ml[−1] soluble aCD28 Ab in the presence of 0, 1, 10, 100, 1,000 or 10,000 nM cyclosporine A (Thermo Scientific Chemicals, cat. no. 457970010) for 48 h.

### CRISPR–Cas9 RNP KO in activated P14 CD8[+] T cells and adoptive transferring

sgRNAs targeting *Klf2* were adapted from a published study[51], whereas sgRNAs for other candidates were designed by Integrated DNA Technologies (IDT); all were purchased from IDT. Sequences are listed in Supplementary Table 2. For experiments using P14 or P14 KLF2–EGFP mice, on day 0, fresh splenocytes were isolated and stimulated with 1 μg ml[−1] LCMV GP$_{33–41}$ peptide (GenScript, cat. no. RP20257) in the presence of 100 U ml[−1] rIL-2 (Biological Resources Branch at NIH) in cTCM at $1 \times 10^6$ cells ml[−1] in 24-well plates. After 3 days, cells were collected and counted using a Bio-Rad TC20 automated cell counter. Cas9 (IDT, cat. no. 1081059) and sgRNAs were combined and incubated at room temperature for 20 min. Two sgRNAs were used per target to increase KO efficiency. Electroporation was performed using the 4D-Nucleofector 4 Core Unit (Lonza) and P4 primary cell 4D-Nucleofector kit S (Lonza, cat. no. V4XP-4032) with program CM137. Following electroporation, cells were kept at room temperature for 10 min, after which 200 μl prewarmed cTCM was added to each well. Cells were then rested at 37 °C and 5% $CO_2$ for 2 h, counted, resuspended in cTCM with 100 U ml[−1] rhIL-2 at $0.5 \times 10^6$ cells ml[−1] in 24-well plates, and returned to the incubator. Two days postelectroporation, CRISPR-edited P14 or P14 KLF2–EGFP CD8[+] T cells were collected for flow cytometry analysis or 5,000 cells from each condition were adoptively transferred into C57BL/6 WT recipient mice by intravenous injection, followed by LCMV Cl13 infection on the same day.

### CRISPR–Cas9 RNP KO of genomic region in activated P14 KLF2–EGFP CD8[+] T cells

sgRNAs targeting the distal regulatory element of *Klf2* (*Klf2*$^{+10.9kb/element}$) were designed and purchased from IDT (sequences in Supplementary Table 2). CRISPR–RNP transfection was performed as described in the 'CRISPR–Cas9 RNP KO in activated P14 CD8[+] T cells and adoptive transferring' section.

Targeting efficiency on genomic DNA was assessed by PCR amplification of 50 ng genomic DNA (extracted using New England Biolabs, cat. no. T3010L) using Platinum SuperFi II PCR Master Mix (Thermo Fisher, cat. no. 12369010). PCR products were purified (Qiagen, cat. no. 28506) and Sanger sequenced (Azenta). Editing efficiency was assessed using the Inference of CRISPR edits tool (Synthego).

Two days postelectroporation, cells were subjected to RNA extraction and quantitative PCR analysis, flow cytometry, or adoptive transfer into C57BL/6J WT recipient mice as described above.

### Human CD8[+] T cell activation and CRISPR–Cas9 RNP KO

Naive human CD8[+] T cells were isolated from cryopreserved healthy donor PBMCs using EasySep immunomagnetic negative selection kits (STEMCELL, cat. no. 17953). Cells were resuspended in cTCM with 100 U ml[−1] rhIL-2 at $1 \times 10^6$ cells ml[−1] and stimulated with Dynabeads Human T-Activator CD3/CD28 (Gibco, cat. no. 11131D). On day 3 post-stimulation, cells were collected and counted. sgRNAs were designed and purchased from IDT (sequences in Supplementary Table 2).

Electroporation was performed using the 4D-Nucleofector 4 Core Unit and P2 primary cell 4D-Nucleofector kit S (Lonza, cat. no. V4XP-2024) with program DN100. Cells were rested at room temperature for 10 min, recovered with 200 μl prewarmed cTCM per well, incubated at 37 °C and 5% $CO_2$ for 2 h, then counted and replated in cTCM with 100 U ml[−1] rhIL-2 at $0.5 \times 10^6$ cells ml[−1]. Two days postelectroporation, CRISPR-edited human CD8[+] T cells were collected for flow cytometry.

### Quantitative PCR

Total RNA was extracted using Direct-zol RNA Microprep Kits (Zymo Research, cat. no. R2062) according to the manufacturer's instructions. For each sample, 1 μg total RNA was reverse transcribed using the iScript cDNA Synthesis Kit (Bio-Rad, cat. no. 1708891) in a 20 μl reaction. qPCR was performed using SsoAdvanced Universal SYBR Green Supermix (Bio-Rad, cat. no. 1725272) on a StepOne Real-Time PCR System (Applied Biosystems). Primers were purchased from IDT: *Klf2* (forward: 5′-ACCAACTGCGGCAAGACCTA-3′; reverse: 5′-CATCCTTCCCAGTTGCAATGA-3′)[51]; β-actin (forward: 5′-AGCTGAGAGGGAAATCGTGC-3′; reverse: 5′-TCCAGGGAGGAA GAGGATGC-3′)[24].

### Dual reporter luciferase assay

A 1,913-bp genomic region spanning *Klf2*$^{+10.9kb/element}$ was cloned into the pGL3 Promoter Vector (Addgene, cat. no. 212939) 15 bp upstream of the SV40 minimal promoter by GenScript to generate the pGL3-*Klf2*$^{+10.9kb/element}$ vector.

Two days before electroporation, EL4 cells were seeded at $3 \times 10^5$ cells ml[−1] in T-75 flasks. On the day of electroporation, EL4 cells were collected and counted. CRISPR–RNPs were prepared with an NT sgRNA or the same pair of *Zfp148*-targeting sgRNAs as in mouse CD8[+] T cells (sequences in Supplementary Table 2). Electroporation was performed using the 4D-Nucleofector system and SF Cell Line 4D-Nucleofector X Kit S (Lonza, cat. no. V4XC-2032) with program CM120. Electroporated EL4 cells were rested at room temperature for 10 min and recovered in prewarmed electroporation medium (RPMI-1640 with 10% FBS).

For luciferase transfection, 2 days after CRISPR–RNP electroporation, $4 \times 10^5$ EL4$_{NT}$ or EL4$_{ZFP148-null}$ cells were transfected with 3 μg pGL3-*Klf2*$^{+10.9kb/element}$ and 20 ng SV40-Renilla vectors using program CM120. After 24 h, cells were collected and luciferase activity was measured using the Dual-Luciferase Reporter Assay System (Promega, cat. no. E1910) on a SpectraMax iD3 reader in technical duplicates.

### Incucyte cytotoxicity assay

B16-GP cells were seeded at $2 \times 10^3$ cells per well in flat-bottom 96-well plates and incubated at 37 °C for 30 min in the presence of Incucyte Annexin V Green Reagent (Sartoris, cat. no. 4642). Following initial imaging on the IncuCyte S3 Live-Cell Analysis System, CD8[+] T cells were added: (1) mixed CD44$^{hi}$GP$_{33–41}$ Tet[+] and CD44$^{hi}$GP$_{276–286}$ Tet[+]CD8[+] T cells sorted from spleens of *Zfp148*$^{fl/fl}$ or ZFP148 cKO mice at day 22 post-LCMV Cl13 infection (E:T = 25:1), or (2) P14 CD8[+] T cells CRISPR-edited with NT or *Zfp148*-targeting sgRNAs (E:T = 5:1). Images were acquired every 2 h and analyzed using IncuCyte S3 software.

### Comparison of *KLF2* and *PDCD1* mRNA expression in *ZNF148*$^{hi}$ versus *ZNF148*$^{lo}$ patient cohorts

mRNA expression of *PDCD1* and *KLF2* was compared between *ZNF148*$^{hi}$ and *ZNF148*$^{lo}$ patients from the TCGA colorectal adenocarcinoma dataset. *ZNF148*$^{hi}$ was defined as *ZNF148* mRNA expression *z* scores greater than 2 (*n* = 51) relative to normal and *ZNF148*$^{lo}$ as *z* scores less than −2 (*n* = 123) relative to normal.

## Single-cell multiome library preparation

CD44$^{hi}$GP$_{33-41}$ Tet$^+$CD8$^+$ T cells were FACs-sorted from the spleens of *Zfp148*$^{fl/fl}$ or ZFP148 cKO mice at day 21 post-LCMV C13 infection. After sorting, cells were washed with PBS containing 0.04% BSA, then approximately 10,000 nuclei of either *Zfp148*$^{fl/fl}$ or ZFP148 cKO sample were isolated and processed with the Chromium Single Cell Multiome ATAC+ Gene Expression Reagent Kit (10x genomics, cat. no. 1000283) following the manufacturer's manual. GEX and ATAC libraries were generated per manufacturer instructions, quality controlled by TapeStation and sequenced on an Illumina NovaSeq X Plus platform (Azenta Life Sciences).

## Single-cell multiomic sequencing alignment and downstream analysis

scRNA-seq and scATAC-seq data were processed using the 10x Genomics Cell Ranger ARC pipeline (v.2.0.2) and aligned to the mm10 reference genome. Downstream analyses were performed in R (v.4.4.0) using Seurat (v.5.1.0) and Signac (v.1.14.0) with default parameters unless otherwise specified. For quality control, high-quality cells were defined as those with ATAC counts between $5 \times 10^3$ and $7 \times 10^5$, RNA counts between 1,000 and 25,000, and mitochondrial gene expression <20%. For the RNA modality, standard Seurat preprocessing was performed, including SCTransform normalization, principal component analysis (RunPCA), and dimensionality reduction using UMAP (RunUMAP). For the ATAC modality, preprocessing included term frequency–inverse document frequency normalization (RunTFIDF), feature selection (FindTopFeatures), singular value decomposition (RunSVD) and UMAP projection. To integrate modalities, we applied the weighted nearest neighbor (WNN) algorithm using the FindMultiModalNeighbors function in Seurat to construct a joint neighbor graph representing weighted contributions of RNA and ATAC modalities. WNN clusters were identified at a resolution of 0.1. Cell types were annotated based on mRNA expression of canonical marker genes and signature scores derived from previously published genesets calculated using AUCell (v.1.26.0). Gene activity scores were computed from chromatin accessibility data using the GeneActivity function in Signac. Differential gene expression analyses were performed on RNA or gene activity matrices using FindAllMarkers or FindMarkers with min.pct = 0.25, filtered at log$_2$ fold change ≥ 0.25 and adjusted $P$ < 0.05. DACRs were identified using FindAllMarkers or FindMarkers with logistic regression testing, min.pct = 0.05, log$_2$ fold change ≥ 0.25 and adjusted $P$ < 0.05, including total counts as a latent variable. Genes linked to DACRs were identified using the Links() function in Signac. GO enrichment analysis of DEGs was performed using clusterProfiler (v.4.12.6). Developmental trajectories and pseudotime relationships were inferred using Slingshot (v.2.12.0). Motif deviation (accessibility) analysis was conducted using chromVAR (v.1.26.0) to quantify variability in transcription factor motif accessibility across single cells. Differential transcription factor motif enrichment was calculated using FindAllMarkers or FindMarkers for pairwise comparisons with logistic regression testing, min. pct = 0.05, log$_2$ fold change threshold = 2 and adjusted $P$ < 0.05, with total counts included as a latent variable. All heatmaps were generated using ComplexHeatmap (v.2.20.0).

## CUT&Tag-seq

Naive CD8$^+$ T cells were isolated from spleens of C57BL/6 WT mice and activated with 5 µg ml$^{-1}$ plate-bound aCD3 Ab and 2 µg ml$^{-1}$ soluble aCD28 Ab for 24 h in 24-well plates. A total of $1 \times 10^6$ cells were used per target (ZFP148 or IgG control) for library construction using CUT&Tag (Epicypher, cat. no. 14-1102-48s1). Libraries were pooled at equimolar ratios and sequenced on an Illumina NovaSeq X Plus platform (Azenta Life Sciences), generating 5–10 million reads per sample.

## CUT&Tag-seq analysis

CUT&Tag sequencing data were processed using the nf-core/cutandrun pipeline (v.3.0.0) (https://nf-co.re/cutandrun/3.2.2/)—a community-curated Nextflow pipeline. Raw sequencing reads were first subjected to adapter trimming using fastp (v.0.23.2), followed by alignment to the mouse reference genome (mm10) using Bowtie2 (v.2.4.4). Aligned reads were filtered to remove low-quality mappings, PCR duplicates (using Picard MarkDuplicates v.2.27.5) and mitochondrial reads. Peaks were called using SEACR (v.1.3) in 'relaxed' mode with appropriate IgG or input control normalization. Genome-wide signal tracks were generated using deepTools (v.3.5.1) and IGV (v.2.18.4) for visualization. Quality control metrics, including fragment size distribution, duplication rates and library complexity, were assessed and summarized using MultiQC (v.1.13). All steps were run with default settings unless otherwise specified.

## Secondary analysis of scRNA-seq datasets of human cancer patients

**Pan-cancer scRNA-seq data assembly.** An extensive compendium of single-cell transcriptomes was constructed by aggregating profiles from 346 tumor specimens, representing 251 patients, across 20 publicly available scRNA-seq datasets[52–71]. To reduce platform-specific biases and ensure consistency, only datasets generated with the 10x Genomics droplet-based system was included.

**Quality assessment and preprocessing.** Raw single-cell data were filtered stringently using Scanpy (v.1.9.5). Cells were retained only if they expressed at least 200 genes and exhibited mitochondrial gene fractions below 20% of total counts. Additional filters eliminated barcodes suggestive of debris (fewer than 400 genes or 500 unique molecular identifiers) and excluded potential doublets (cells with more than 5,500 genes or 30,000 unique molecular identifiers). After these quality control steps, the raw count matrices and corresponding AnnData objects were merged. Data normalization to transcripts per million was performed using sc.pp.normalize_total, followed by a logarithmic transformation with sc.pp.log1p. Only tumor-derived cells were retained, resulting in a final dataset comprising 1,030,968 high-quality cells and 14,090 genes for further analyses.

**Batch correction and integration.** To reconcile differences between studies while preserving genuine biological variation, batch effects were mitigated using the scVI Python package (scvi-tools v.1.0.4). By incorporating sample identity as a covariate, the scVI model effectively reduced technical variability across samples. The performance of the batch correction was evaluated by measuring the reduction of study-specific biases while retaining critical biological signals. Subsequent analyses—including clustering, differential expression and trajectory inference—were conducted on the integrated dataset. Cellular heterogeneity was visualized using UMAP, which delineated variations across batches, datasets, sex, tissue origin and cancer type.

**Cell type identification.** Cell populations were classified using the scANVI algorithm (scvi-tools v.1.0.4), which leverages pre-annotated reference data for main immune cell lineages such as epithelial, endothelial, fibroblast, lymphoid, myeloid and plasma cells as well as subsets of CD8$^+$ TILs. Initial clustering within the scANVI latent space was refined with Leiden clustering to assign discrete cell identities. The model was trained for 20 epochs with cell-type labels transferred using 100 samples per label. For a more detailed resolution of T cell subpopulations, corresponding AnnData objects were further integrated and subjected to scVI-based batch correction.

**Functional signature score analysis.** Functional states across individual cells were quantified by computing gene signature scores using the scanpy.tl.score_genes function (Scanpy v.1.9.5). This approach enabled the assessment of cellular functional signatures within the dataset.

**Validation and prognostic evaluation of the ZFP148 KO gene signature in CD8⁺ TILs.** The clinical relevance of the ZFP148 KO gene signature in CD8⁺ T cells was examined using processed scRNA-seq data from 116 liver cancer samples obtained from 94 male patients[34]. Analysis was confined to primary tumors and metastatic lesions. After applying the same rigorous quality control, batch correction, and cell-type annotation pipeline, CD8⁺ T cells were isolated and ZFP148 KO signature score was computed using scanpy.tl.score_genes.

**Survival analysis using expression of ZFP148 KO gene signature in CD8⁺ TILs.** To determine the prognostic impact of ZFP148 KO gene signature expression in CD8⁺ TILs, Kaplan–Meier survival curves were generated and differences assessed via the log-rank test and univariate Cox proportional hazards (Cox PH) models. Two additional multivariable Cox PH models were also employed to adjust for potential confounders, with hazard ratios and 95% confidence intervals reported accordingly. The optimal cutoff for stratifying ZFP148 KO gene signature expression levels was established using the surv_cutpoint function from the survminer R package, which applies maximally selected rank statistics from the maxstat[72] package. Continuous covariates in the Cox PH models were examined for linearity to validate the model assumptions.

**Expression of *ZNF148* mRNA, ZFP148 KO gene signature, *KLF2* mRNA and KLF2 gene signature in immunotherapy-treated cohorts.** Expression of *ZNF148* mRNA, ZFP148 KO gene signature, *KLF2* mRNA and KLF2 gene signature was investigated further in independent scRNA-seq datasets derived from patients undergoing various immunotherapeutic regimens. These cohorts included patients receiving anti-CD19 CAR-T cells for DLBCL[38] and aPD-1 + aCTLA-4 Abs for ccRCC[37]. For each dataset, the identical preprocessing pipeline—comprising quality filtering, batch correction and cell-type annotation—was applied.

### OS analysis of human cancer patients using published datasets

OS analyses of human cancer patients were performed using webservers that have access to published datasets. Treatment-naive gastric cancer, colon cancer, ovarian cancer, pancreatic cancer, liver hepatocellular carcinoma, pancreatic ductal adenocarcinoma, sarcoma, thyroid carcinoma and uterine corpus endometrial carcinoma patients were stratified into *ZNF148*ʰⁱ and *ZNF148*ˡᵒ groups using the expression-based 'best cutoff' option of the Kaplan–Meier Plotter webserver[33], and OS was visualized using Kaplan–Meier curves. Melanoma patients treated with aPD-1 or aCTLA-4 Ab were stratified into high- and low- expression groups based on pretreatment *ZNF148* or *KLF2* mRNA levels using the expression-based 'best cutoff' option of the Kaplan–Meier Plotter webserver[35] or expression of the ZFP148 KO gene signature or KLF2 gene signature[30] using the Tumor Immune Dysfunction and Exclusion (TIDE) webserver[36]; OS was visualized by Kaplan–Meier analysis.

### Statistics and reproducibility

Statistical analyses for flow cytometry, tumor growth curves and mouse survival were performed using GraphPad Prism (v.10). Unpaired or paired two-sided *t*-tests were used for comparisons between two unpaired or paired groups, respectively. One-way analysis of variance (ANOVA) followed by Tukey's multiple comparisons test was used for comparisons among three or more groups. One-way ANOVA followed by Holm–Šídák's multiple comparisons test was used for comparisons between preselected pairs among three or more groups. Two-way ANOVA was used to compare time-course curves, with Bonferroni correction for multiple comparisons. The log-rank test was used to compare OS of mice across several treatment groups, with Bonferroni correction for multiple comparisons.

Analyses of mouse scRNA-seq and scATAC-seq data were performed using R (v.4.4.0) with the packages Seurat (v.5.1.0), Signac (v.1.14.0), AUCell (v.1.26.0), slingshot (v.2.12.0), chromVAR (v.1.26.0), clusterProfiler (v.4.12.6), ComplexHeatmap (v.2.20.0) and EnhancedVolcano (v.1.22.0). Differentially expressed genes (DEGs) were identified using the FindMarkers() or FindAllMarkers() functions in Seurat, with statistical significance assessed by a two-sided Wilcoxon rank-sum test, with Benjamini–Hochberg correction for multiple comparisons. DACRs and differential transcription factor motif accessibility were identified using FindMarkers() or FindAllMarkers() with statistical significance assessed by a two-sided logistic regression likelihood-ratio test, with Benjamini–Hochberg correction for multiple comparisons. GO enrichment analysis was performed using a one-sided hypergeometric test, with Benjamini–Hochberg correction for multiple comparisons. A two-sided Wilcoxon rank-sum test was used to compare mRNA expression, promoter chromatin accessibility, motif accessibility and gene signature scores between *Zfp148*ᶠˡ/ᶠˡ and ZFP148 cKO CD8⁺ T cells. Analyses of the integrated human scRNA-seq dataset were performed using Python (v.3.10.9) packages Scanpy (v.1.9.5), Pandas (v.2.0.0), Statsmodels (v.0.14.0), NumPy (v.1.24.2), SciPy (v.1.10.1), Matplotlib (v.3.8.0), Seaborn (v.0.11.2) and scikit-learn (v.1.3.2), as well as R (v.4.3.1) packages Circlize (v.0.4.16), GseaVis (v.0.0.5), Enrichplot (v.1.22.0), GridExtra (v.2.3.0), pheatmap (v.1.0.12) and DEGreport (v.1.38.5). A two-sided Wilcoxon rank-sum test was used for comparisons between two groups. OS between two groups of patients was compared using the log-rank test. A *P* value ≤ 0.05 (or adjusted *P* value ≤ 0.05 after multiple-testing correction) was considered statistically significant.

No statistical methods were used to predetermine sample size, but sample sizes were similar to those reported in previous publications[24,73]. Data distribution was assumed to be normal but was not formally tested. Age- and sex-matched animals were assigned randomly to experimental conditions. Investigators were not blinded to group allocation during data collection and analysis. No data were excluded. All data were generated from at least two independent experiments with a minimum of three biological replicates, yielding consistent phenotypes to ensure reproducibility, with the exceptions of the paired scRNA-seq and scATAC-seq using CD44ʰⁱGP₃₃₋₄₁ Tet⁺CD8⁺ T cells in spleens of *Zfp148*ᶠˡ/ᶠˡ and ZFP148 cKO mice at day 21 post-LCMV Cl13 infection and the CUT&Tag-seq using activated splenic CD8⁺ T cells from C57BL/6 WT mice. To minimize intermouse variability and enhance reproducibility, cells used for the single-cell multiome experiment were pooled from 10 mice for *Zfp148*ᶠˡ/ᶠˡ and 12 mice for ZFP148 cKO; cells used for the CUT&Tag-seq experiment were pooled from 5 mice.

### Reporting summary

Further information on research design is available in the Nature Portfolio Reporting Summary linked to this article.

### Data availability

Paired scRNA-seq and scATAC-seq and CUT&Tag-seq data are available in the NCBI database under accession numbers GSE297040 and GSE296311, respectively. Further information and requests for data should be directed to the corresponding author, Z. Li. Source data are provided with this paper.

### Code availability

Scripts generated for analysis in this manuscript have been deposited at https://github.com/xt271061109/Xiao2026/tree/main.

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

## Acknowledgements

We thank Y. Koguchi, K. Reynolds, K. Das, R. Davenport and A. Menon from the Immune Monitoring and Discovery Platform (IMDP) of the Pelotonia Institute for Immuno-Oncology (PIIO) for technical support. We thank D. Bucci and V. Balatti for administrative assistance. We are grateful to all members of the Z. Li laboratory and the PIIO for helpful discussions throughout the study. J. Shen from the W. Cui laboratory at Northwestern University contributed to the development of the CUT&Tag-seq methodology. C. Wang from the Center for AI & Bioinformatics in Immuno-Oncology at the PIIO provided guidance on CUT&Tag-seq data analysis. This work was supported by the National Cancer Institute (NCI) grant P01 CA278732 (Z.L.), the National Institutes of Health (NIH) grant R01 CA255334 (Z.L.), The Ohio State University College of Medicine Research Innovation Career Development Award (D.S.), the Naren Patel Genitourinary Research Fund (D.S.), the Bladder Cancer Immuno-Oncology Research Fund (D.S.), NIH grant R01 CA269984 (G.X.), Susan G. Komen grant CCR231013713 (G.X.) and the American Cancer Society grant RSG-23-1036499-01 (G.X.). T.X. was supported by the Pelotonia Graduate Fellowship Program through The Ohio State University Comprehensive Cancer Center (OSUCCC). This work was further enabled by the resources, expertise, and infrastructure provided by the PIIO, supported by the Pelotonia community and the OSUCCC.

## Author contributions

Z.L. and T.X. conceived the project. T.X. and Z.L. designed experiments and wrote the manuscript. T.X. performed most experiments described herein and related analyses. X.C. performed analyses of the integrated human cancer scRNA-seq dataset. N.-J.S. contributed to in vivo tumor experiments. R.J.B. performed cotransferring of CRISPR-edited P14 CD8[+] T cells followed by LCMV Cl13 infection. A.M. contributed to the analysis of mouse single-cell multiomic data. J.K.M. performed data analysis of publicly available human bulk RNA-seq datasets. A.Y., Y.W., M.Q.M.L., J.L., F.G., B.L., H.-Y.C., F.-Y.L. and H.E.G. contributed to the LCMV infection experiment. D.S. provided access to specimens from bladder cancer patients. M.V. contributed to manuscript preparation. P.W. contributed to management of all experimental mouse strains. J.L.M. provided the *Zfp148*^fl/fl mice. M.P.R., K.O., C.-W.J.L., X.L., D.T., G.X., Q.M., W.C. and Z.L. provided intellectual input and critical edits to the manuscript. Z.L. supervised the entire project. All authors reviewed and approved the manuscript.

## Competing interests

Z.L. serves as a member of scientific advisory boards for HanchorBio. The other authors declare no competing interests.

## Additional information

**Extended data** is available for this paper at https://doi.org/10.1038/s41590-026-02461-2.

**Correspondence and requests for materials** should be addressed to Zihai Li.

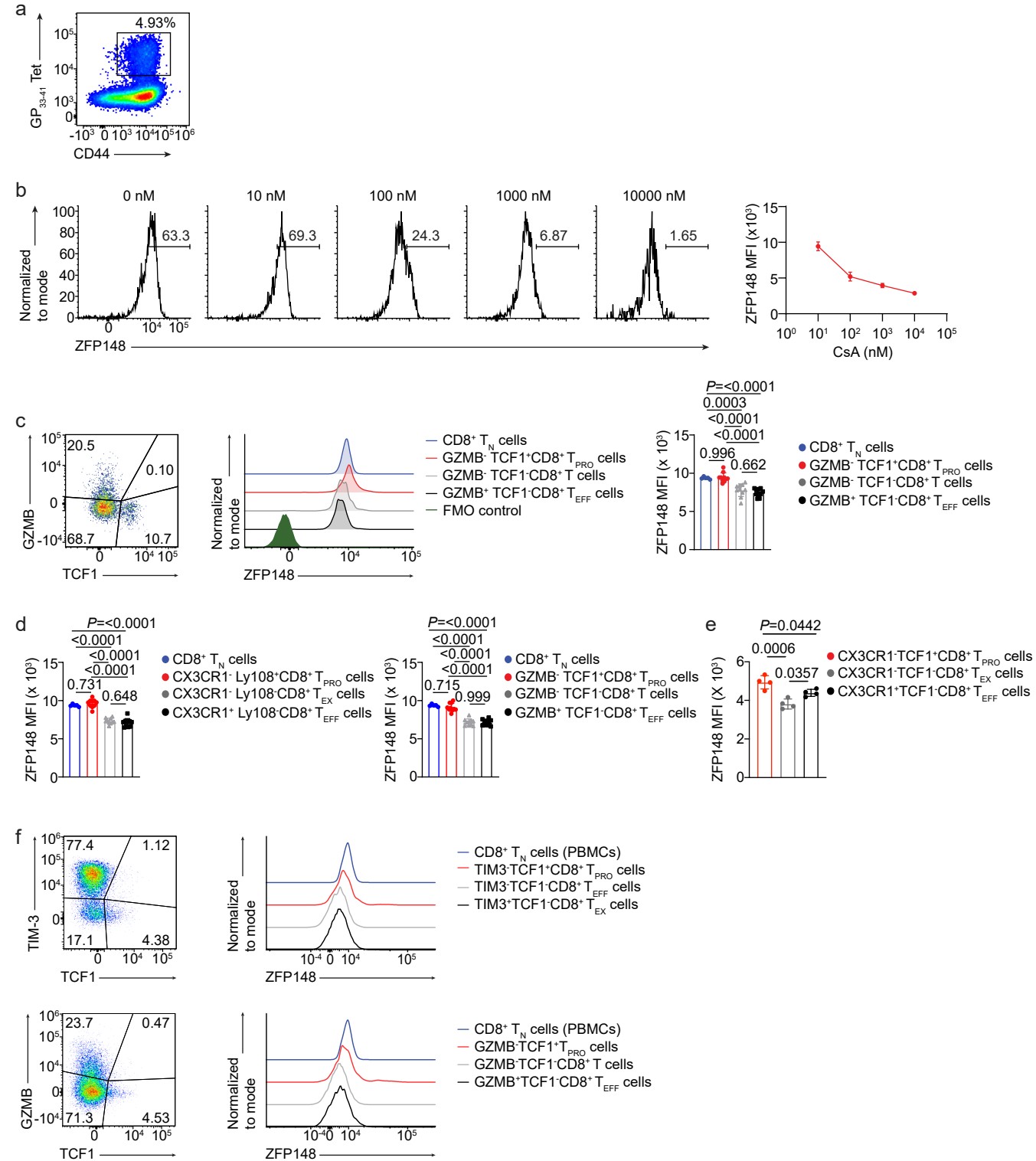

**Extended Data Fig. 1 | See next page for caption.**

**Extended Data Fig. 1 | ZFP148 expression associates with CD8$^+$ T$_{PRO}$ cell states in both chronic LCMV infection and cancer. a**, Gating of splenic CD44$^{hi}$GP$_{33-41}$ Tet$^+$CD8$^+$ T cells in C57BL/6 WT mice at day 22 post-LCMV Cl13 infection. **b**, Density plots showing ZFP148 protein expression (left) and MFI of ZFP148 (right) in splenic CD8$^+$ T$_N$ cells in C57BL/6 WT mice stimulated with 5 µg ml$^{-1}$ aCD3 Ab and 2 µg ml$^{-1}$ aCD28 Ab along with 0, 10, 100, 1000 and 10000 nM CsA for 48 h (n = 4). **c**, Scatter plots showing GZMB versus TCF1 protein expression in splenic CD44$^{hi}$GP$_{33-41}$ Tet$^+$CD8$^+$ T cells (left) and density plots showing expression of ZFP148 protein (middle) and MFI of ZFP148 (right) in GZMB$^-$TCF1$^+$CD8$^+$ T$_{PRO}$ cells, GZMB$^-$TCF1$^-$CD8$^+$ T cells and GZMB$^+$TCF1$^-$CD8$^+$ T$_{EFF}$ cells in C57BL/6 WT mice at day 22 post-LCMV Cl13 infection (n = 10) and splenic CD8$^+$ T$_N$ cells in non-infected C57BL/6 WT mice as a control (n = 5). **d**, MFI of ZFP148 in CX3CR1$^-$Ly108$^+$CD8$^+$ T$_{PRO}$ cells, CX3CR1$^-$Ly108$^-$CD8$^+$ T cells and CX3CR1$^+$Ly108$^-$CD8$^+$ T$_{EFF}$ cells (left) or GZMB$^-$TCF1$^+$CD8$^+$ T$_{PRO}$ cells, GZMB$^-$TCF1$^-$CD8$^+$ T cells and GZMB$^+$TCF1$^-$CD8$^+$ T$_{EFF}$

cells (right) of splenic CD44$^{hi}$GP$_{276-286}$ Tet$^+$CD8$^+$ T cells in LCMV-infected mice as in **c** (n = 10) and splenic CD8$^+$ T$_N$ cells in non-infected C57BL/6 WT mice as a control (n = 5). **e**, MFI of ZFP148 in CX3CR1$^-$TCF1$^+$CD8$^+$ T$_{PRO}$ cells, CX3CR1$^-$TCF1$^-$CD8$^+$ T$_{EX}$ cells and CX3CR1$^+$ TCF1$^-$CD8$^+$ T$_{EFF}$ cells in CD8$^+$ TILs in C57BL/6 WT mice at day 14 post-subcutaneous injection of MC38 tumors (n = 4). **f**, Scatter plots showing TIM-3 versus TCF1 protein expression or GZMB versus TCF1 protein expression (left) and density plots showing expression of ZFP148 protein in TIM-3$^-$TCF1$^+$CD8$^+$ T$_{PRO}$ cells, TIM-3$^-$TCF1$^-$CD8$^+$ T$_{EFF}$ cells and TIM-3$^+$TCF1$^-$CD8$^+$ T$_{EX}$ cells or GZMB$^-$TCF1$^+$CD8$^+$ T$_{PRO}$ cells, GZMB$^-$TCF1$^-$CD8$^+$ T cells and GZMB$^+$TCF1$^-$CD8$^+$ T$_{EFF}$ cells (right) in CD8$^+$ TILs in human muscle-invasive bladder tumors and CD45RA$^+$CD45RO$^-$CD8$^+$ T$_N$ cells in PBMCs of the same patient as a control. Data are representative of 2-3 independent experiments (**a-e**); representative of two patients (**f**). Data are presented as mean ± s.d. Statistical analysis was performed using one-way ANOVA followed by Tukey's multiple comparisons test (**c-e**).

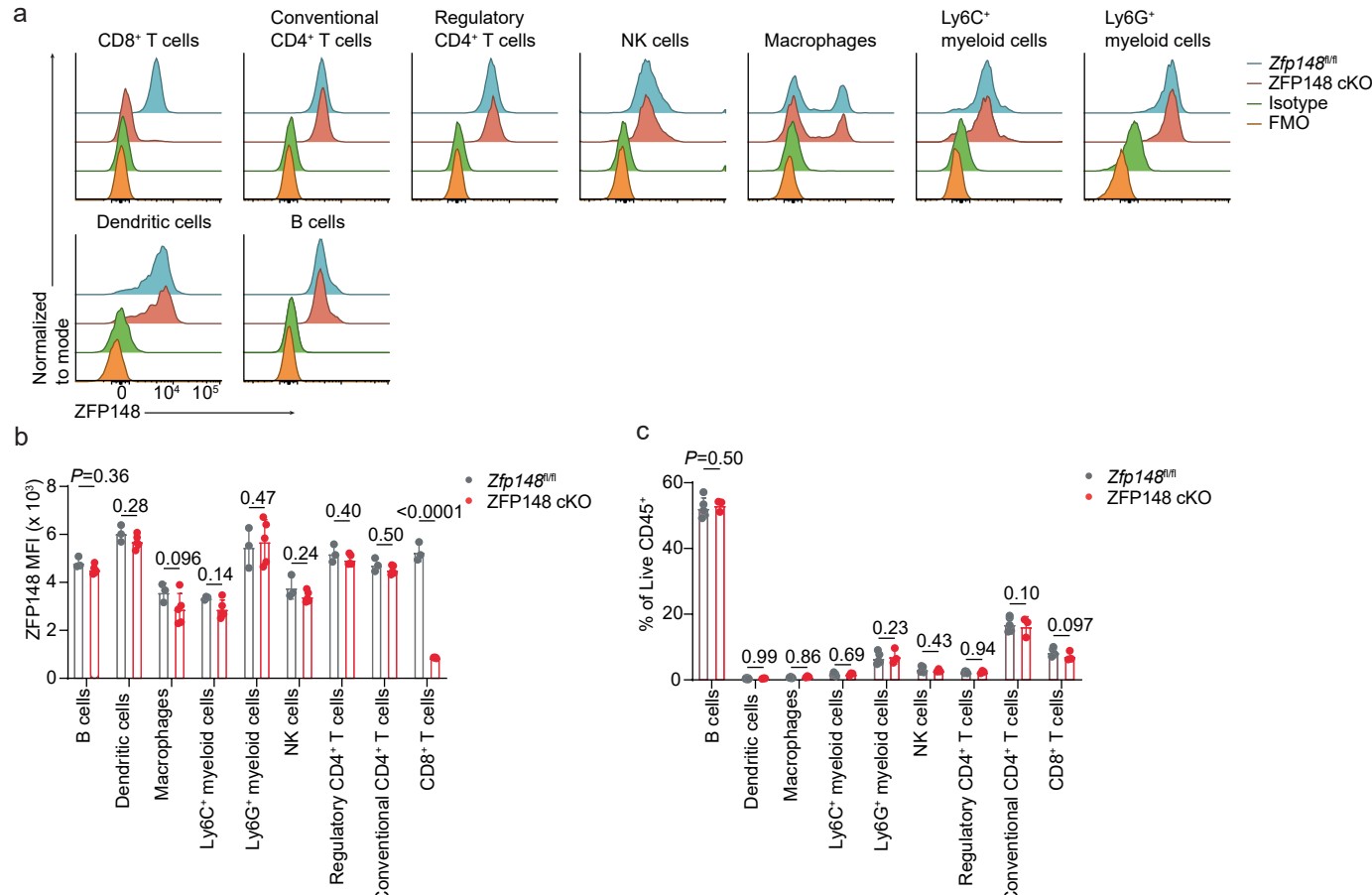

**Extended Data Fig. 2 | Baseline immune phenotyping of CD8⁺ T cell-specific ZFP148 KO mice. a**, Density plots showing expression of ZFP148 protein in CD8⁺ T cells, conventional CD4⁺ T cells, regulatory CD4⁺ T cells, NK cells, macrophages, Ly6C⁺ myeloid cells, Ly6G⁺ myeloid cells, dendritic cells and B cells in spleens of non-infected *Zfp148*ᶠˡ/ᶠˡ and ZFP148 cKO mice. A PE-conjugated isotype antibody-stained control and a fluorescence minus one (FMO) control for the antibody staining ZFP148 were included for every cell type described above.

**b**, MFI of ZFP148 in splenic immune cell lineages in non-infected *Zfp148*ᶠˡ/ᶠˡ (n = 3) and ZFP148 cKO mice (n = 5) as in **a. c**, Frequency of splenic immune cell lineages in spleens of non-infected *Zfp148*ᶠˡ/ᶠˡ (n = 3) and ZFP148 cKO mice (n = 5) as in **a**. Data are representative of 2 independent experiments (**a-c**). Data are presented as mean ± s.d. Statistical analysis was performed using multiple unpaired two-sided t-tests (**b,c**).

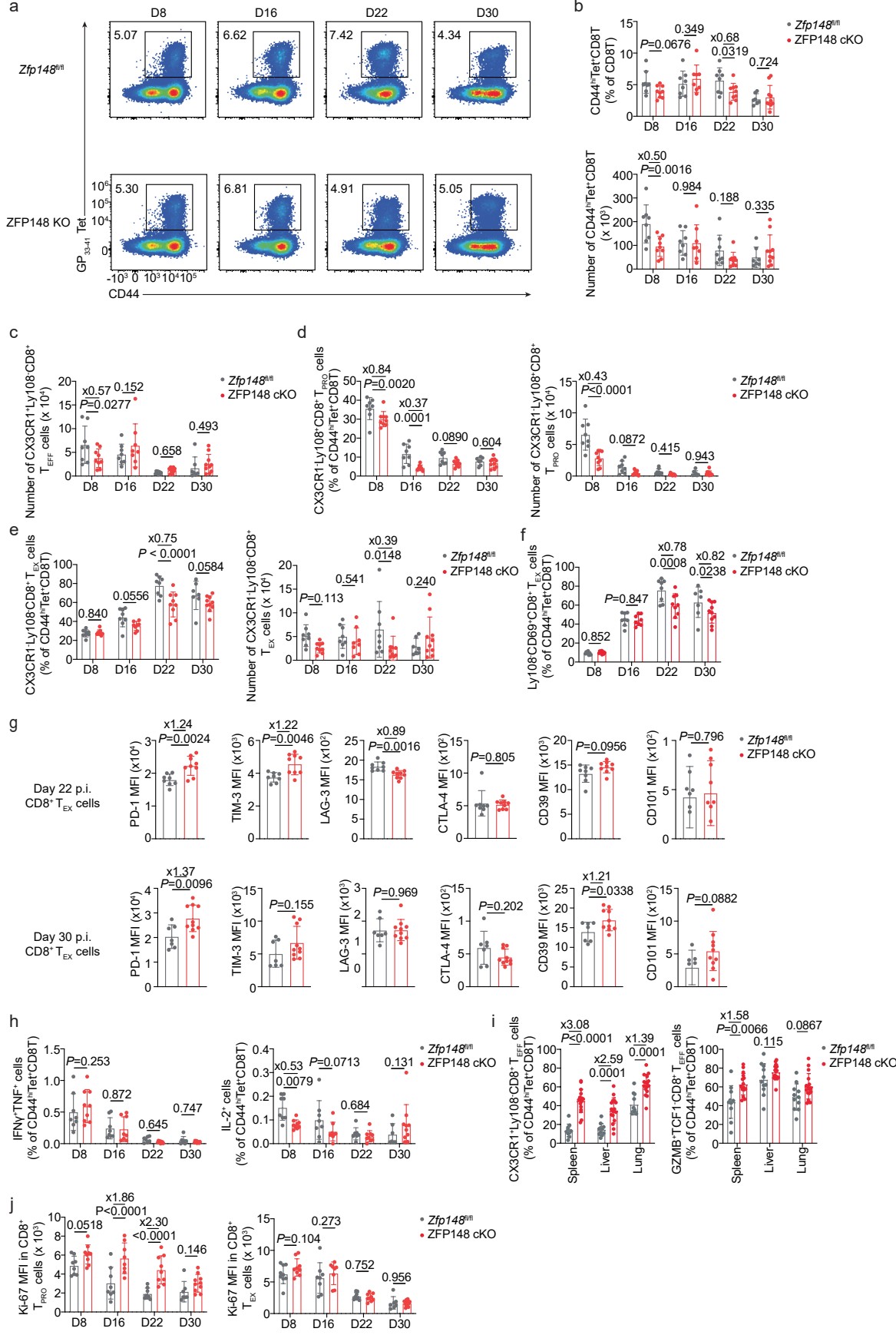

**Extended Data Fig. 3 | See next page for caption.**

**Extended Data Fig. 3 | ZFP148 restrains effector differentiation of antigen-specific CD8⁺ T cells during chronic LCMV infection. a**, Scatter plots showing GP$_{33-41}$ tetramer staining versus CD44 protein expression in splenic CD8⁺ T cells in *Zfp148*$^{fl/fl}$ or ZFP148 cKO mice at days 8, 16, 22 and 30 post-LCMV Cl13 infection. **b**, Frequency (top) and number (bottom) of splenic CD44$^{hi}$GP$_{33-41}$ Tet⁺CD8⁺ T cells in *Zfp148*$^{fl/fl}$ or ZFP148 cKO mice at days 8 (*Zfp148*$^{fl/fl}$, n = 8; ZFP148 cKO, n = 9), 16 (*Zfp148*$^{fl/fl}$, n = 8; ZFP148 cKO, n = 8), 22 (*Zfp148*$^{fl/fl}$, n = 8; ZFP148 cKO, n = 9) and 30 (*Zfp148*$^{fl/fl}$, n = 7; ZFP148 cKO, n = 10) post-LCMV Cl13 infection. **c**, Number of CX3CR1⁺Ly108⁻CD8⁺ T$_{EFF}$ cells in splenic CD44$^{hi}$GP$_{33-41}$ Tet⁺CD8⁺ T cells in *Zfp148*$^{fl/fl}$ or ZFP148 cKO mice as in **b. d**, Frequency (left) and number (right) of CX3CR1⁻Ly108−CD8⁺ T$_{PRO}$ cells in splenic CD44$^{hi}$GP$_{33-41}$ Tet⁺CD8⁺ T cells in *Zfp148*$^{fl/fl}$ or ZFP148 cKO mice as in **b. e**, Frequency (left) and number (right) of CX3CR1⁻Ly108⁻CD8⁺ T$_{EX}$ cells in splenic CD44$^{hi}$GP$_{33-41}$ Tet⁺CD8⁺ T cells in *Zfp148*$^{fl/fl}$ or ZFP148 cKO mice as in **b. f**, Frequency of Ly108⁻CD69⁺CD8⁺ T$_{EX}$ cells in splenic CD44$^{hi}$GP$_{33-41}$ Tet⁺CD8⁺ T cells in *Zfp148*$^{fl/fl}$ or ZFP148 cKO mice as in **b. g**, MFI of PD-1, TIM-3, LAG-3, CTLA-4, CD39 and CD101 in splenic CX3CR1⁻Ly108⁻CD8⁺ T$_{EX}$ cells in *Zfp148*$^{fl/fl}$ or ZFP148 cKO mice at day 22 (*Zfp148*$^{fl/fl}$, n = 8; ZFP148 cKO, n = 9) and 30 (*Zfp148*$^{fl/fl}$, n = 7; ZFP148 cKO, n = 10) post-LCMV Cl13 infection. **h**, Frequency of IFNγ⁺TNF⁺ (left) or IL-2⁺ subsets (right) in splenic CD8⁺ T cells in *Zfp148*$^{fl/fl}$ or ZFP148 cKO mice as in **b**, restimulated ex vivo. **i**, Frequency of CX3CR1⁺Ly108⁻CD8⁺ T$_{EFF}$ cells (left) or GZMB⁺TCF1⁻CD8⁺ T$_{EFF}$ cells (right) in CD44$^{hi}$GP$_{33-41}$ Tet⁺CD8⁺ T cells in spleens, livers and lungs of *Zfp148*$^{fl/fl}$ (n = 11) or ZFP148 cKO mice (n = 16) at day 22 post-LCMV Cl13 infection. **j**, MFI of Ki-67 in splenic CX3CR1⁺Ly108⁻CD8⁺ T$_{PRO}$ cells (left) or CX3CR1⁻Ly108⁻CD8⁺ T$_{EX}$ cells (right) in *Zfp148*$^{fl/fl}$ or ZFP148 cKO mice as in **b**. Fold changes in **b-j** were calculated as mean (ZFP148 cKO) divided by mean (*Zfp148*$^{fl/fl}$). Data are representative of at least 3 independent experiments. Data are presented as mean ± s.d. Statistical analysis was performed using multiple unpaired two-sided t-tests (**b-f, h-j**); an unpaired two-sided t-test (**g**).

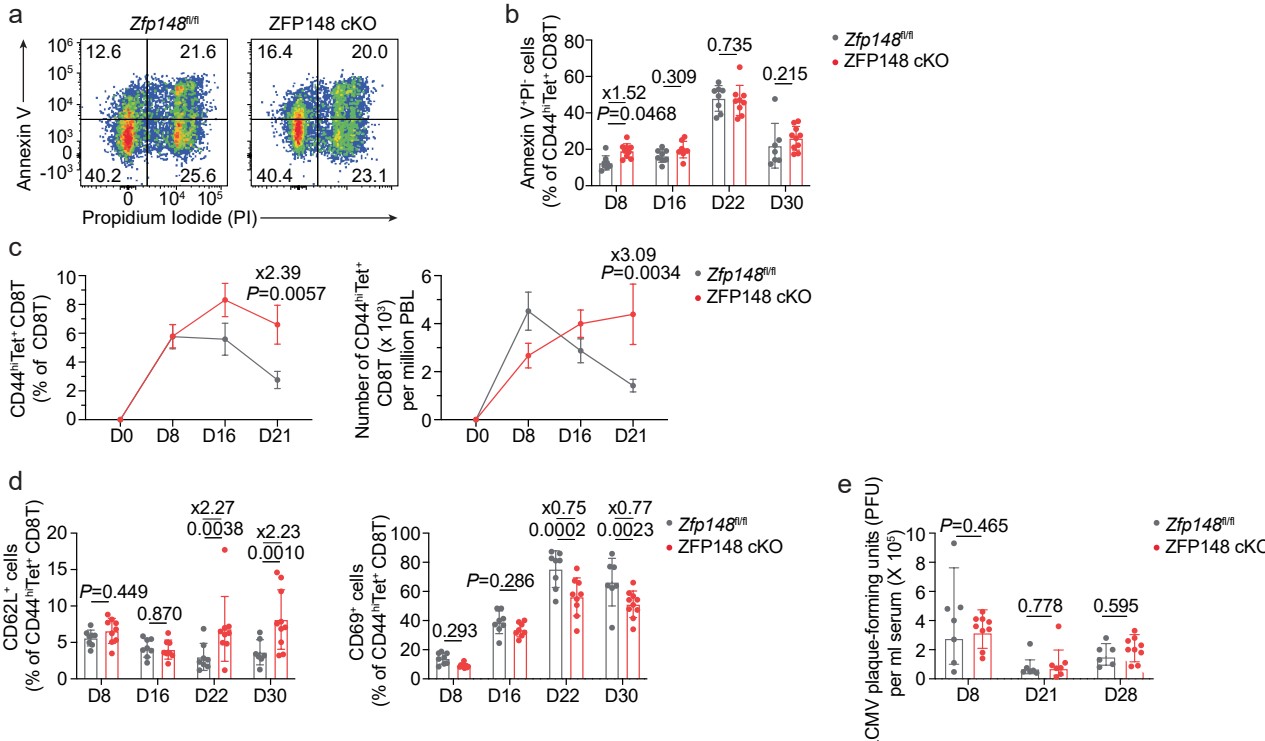

Extended Data Fig. 4 | ZFP148 loss promotes circulation but not apoptosis of antigen-specific CD8+ T cells. a,b, Scatter plots showing Annexin V versus propidium iodide (PI) staining in splenic CD44hiGP33-41 Tet+CD8+ T cells at day 8 post-LCMV Cl13 infection (*Zfp148*fl/fl, n = 8; ZFP148 cKO, n = 9) (a) and frequency of apoptotic Annexin V+PI− subset (b) in splenic CD44hiGP33-41 Tet+CD8+ T cells at days 8 (*Zfp148*fl/fl, n = 8; ZFP148 cKO, n = 9), 16 (*Zfp148*fl/fl, n = 8; ZFP148 cKO, n = 8), 22 (*Zfp148*fl/fl, n = 8; ZFP148 cKO, n = 9) and 30 (*Zfp148*fl/fl, n = 7; ZFP148 cKO, n = 10) post-LCMV Cl13 infection. c, Frequency (left) and number (right) of CD44hiGP33-41 Tet+ CD8+ T cells in peripheral blood (PBL) of *Zfp148*fl/fl or ZFP148 cKO mice at day 8 (*Zfp148*fl/fl, n = 8; ZFP148 cKO, n = 10), 16 (*Zfp148*fl/fl, n = 8; ZFP148 cKO, n = 10) and 21 (*Zfp148*fl/fl, n = 7; ZFP148 cKO, n = 10) post-LCMV Cl13 infection. d, Frequency of CD62+ subset (left) and CD69+ subset (right) in *Zfp148*fl/fl or ZFP148 cKO splenic CD44hiGP33-41 Tet+CD8+ T cells as in b. e, Serum LCMV Cl13 titers in *Zfp148*fl/fl or ZFP148 cKO mice at day 8 (*Zfp148*fl/fl, n = 7; ZFP148 cKO, n = 9), 16 (*Zfp148*fl/fl, n = 6; ZFP148 cKO, n = 7) and 21 (*Zfp148*fl/fl, n = 6; ZFP148 cKO, n = 9) post-LCMV Cl13 infection. Fold changes in b-d were calculated as mean (ZFP148 cKO) divided by mean (*Zfp148*fl/fl). Data are representative of 2-3 independent experiments (a-e). Data are presented as mean ± s.d. Statistical analysis was performed using multiple unpaired two-sided t-tests (b-e).

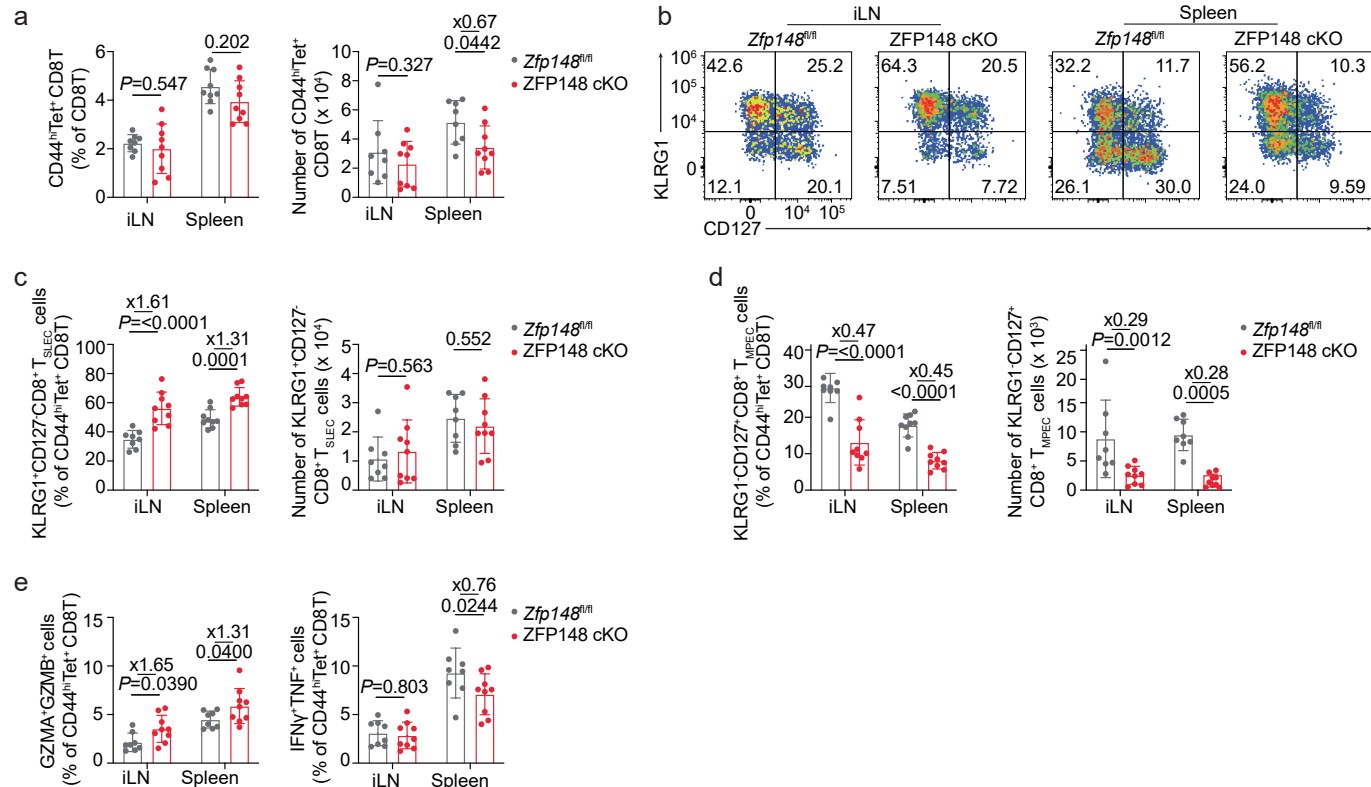

**Extended Data Fig. 5 | ZFP148 restrains KLRG1$^+$CD127$^-$CD8$^+$ T$_{SLEC}$ cell formation during LCMV Armstrong infection. a**, Frequency (left) and number (right) of CD44$^{hi}$GP$_{33-41}$ Tet$^+$CD8$^+$ T cells in inguinal lymph nodes (iLNs) or spleens of *Zfp148*$^{fl/fl}$ (n = 8) or ZFP148 cKO mice (n = 9) at day 9 post-LCMV Armstrong infection. **b**, Scatter plots showing KLRG1 versus CD127 protein expression in CD44$^{hi}$GP$_{33-41}$ Tet$^+$CD8$^+$ T cells in iLNs or spleens of *Zfp148*$^{fl/fl}$ or ZFP148 cKO mice as in **a. c,d**, Frequency (left) and number (right) of KLRG1$^+$CD127$^-$CD8$^+$ T$_{SLEC}$ cells (**c**) or KLRG1$^-$CD127$^-$CD8$^+$ T$_{MPEC}$ cells (**d**) in CD44$^{hi}$GP$_{33-41}$ Tet$^+$CD8$^+$ T cells in iLNs or spleens of *Zfp148*$^{fl/fl}$ or ZFP148 cKO mice as in **a. e**, Frequency of GZMA$^+$GZMB$^+$ subset (left) and IFNγ$^+$TNF$^+$ subset (right) in CD8$^+$ T cells in iLNs or spleens of *Zfp148*$^{fl/fl}$ or ZFP148 cKO mice as in **a**, restimulated ex vivo. Fold changes in **a,c-e** were calculated as mean (ZFP148 cKO) divided by mean (*Zfp148*$^{fl/fl}$). Data are representative of 2 independent experiments (**a–e**). Data are presented as mean ± s.d. Statistical analysis was performed using multiple unpaired two-sided t-tests (**a,c-e**).

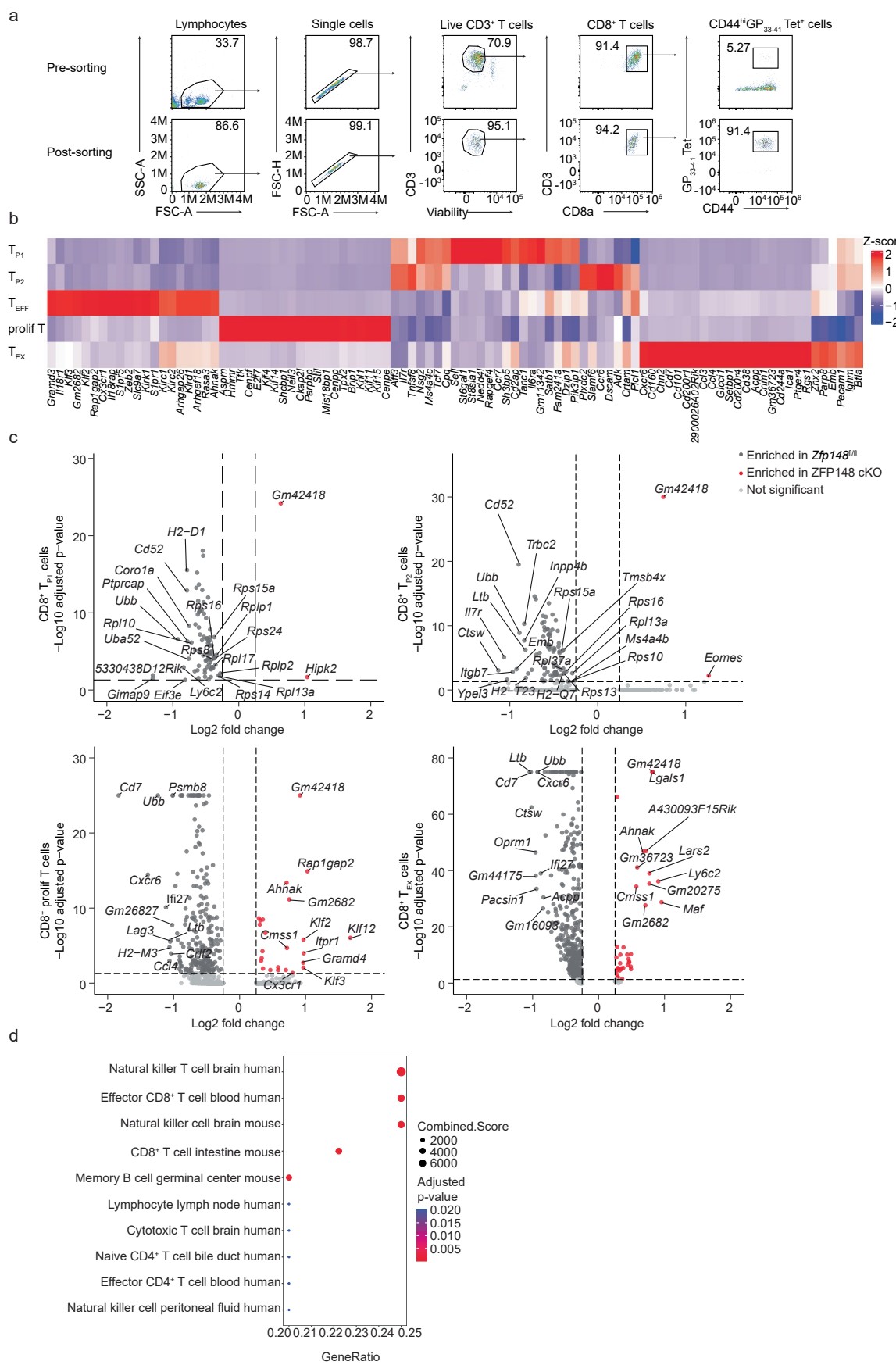

**Extended Data Fig. 6 | See next page for caption.**

**Extended Data Fig. 6 | ZFP148 deficiency transcriptionally skews antigen-specific CD8⁺ T cell differentiation towards CD8⁺ T_EFF cells. a**, Gating and purity of sorted splenic CD44^hiGP_{33-41} Tet⁺CD8⁺ T cells in *Zfp148*^fl/fl or ZFP148 cKO mice at day 21 post-LCMV Cl13 infection. **b**, Heatmap showing subset-specific DEGs among CD8⁺ T_{P1}, CD8⁺ T_{P2}, CD8⁺ T_EFF, CD8⁺ prolif T and CD8⁺ T_EX subsets of combined *Zfp148*^fl/fl and ZFP148 cKO splenic CD44^hiGP_{33-41} Tet⁺CD8⁺ T cells as in **a**. **c**, Volcano plot showing DEGs in CD8⁺ T_{P1}, CD8⁺ T_{P2}, CD8⁺ prolif T or CD8⁺ T_EX

cells comparing between *Zfp148*^fl/fl and ZFP148 cKO mice as in **a**. **d**, Gene Ontology enrichment analysis of DEGs between *Zfp148*^fl/fl and ZFP148 cKO CD8⁺ T_EFF cells as in **a** by using the Enrichr webserver with the 'CellMarker 2024' database. Data are pooled from 10 mice for *Zfp148*^fl/fl and 12 mice for ZFP148 cKO. Statistical analysis was performed using a two-sided Wilcoxon rank-sum test with Benjamini–Hochberg correction for multiple comparisons (**c**); a one-sided hypergeometric test with Benjamini–Hochberg correction for multiple comparisons (**d**).

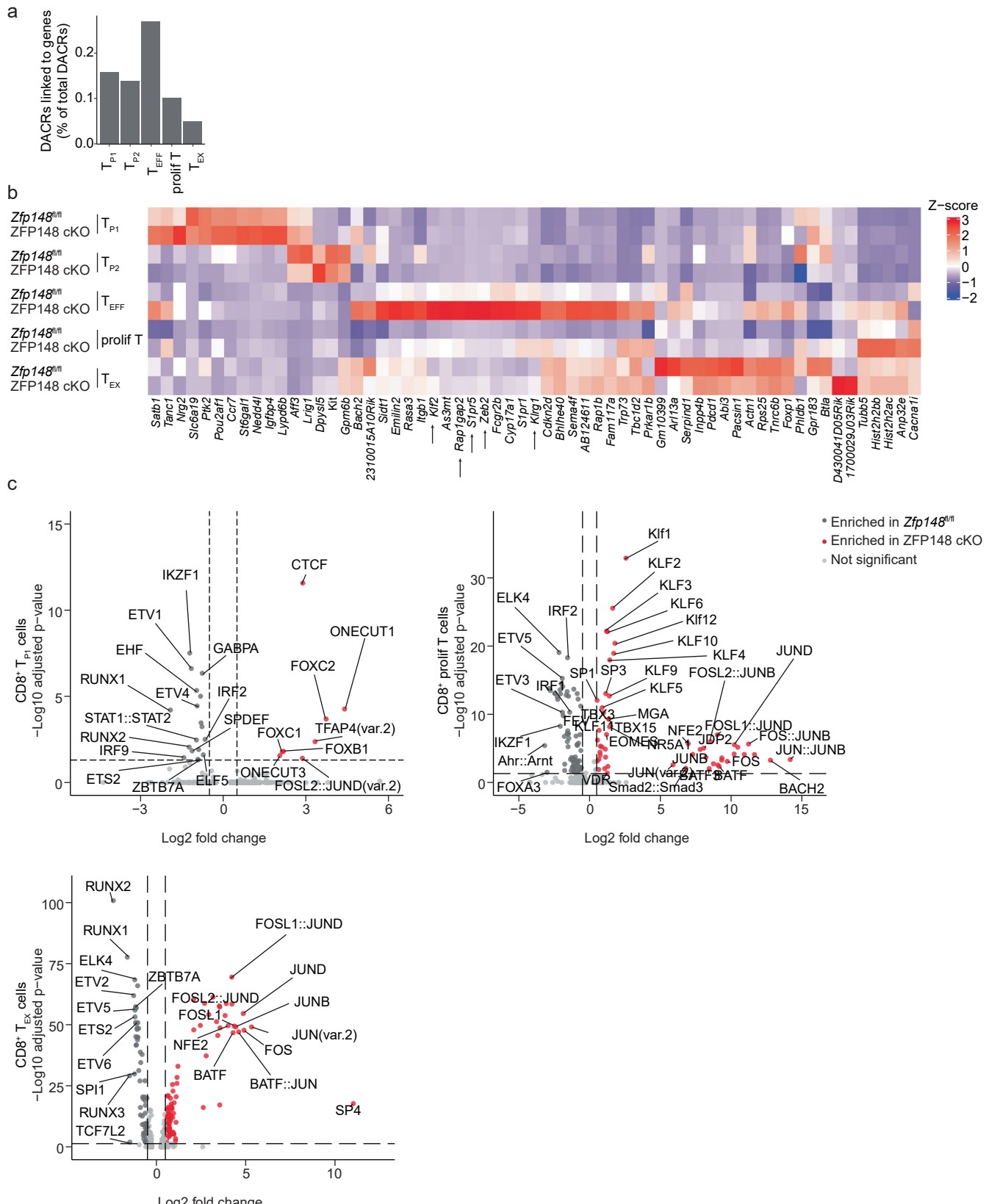

**Extended Data Fig. 7 | ZFP148 epigenetically restrains motif accessibility of CD8⁺ T_EFF cell-driving transcription factors. a**, Frequency of gene-linked DACRs among total DACRs in splenic T_P1, CD8⁺ T_P2, CD8⁺ T_EFF, CD8⁺ prolif T and CD8⁺ T_EX cells in *Zfp148*^fl/fl or ZFP148 cKO mice at day 21 post-LCMV Cl13 infection. **b**, Heatmap showing expression of genes linked to subset-specific DACRs in *Zfp148*^fl/fl versus ZFP148 cKO splenic CD44^hiGP_33-41 Tet⁺CD8⁺ T cells as in **a**.

**c**, Volcano plot showing differentially accessible transcription factor motifs in *Zfp148*^fl/fl versus ZFP148 cKO splenic CD8⁺ T_P1, CD8⁺ prolif T and CD8⁺ T_EX cells as in **a**. Data are pooled from 10 mice for *Zfp148*^fl/fl and 12 mice for ZFP148 cKO. Statistical analysis was performed using a two-sided Wilcoxon rank-sum test with Benjamini–Hochberg correction for multiple-comparisons (**c**).

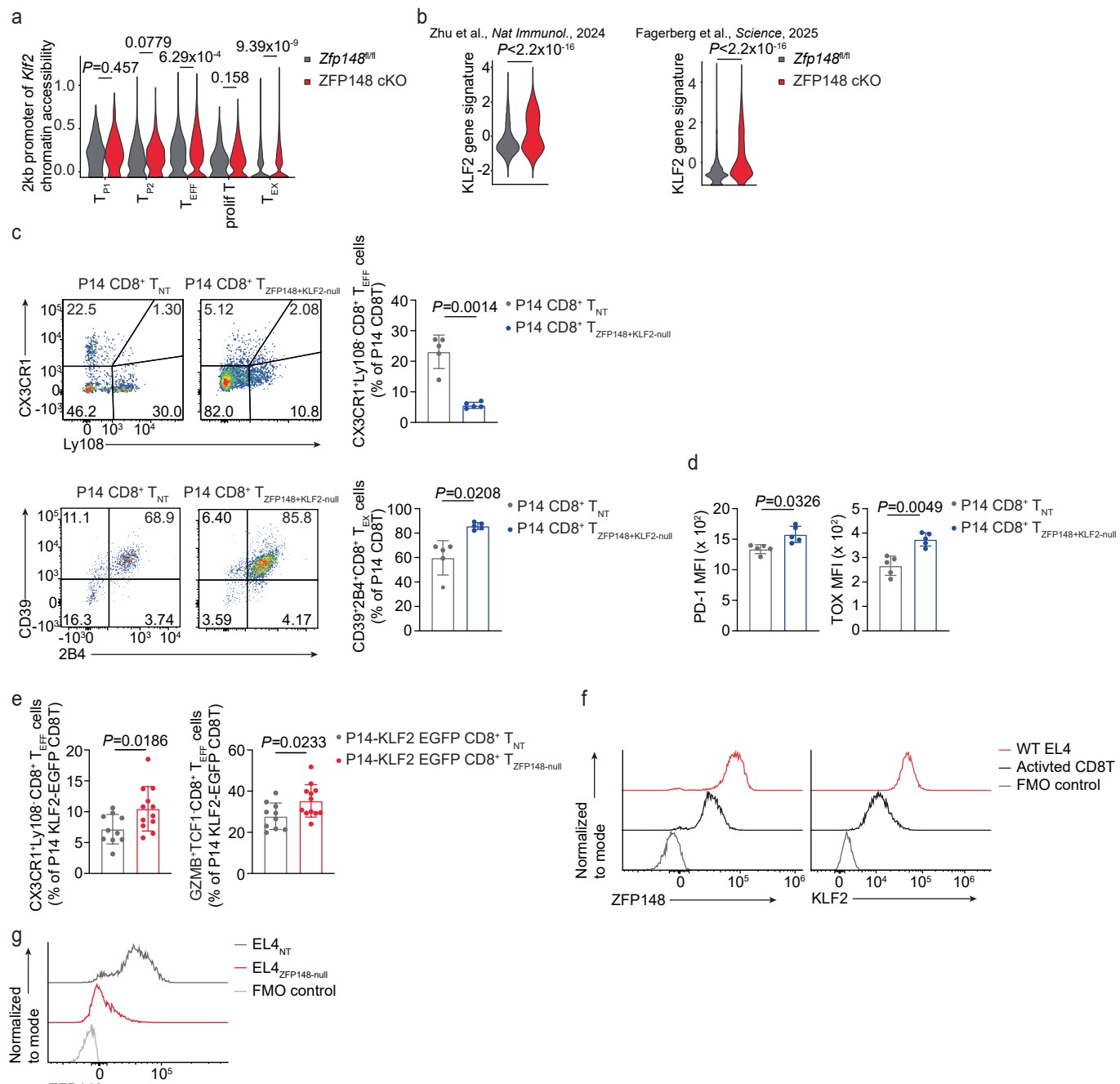

**Extended Data Fig. 8 | ZFP148 epigenetically restrains CD8+ T_EFF differentiation through repressing KLF2. a**, Violin plots showing chromatin accessibility in the 2 kb promoter region of *Klf2* in splenic CD8+ T_P1, CD8+ T_P2, CD8+ T_EFF, CD8+ prolif T and CD8+ T_EX cells in *Zfp148*^fl/fl or ZFP148 cKO mice at day 21 post-LCMV Cl13 infection. **b**, Violin plots showing enrichment of published KLF2 gene signatures[20,30] in *Zfp148*^fl/fl or ZFP148 cKO CD44^hiGP33-41 Tet+CD8+ T cells at day 21 post-LCMV Cl13 infection. **c**, Scatter plots showing CX3CR1 versus Ly108 or CD39 versus 2B4 protein expression (left) and frequency of CX3CR1+Ly108,−CD8+ T_EFF cells or CD39+2B4+CD8+ T_EX cells (right) in transferred P14 CD8+ T_NT cells or P14 CD8+ T_ZFP148+KLF2-null cells in spleens of C57BL/6 WT recipient mice at day 22 post-T cell transfer and LCMV Cl13 infection (n = 5). **d**, MFI of PD-1 and TOX in transferred P14 CD8+ T_NT cells or P14 CD8+ T_ZFP148+KLF2-null cells as in **c. e**, Frequencies of CX3CR1+Ly108−CD8+ T_EFF cells and GZMB+TCF1−CD8+ T_EFF cells in transferred P14 KLF2-EGFP CD8+ T_NT cells (n = 10) or P14 KLF2-EGFP CD8+ T_ZFP148-null cells (n = 12) in spleens of C57BL/6 WT recipient mice at day 22 post-T cell transfer and LCMV Cl13 infection. **f**, Density plots showing expression of ZFP148 and KLF2 protein in WT EL4 cells and CD8+ T cells activated for 3 days in vitro. FMO controls for antibodies staining either ZFP148 or KLF2 protein were included. **g**, Density plots showing expression of ZFP148 protein in EL4_NT and EL4_ZFP148-null cells. FMO control for the antibody staining ZFP148 was included. Data are pooled from 10 mice for *Zfp148*^fl/fl and 12 mice for ZFP148 cKO (**a,b**); representative of 2-3 independent experiments (**c-d,f,g**); pooled from 2 independent experiments (**e**). Data are presented as mean ± s.d. Statistical analysis was performed using a two-sided Wilcoxon rank-sum test (**a-d**).

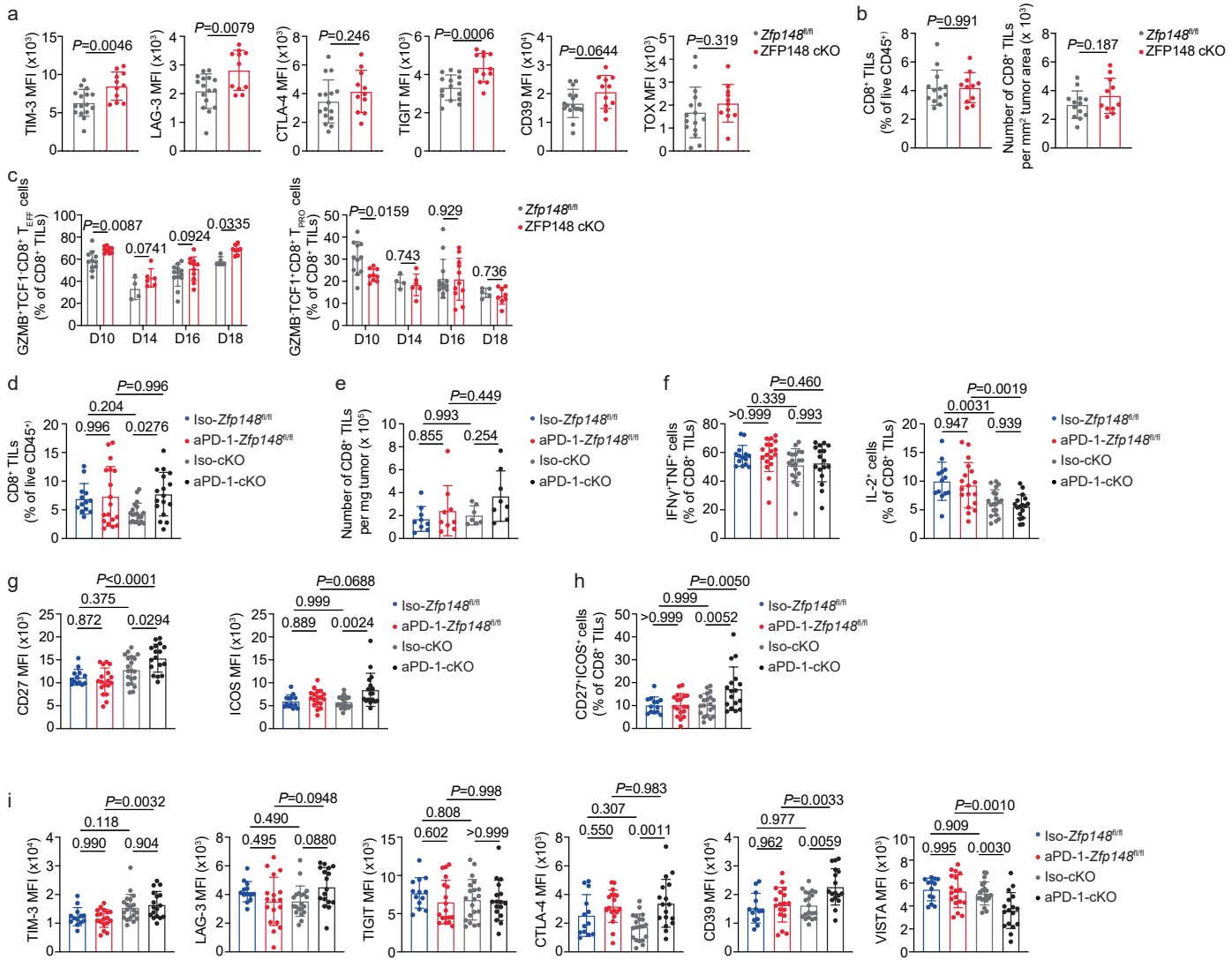

**Extended Data Fig. 9 | ZFP148 deficiency promotes effector differentiation in CD8⁺ TILs and enhances PD-1 blockade efficacy. a**, MFI of TIM-3, LAG-3, CTLA-4, TIGIT, CD39 and TOX in CD8⁺ TILs in *Zfp148*^fl/fl (n = 16) or ZFP148 cKO mice (n = 11) at day 16 post-subcutaneous injection of MC38 tumors. **b**. Frequency (left) and number (right) of CD8⁺ TILs in *Zfp148*^fl/fl (n = 13) or ZFP148 cKO mice (n = 11) as in **a**. **c**, Frequency of GZMB⁺TCF1⁻CD8⁺ T_EFF cells (left) and GZMB⁻TCF1⁺CD8⁺ T_PRO cells (right) in CD8⁺ TILs in *Zfp148*^fl/fl or ZFP148 cKO mice at days 10 (*Zfp148*^fl/fl, n = 11; ZFP148 cKO, n = 9), 14 (*Zfp148*^fl/fl, n = 4; ZFP148 cKO, n = 6), 16 (*Zfp148*^fl/fl, n = 13; ZFP148 cKO, n = 11) and 18 (*Zfp148*^fl/fl, n = 5; ZFP148 cKO, n = 8) post-subcutaneous injection of MC38 tumors. **d,e** Frequency (**d**) and number (**e**) of CD8⁺ TILs in Iso-*Zfp148*^fl/fl (frequency, n = 13; number, n = 9), aPD-1-*Zfp148*^fl/fl (frequency,

n = 18; number, n = 9), Iso-cKO (frequency, n = 20; number, n = 7) or aPD-1-cKO (frequency, n = 17; number, n = 8) mice at day 13 post-subcutaneous injection of MC38 tumors. **f**, Frequency of IFNγ⁺TNF⁺ (left) and IL-2⁺ subsets (right) in CD8⁺ TILs as in **d**, restimulated ex vivo. **g**, MFI of CD27 and ICOS in CD8⁺ TILs as in **d**. **h**. Frequency of CD27⁺ICOS⁻ subset in CD8⁺ TILs as in **d**. **i**, MFI of TIM-3, LAG-3, TIGIT, CTLA-4, CD39 and VISTA in CD8⁺ TILs as in **d**. Data are representative of 2-3 independent experiments (**c**); pooled from 2–3 independent experiments (**a,b,d-i**). Data are presented as mean ± s.d. Statistical analysis was performed using an unpaired two-sided t-test (**a,b**); multiple unpaired two-sided t-tests (**c**); one-way ANOVA followed by Holm–Šídák's multiple-comparisons test (**d-i**).

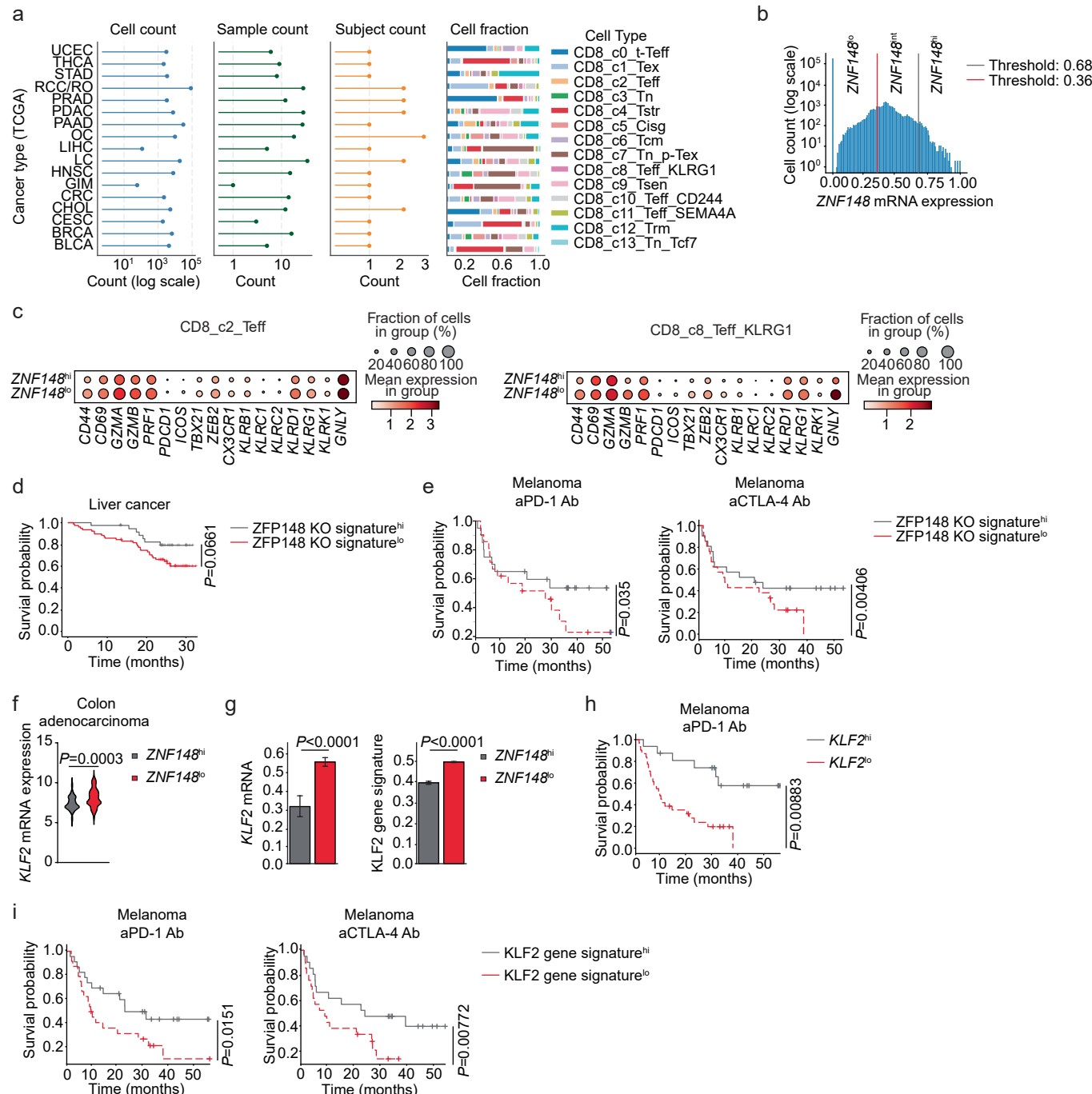

**Extended Data Fig. 10 | Clinical and prognostic relevance of *ZNF148* in human cancers. a**, From left to right, total cell count, sample count, subject count and proportions of major CD8⁺ TIL subsets in each TCGA-classified cancer type of the integrated human cancer scRNA-seq dataset. **b**, Histogram showing *ZNF148* mRNA expression across all CD8⁺ TILs in the integrated human cancer scRNA-seq dataset as in **a**. Vertical dashed lines indicate the cutoffs used to define *ZNF148*ᴸᵒ (left of the red line, above zero), *ZNF148*ⁱⁿᵗ (between the red and grey lines), and *ZNF148*ʰⁱ (right of the grey line) groups. **c**, Dot plots showing mean expression of genes in the 'CD8_c2_Teff' and 'CD8_c8_Teff_KLRG1' subsets of *ZNF148*ʰⁱ and *ZNF148*ᴸᵒ CD8⁺ TILs in the integrated human cancer scRNA-seq dataset. **d**, Kaplan-Meier survival curves showing OS of liver cancer patients stratified by high (n = 39) versus low (n = 77) expression of ZFP148 KO signature in the CD8⁺ T_EFF subset of CD8⁺ TILs ('CD8T_04_GNLY'). **e**, Kaplan-Meier survival curves of

melanoma patients treated with aPD-1 or aCTLA-4 Ab, stratified into ZFP148 KO gene signatureʰⁱ (aPD-1, n = 20; aCTLA-4, n = 21) versus ZFP148 KO gene signatureᴸᵒ (aPD-1, n = 21; aCTLA-4, n = 21) groups. **f**, *KLF2* mRNA expression in *ZNF148*ʰⁱ (n = 51) versus *ZNF148*ᴸᵒ (n = 123) colorectal adenocarcinoma patients from the TCGA database. **g**, Expression of *KLF2* mRNA or a published KLF2 gene signature[30] in *ZNF148*ʰⁱ versus *ZNF148*ᴸᵒ CD8⁺ TILs from the integrated human cancer scRNA-seq dataset. **h**, Kaplan-Meier survival curves showing OS of *KLF2*ʰⁱ (n = 16) versus *KLF2*ᴸᵒ (n = 31) melanoma patients treated with aPD-1 Ab. **i**, Kaplan-Meier survival curves showing OS of melanoma patients treated with aPD-1 or aCTLA-4 Ab, stratified into KLF2 gene signatureʰⁱ (aPD-1, n = 23; aCTLA-4, n = 21) versus KLF2 gene signatureᴸᵒ (aPD-1, n = 24; aCTLA-4, n = 21) groups. Data are presented as mean ± s.d. Statistical analysis was performed using a two-sided Wilcoxon rank-sum test (**f,g**); a log-rank test (**d,e,h,i**).

# Reporting Summary

## Statistics

For all statistical analyses, confirm that the following items are present in the figure legend, table legend, main text, or Methods section.

| n/a | Confirmed | |
|---|---|---|
| ☐ | ☒ | The exact sample size (*n*) for each experimental group/condition, given as a discrete number and unit of measurement |
| ☐ | ☒ | A statement on whether measurements were taken from distinct samples or whether the same sample was measured repeatedly |
| ☐ | ☒ | The statistical test(s) used AND whether they are one- or two-sided *Only common tests should be described solely by name; describe more complex techniques in the Methods section.* |
| ☐ | ☒ | A description of all covariates tested |
| ☐ | ☒ | A description of any assumptions or corrections, such as tests of normality and adjustment for multiple comparisons |
| ☐ | ☒ | A full description of the statistical parameters including central tendency (e.g. means) or other basic estimates (e.g. regression coefficient) AND variation (e.g. standard deviation) or associated estimates of uncertainty (e.g. confidence intervals) |
| ☐ | ☒ | For null hypothesis testing, the test statistic (e.g. *F*, *t*, *r*) with confidence intervals, effect sizes, degrees of freedom and *P* value noted *Give P values as exact values whenever suitable.* |
| ☒ | ☐ | For Bayesian analysis, information on the choice of priors and Markov chain Monte Carlo settings |
| ☒ | ☐ | For hierarchical and complex designs, identification of the appropriate level for tests and full reporting of outcomes |
| ☐ | ☒ | Estimates of effect sizes (e.g. Cohen's *d*, Pearson's *r*), indicating how they were calculated |

*Our web collection on statistics for biologists contains articles on many of the points above.*

## Software and code

Policy information about availability of computer code

| Data collection | Flow cytometry: Cytek® Aurora<br>Cell sorting: Cytek Aurora™ CS System<br>Matched scRNA-seq and scATAC-seq: The illumina Novaseq X plus 840 platform<br>CUT&Tag-seq: The illumina Novaseq X plus 840 platform |
|---|---|
| Data analysis | Statistical analyses for flow cytometry, tumor growth curves, and mouse survival were performed using GraphPad Prism (v.10). Unpaired or paired two-sided t-tests were used for comparisons between two unpaired or paired groups, respectively. One-way ANOVA followed by Tukey's multiple-comparisons test was used for comparisons among three or more groups. One-way ANOVA followed by Holm–Šídák's multiple-comparisons test was used for comparisons between pre-selected pairs among three or more groups. Two-way ANOVA was used to compare time-course curves, with Bonferroni correction for multiple comparisons. The log-rank test was used to compare overall survival of mice across multiple treatment groups, with Bonferroni correction for multiple comparisons.<br>Analyses of mouse scRNA-seq and scATAC-seq data were performed using R (v.4.4.0) with the packages Seurat (v.5.1.0), Signac (v.1.14.0), AUCell (v.1.26.0), slingshot (v.2.12.0), chromVAR (v.1.26.0), clusterProfiler (v.4.12.6), ComplexHeatmap (v.2.20.0), and EnhancedVolcano (v.1.22.0). Differentially expressed genes were identified using the FindMarkers() or FindAllMarkers() functions in Seurat, with statistical significance assessed by a two-sided Wilcoxon rank-sum test, with Benjamini–Hochberg correction for multiple comparisons. Differentially accessible chromatin regions (DACRs) and transcription factor motif accessibility were identified using FindMarkers() or FindAllMarkers() with statistical significance assessed by a two-sided logistic regression likelihood-ratio test, with Benjamini–Hochberg correction for multiple comparisons. Gene Ontology enrichment analysis was performed using a one-sided hypergeometric test, with Benjamini–Hochberg correction for multiple comparisons. A two-sided Wilcoxon rank-sum test was used to compare mRNA expression, promoter chromatin accessibility, motif accessibility, and gene signature scores between Zfp148fl/fl and ZFP148 cKO CD8⁺ T cells. Analyses of the integrated human scRNA-seq |

dataset were performed using Python (v.3.10.9) packages Scanpy (v.1.9.5), Pandas (v.2.0.0), Statsmodels (v.0.14.0), NumPy (v.1.24.2), SciPy (v.1.10.1), Matplotlib (v.3.8.0), Seaborn (v.0.11.2), and scikit-learn (v.1.3.2), as well as R (v.4.3.1) packages Circlize (v.0.4.16), GseaVis (v.0.0.5), Enrichplot (v.1.22.0), GridExtra (v.2.3.0), pheatmap (v.1.0.12), and DEGreport (v.1.38.5). A two-sided Wilcoxon rank-sum test was used for comparisons between two groups. Overall survival between two groups of patients was compared using the log-rank test. A P value ≤ 0.05 (or adjusted P value ≤ 0.05 after multiple-testing correction) was considered statistically significant.

For manuscripts utilizing custom algorithms or software that are central to the research but not yet described in published literature, software must be made available to editors and reviewers. We strongly encourage code deposition in a community repository (e.g. GitHub). See the Nature Portfolio guidelines for submitting code & software for further information.

## Data

Policy information about availability of data

All manuscripts must include a data availability statement. This statement should provide the following information, where applicable:
- Accession codes, unique identifiers, or web links for publicly available datasets
- A description of any restrictions on data availability
- For clinical datasets or third party data, please ensure that the statement adheres to our policy

Parallel scRNA-seq and scATAC-seq and CUT&Tag-seq data are available in the NCBI database under accession numbers GSE297040 and GSE296311, respectively. Source data are provided with this paper. Further information and requests for data should be directed to the corresponding author, Z. Li.

## Research involving human participants, their data, or biological material

Policy information about studies with human participants or human data. See also policy information about sex, gender (identity/presentation), and sexual orientation and race, ethnicity and racism.

| | |
|---|---|
| Reporting on sex and gender | N/A All blood and tumor samples were completely deidentified. |
| Reporting on race, ethnicity, or other socially relevant groupings | N/A All blood and tumor samples were completely deidentified. |
| Population characteristics | The patient cohort had a median age of 68 years. |
| Recruitment | The consenting treatment-naive patient was prospectively enrolled in The Ohio State University Comprehensive Cancer Center bladder cancer tissue registry (Buck-IRB 2018C0181). |
| Ethics oversight | The Ohio State University Comprehensive Cancer Center's bladder cancer tissue registry (Buck-IRB 2018C0181). |

Note that full information on the approval of the study protocol must also be provided in the manuscript.

# Field-specific reporting

Please select the one below that is the best fit for your research. If you are not sure, read the appropriate sections before making your selection.

☒ Life sciences  ☐ Behavioural & social sciences  ☐ Ecological, evolutionary & environmental sciences

For a reference copy of the document with all sections, see nature.com/documents/nr-reporting-summary-flat.pdf

# Life sciences study design

All studies must disclose on these points even when the disclosure is negative.

| | |
|---|---|
| Sample size | No statistical methods were used to predetermine sample size. Sample sizes are indicated in the figure legends and were similar to those reported in previous publications, as referenced in the Methods section. |
| Data exclusions | No data were excluded from this study. |
| Replication | All data were generated from at least two independent experiments with a minimum of three biological replicates, yielding consistent phenotypes to ensure reproducibility, with the exception of the parallel scRNA-seq and scATAC-seq on CD44hiGP33-41 Tet+CD8+ T cells in spleens of Zfp148fl/fl and ZFP148 cKO mice at day 21 post-LCMV Cl13 infection. To minimize inter-mouse variability and enhance reproducibility, cells used for the single-cell multiome experiment were pooled from 10 mice for Zfp148fl/fl and 12 mice for ZFP148 cKO. |
| Randomization | Age- and sex-matched animals were randomly assigned to experimental conditions. |
| Blinding | Blinding was not performed due to the necessity of cage labeling and the use of mice with distinct genotypes and/or sexes. Investigators were |

# Reporting for specific materials, systems and methods

We require information from authors about some types of materials, experimental systems and methods used in many studies. Here, indicate whether each material, system or method listed is relevant to your study. If you are not sure if a list item applies to your research, read the appropriate section before selecting a response.

## Materials & experimental systems

| n/a | Involved in the study |
|-----|----------------------|
| ☐ | ☒ Antibodies |
| ☐ | ☒ Eukaryotic cell lines |
| ☒ | ☐ Palaeontology and archaeology |
| ☐ | ☒ Animals and other organisms |
| ☒ | ☐ Clinical data |
| ☒ | ☐ Dual use research of concern |
| ☒ | ☐ Plants |

## Methods

| n/a | Involved in the study |
|-----|----------------------|
| ☐ | ☒ ChIP-seq |
| ☐ | ☒ Flow cytometry |
| ☒ | ☐ MRI-based neuroimaging |

## Antibodies

Antibodies used

Flow cytometry antibodies
Mouse CD8 BUV496 53-6.7 1:400 BD 569181
Mouse CD3 BUV737 145-2C11 1:200 BD 612771
Mouse CD45 BV510 30-F11 1:100 BioLegend 103138
Mouse CD25 BB515 PC61 1:200 BD 564424
Mouse PD-1 FITC J43 1:200 eBioscience 11-9985-85
Mouse PD-1 Percp-Fire 806 29F.1A12 1:200 BioLegend 135262
Mouse CD11b Alexa Fluor 532 M1/70 1:800 eBioscience 58-0112-82
Mouse Ki-67 BUV395 B56 1:200 BD 564071
Mouse CD27 BUV563 LG.3A10 1:400 BD 741275
Mouse GITR BUV615 DTA-1  1:800 BD 751532
Mouse CD44 BUV661 IM7 1:400 BD 741471
Mouse LAG-3 BUV805 C9B7W 1:100 BD 748540
Mouse CD62L BV421 MEL-14 1:400 BioLegend 104436
Mouse ICOS Super Bright 436 C398.4A 1:400 eBioscience 62-9949-82
Mouse CD95 BV480 Jo2 1:200 BD 746755
Mouse KLRG1 Pacific Orange 2F1 1:200 eBioscience 79-5893-82
Mouse KLRG1 BV605 2F1/KLRG1 1:200 BioLegend 138419
Mouse VISTA Super Bright 600 MIH64 1:400 eBioscience 63-1083-82
Mouse TIGIT BV650 1G9  1:200 BD 744213
Mouse TIM-3 BV711 RMT3-23 1:200 BioLegend 119727
Mouse CD38 BV750 90/CD38 1:400 BD 747103
Mouse T-bet BV786 O4-46 1:100 BD 564141
Mouse EOMES PerCP-eFluor 710 Dan11mag 1:200 eBioscience 46-4875-82
Mouse TOX PE REA473/TXRX10 1:600 Miltenyi Biotech 130-120-716
Mouse TOX APC REA473/TXRX10 1:600 Miltenyi Biotech 130-118-335
Mouse CTLA4 PE-Dazzle594 UC10-4B9  1:400 BioLegend 106318
Mouse CD69 PE-Cy5 H1.2F3  1:1000 BioLegend 104510
Mouse TCF1 PE-Cy7 C63D9 1:600 Cell Signaling Technology 90511S
Mouse SLAMF6 APC 13G3-19D 1:200 eBioscience 17-1508-82
Mouse SLAMF6 PE 13G3 1:200 BD 561540
Mouse BCL-2 Alexa Fluor 647 BCL/10C4 1:200 BioLegend 633510
Mouse Granzyme B Alexa Fluor 700 QA16A02 1:200 BioLegend 372222
Mouse CX3CR1 APC-Fire 750 SA011F11  1:400 BioLegend 149040
Mouse CD39 PerCP-Cy5.5 Y23-1185 1:200 BD 567270
Mouse CD4 APC-Fire810 GK1.5 1:400 BioLegend 100480
Mouse IFN-γ PE-Cy7 XMG1.2 1:1000 BioLegend 505826
Mouse TNF-? APC MP6-XT22 1:2000 BioLegend 506308
Mouse Perforin FITC S16009A 1:100 BioLegend 154310
Mouse IL-2 PE A21001C 1:100 BioLegend 606553
Mouse CD101 PE-Cy7 S18006K 1:200 BioLegend 158210
Mouse CD127 PE A7R34 1:200 BioLegend 135010
Mouse Granzyme A PE 3G8.5 1:800 BioLegend 149703
Mouse 2B4 PE-Cy7 m2B4 (B6)458.1 1:200 BioLegend 133512
Mouse ZFP148 Alexa Fluor 647 H-7 1:400 Santa Cruz Biotechnology sc-137171
Mouse ZFP148 PE H-7 1:400 Santa Cruz Biotechnology sc-137171
Mouse  KLF2 PE E7K8Y 1:200 Cell Signaling Technology 51221S
Mouse H-2D(b) LCMV gp33-41 Tetramer BV421 N/A 1:100 NIH Tetramer Core Facility N/A
Mouse H-2D(b) LCMV gp276-286 Tetramer PE N/A 1:100 NIH Tetramer Core Facility N/A

Human CD45 BV510 2D1 1:200 BioLegend 368526
Human CD3 BV570 UCHT1 1:100 BioLegend 300436
Human CD8 Super bright 436 OKT8 1:100 eBioscience 62-0086-42
Human CD4 APC Fire 810 SK3 1:400 BioLegend 344662
Human FOXP3 eFluor 450 PCH101 1:200 eBioscience 48-4776-41
Human CD11b BUV661 M1/70 1:5000 BD 612977
Human CD56 BV750 5.1H11 1:200 BioLegend 362556
Human CD45RA AF532 HI100 1:200 eBioscience 56-0458-42
Human CD45RO BB515 UCHL1 1:200 BD 564529
Human CD25 sup600 BC96 1:100 eBioscience 63-0259-42
Human PD-1 BUV737 EH12.1  1:100 BD 612791
Human TIM-3 BV711 7D3 1:800 BD 565567
Human TOX APC REA473/TXRX10 1:200 Miltenyi 130-118-335
Human TCF1 PE 7F11A10 1:200 BioLegend 655208
Human CD62L BV421 DREG-56  1:800 BD 563862
Human CTLA4 PEDAZZLE 594 BNI3 1:100 BioLegend 369616
Human LAG-3 PEcy5 3DS223H 1:100 eBioscience 15-2239-42
Human KLRG1 PE-Cy7 MAFA 1:800 BioLegend 138416
Human T-bet BV786 O4-46 1:100 BD 564141
Human Ki-67 BUV395 B56 1:200 BD 564071
Human Granzyme B AF700 N4TL33 1:100 eBioscience 58-8896-42
Human ICOS AF488 C398.4A 1:100 BioLegend 313514
Human CD69 BUV805 FN50 1:100 BD 748763
Human NKG2D BV480 1D11 1:100 BD 746404
Human NKG2A BUV615 131411 1:100 BD 752302
Human TIGIT BV650 741182 1:200 BD 747840
Human KIR2DL1 APCCy7 HP-MA4 1:200 BioLegend 339520
Human KIR3DL1 BUV563 DX9 1:100 BD 748923
Human CD27 SPARK NIR 685 O323 1:400 BioLegend 302856
Human CCR7 BUV496 2-L1-A 1:100 BD 749827
Human BCL2 AF647 Bcl-2/100 1:800 BD 563600
Human EOMES PE-Cy5.5 WD1928 1:400 eBioscience 35-4877-42
Human CD28 percp5.5 CD28.2 1:100 eBioscience 45-0289-42
Human ZFP148 Alexa Fluor 488 H-7 1:400 Santa Cruz Biotechnology sc-137171
Human IFN-γ FITC 4S.B3 1:400 BioLegend 502506
Human TNF-? PE-Cy7 MAb11 1:200 BioLegend 502930
Human IL-2 PE MQ1-17H12 1:200 eBioscience 12-7029-42

CUT&Tag-seq antibodies
Rabbit Anti-ZBP89 (ZFP148) Bethyl Laboratories # A303-116A
Anti-Rabbit Secondary Antibody  EpiCypher, # SKU: 13-0047
Rabbit IgG Negative Control Antibody EpiCypher, # SKU: 13-0042

Cell culture antibodies
anti-mouse CD3ε 145-2C11 BioLegend
anti-mouse CD28 37.51 BioLegend

| Validation | All antibodies were obtained from reputable and established commercial sources. Prior to experimental use, each antibody was tested and titrated to ensure optimal performance. Additional validation details are available on the manufacturers' websites using the catalog numbers provided in the supplementary materials. For flow cytometry, antibody specificity was validated using fluorescence minus one (FMO) controls and appropriate co-staining strategies. |
|---|---|

# Eukaryotic cell lines

Policy information about cell lines and Sex and Gender in Research

| Cell line source(s) | B16-GP cell line was kindly provided by A. Wieland at The Ohio State University. B16-GP cells were cultured in RPMI 1640 medium (Gibco, 11875-093) with 10% heat-inactivated fetal bovine serum (FBS) (Gibco, 10082-147) and 1% penicillin/ streptomycin (Gibco, 15140-122). The MC38 cell line was purchased from Kerafast (ENH204-FP). MC38 cells were cultured in Dulbecco's modified Eagle's medium (DMEM; Gibco, 11965-092) with 10% FBS and 1% penicillin/streptomycin. EL4 cell line was kindly provided by K. Oestreich at The Ohio State University. EL4 cells were cultured in RPMI-1640 with 10% FBS and 1% penicillin/streptomycin. |
|---|---|
| Authentication | MC38, B16-GP and EL4 cells were not independently authenticated |
| Mycoplasma contamination | All cell lines were tested negative for Mycoplasma contamination. |
| Commonly misidentified lines (See ICLAC register) | N.A. |

# Animals and other research organisms

Policy information about [studies involving animals](); [ARRIVE guidelines]() recommended for reporting animal research, and [Sex and Gender in Research]()

| | |
|---|---|
| Laboratory animals | C57BL/6J (strain 000664) mice were obtained from the Jackson Laboratory. CD8-specific ZFP148-deficient mice were generated by crossing E8iCre (the Jackson Laboratory, strain 008766) mice with Zfp148fl/fl mice kindly provided by J. L. Merchant at University of Arizona. P14 mice were kindly provided by S. M. Kaech at The Salk Institute for Biological Studies. KLF2GFP reporter mice were kindly provided by S. C. Jameson at the University of Minnesota via W. Cui at Northwestern University and crossed with P14 mice at Northwestern University to generate P14 KLF2-EGFP mice. |
| Wild animals | This study did not involve wild animals |
| Reporting on sex | Both male and female mice aged 8–10 weeks were used. |
| Field-collected samples | No field collected samples were used in this study |
| Ethics oversight | Mice were maintained in the animal facility at The Ohio State University under standard conditions (ambient temperature 20–24 °C, relative humidity 30–70%, 12-h dark–light cycle (lights on from 6:00 to 18:00)). Both male and female mice aged 8–10 weeks were used. All procedures were performed in strict accordance with the NIH Guide for the Care and Use of Laboratory Animals and approved by the Committee on the Ethics of Animal Experiments of The Ohio State University. The Ohio State University Institutional Animal Care and Use Committee (IACUC; protocol 2019A00000075), Institutional Biosafety Committee (IBC; protocol 2019R00000046) |

Note that full information on the approval of the study protocol must also be provided in the manuscript.

# Plants

| | |
|---|---|
| Seed stocks | Report on the source of all seed stocks or other plant material used. If applicable, state the seed stock centre and catalogue number. If plant specimens were collected from the field, describe the collection location, date and sampling procedures. |
| Novel plant genotypes | Describe the methods by which all novel plant genotypes were produced. This includes those generated by transgenic approaches, gene editing, chemical/radiation-based mutagenesis and hybridization. For transgenic lines, describe the transformation method, the number of independent lines analyzed and the generation upon which experiments were performed. For gene-edited lines, describe the editor used, the endogenous sequence targeted for editing, the targeting guide RNA sequence (if applicable) and how the editor was applied. |
| Authentication | Describe any authentication procedures for each seed stock used or novel genotype generated. Describe any experiments used to assess the effect of a mutation and, where applicable, how potential secondary effects (e.g. second site T-DNA insertions, mosiacism, off-target gene editing) were examined. |

# ChIP-seq

## Data deposition

☒ Confirm that both raw and final processed data have been deposited in a public database such as [GEO]().

☒ Confirm that you have deposited or provided access to graph files (e.g. BED files) for the called peaks.

| | |
|---|---|
| Data access links<br>May remain private before publication. | CUT&Tag-seq data are available in the NCBI database under accession numbers GSE296311 (https://www.ncbi.nlm.nih.gov/geo/query/acc.cgi?acc=GSE296311) |
| Files in database submission | ZFP148_R1_001.fastq.gz<br>ZFP148_R2_001.fastq.gz<br>IgG_R1_001.fastq.gz<br>IgG_R2_001.fastq.gz<br>ZFP148.bigWig<br>IgG.bigWig |
| Genome browser session<br>(e.g. [UCSC]()) | https://tinyurl.com/2s43vcju |

## Methodology

| | |
|---|---|
| Replicates | CUT&Tag-seq data are from single replicate. Each replicate was pooled together from 5 mice. |
| Sequencing depth | 50 to 70 million reads were generated for each library. |
| Antibodies | Rabbit Anti-ZBP89 (ZFP148) Bethyl Laboratories # A303-116A<br>Anti-Rabbit Secondary Antibody EpiCypher, # SKU: 13-0047<br>Rabbit IgG Negative Control Antibody EpiCypher, # SKU: 13-0042 |

| Peak calling parameters | Peaks were called using SEACR (v.1.3) |
|---|---|
| Data quality | All samples passed FastQC sequencing read quality assessment.<br>133483 peaks called for the ZFP148 sample |
| Software | CUT&Tag sequencing data were processed using the nf-core/cutandrun pipeline (v3.0.0) [https://nf-co.re/cutandrun/3.2.2/], a community-curated Nextflow pipeline. Raw sequencing reads were first subjected to adapter trimming using fastp (v0.23.2), followed by alignment to the mouse reference genome (mm10) using Bowtie2 (v2.4.4). Aligned reads were filtered to remove low-quality mappings, PCR duplicates (using Picard MarkDuplicates v2.27.5), and mitochondrial reads. Peaks were called using SEACR (v1.3) in "relaxed" mode with appropriate IgG or input control normalization. Genome-wide signal tracks were generated using deepTools (v3.5.1) and IGV (v2.18.4)for visualization. Quality control metrics, including fragment size distribution, duplication rates, and library complexity, were assessed and summarized using MultiQC (v1.13). All steps were run with default settings unless otherwise specified. |

## Flow Cytometry

### Plots

Confirm that:

☐ The axis labels state the marker and fluorochrome used (e.g. CD4-FITC).

☒ The axis scales are clearly visible. Include numbers along axes only for bottom left plot of group (a 'group' is an analysis of identical markers).

☒ All plots are contour plots with outliers or pseudocolor plots.

☒ A numerical value for number of cells or percentage (with statistics) is provided.

### Methodology

| Sample preparation | Mouse samples<br>Mouse spleens were mechanically disrupted, washed once with ice-cold PBS, subjected to red blood cell lysis (BioLegend, 420302), passed through 70-µm cell strainers, and resuspended as single-cell suspensions.<br>For liver lymphocyte isolation, mice were perfused with ice-cold PBS via the hepatic portal vein. Livers were mechanically dissociated through 100-µm strainers, followed by lymphocyte isolation using Percoll gradient centrifugation (44% Percoll in RPMI over 67% Percoll in PBS; 450 × g for 20 min at room temperature) and red blood cell lysis.<br>For lung lymphocyte isolation, mice were perfused with ice-cold PBS. Lungs were minced and digested with Collagenase Type III (Worthington Biochemical Corporation, LS004182) for 90 min at 37 °C, passed through 70-µm strainers, and lymphocytes were isolated by Percoll gradient centrifugation (44% Percoll in RPMI over 67% Percoll in PBS; 500 × g for 20 min at room temperature).<br>Tumors were mechanically dissociated and digested with Collagenase Type I and Collagenase Type IV (Worthington Biochemical Corporation, LS004196 and LS004188) for 30 min at 37 °C with shaking at 125 rpm. Digestion was quenched with ice-cold PBS containing 2% bovine serum albumin, and red blood cell lysis (BioLegend) was performed as needed before filtration through 70-µm strainers.<br><br>Human samples<br>Clinical specimens were processed on the day of radical cystectomy. Tumor tissue was manually dissociated, centrifuged (600 × g for 5 min at 4°C), resuspended in human tumor dissociation enzyme solution (Miltenyi Biotec, 130-095-929), and homogenized using a gentleMACS semi-automated dissociator (Miltenyi Biotec). Homogenized tissue was incubated at 37°C for 20min under continuous rotation on a MACSmix Tube Rotator (Miltenyi Biotec). Following addition of 2% bovine serum albumin (BSA), single-cell suspensions were filtered twice through a 70µm cell strainer, washed with PBS, subjected to red blood cell lysis (BioLegend, 420302), and resuspended in RPMI medium.<br>Peripheral blood was collected in heparin–EDTA tubes and processed by Ficoll–Hypaque (Sigma) density gradient centrifugation to isolate peripheral blood mononuclear cells (PBMCs). |
|---|---|
| Instrument | Cytek® Aurora, Cytek Aurora™ CS System |
| Software | All results were analyzed with FlowJo software (v10.7.1, TreeStar) or OMIQ Flow Cytometry software (Dotmatics). |
| Cell population abundance | For sorting with Cytek Aurora™ CS System, cells were reanalyzed after sorting, achieving and confirming a purity of >90%. |
| Gating strategy | The gating strategy for sorting out CD44hiGP33–41 Tet+CD8+ T cells from spleens of Zfp148fl/fl and ZFP148 cKO mice infected with LCMV Cl13 is shown in Extended Data Fig. 6a.<br>For in vitro–stimulated mouse or human CD8+ T cells, lymphocytes were first identified using SSC-A versus FSC-A based on cell size and granularity. Doublets were excluded by FSC-H versus FSC-A gating. Live CD8+ T cells were then selected by excluding viability dye–positive cells and gating on CD8+ cells.<br>For CD44hiGP33–41 Tet+CD8+ T cells or CD44hiGP276–286 Tet+CD8+ T cells from spleens or inguinal lymph nodes of C57BL/6 WT, Zfp148fl/fl, or ZFP148 cKO mice following LCMV Cl13 or LCMV Armstrong infection, lymphocytes were first identified by SSC-A versus FSC-A, and doublets were excluded as described above. Live CD45+ immune cells were then selected, followed by CD3+ T cells (CD3+CD11b−), CD8+ T cells (CD8+CD4−), and finally CD44hi tetramer-positive antigen-specific populations.<br>For analysis of transferred P14 or P14 KLF2-EGFP CD8+ T cells in spleens of C57BL/6 WT recipient mice at day 21 post-transfer and LCMV Cl13 infection, lymphocytes were first identified by SSC-A versus FSC-A, and doublets were excluded as above. Live CD45+ immune cells were selected, followed by CD3+ T cells (CD3+CD11b−), CD8+ T cells (CD8+CD4−), and |

transferred CD45.1+ P14 or P14 KLF2-EGFP cells.

For mouse tumor-infiltrating CD8+ T cells, lymphocytes were first identified by SSC-A versus FSC-A, and doublets were excluded as above. Live CD45+ immune cells were selected, followed by CD3+ T cells (CD3+CD11b−) and CD8+ T cells (CD8 +CD4−).

For human CD8+ T cells isolated from peripheral blood mononuclear cells or muscle-invasive bladder tumors, lymphocytes and singlets were gated as described above. Live CD45+ immune cells were selected, followed by CD3+ T cells (CD3+CD11B−) and CD8+ T cells (CD8+CD4−). Activated or naïve CD8+ T cells were defined as CD45RA−CD45RO+ or CD45RA+CD45RO− populations, respectively.

Appropriate fluorescence minus one (FMO) controls and positive controls were included in each experiment to define gating boundaries.

☒ Tick this box to confirm that a figure exemplifying the gating strategy is provided in the Supplementary Information.

