## [Peer Review File · Nature Immunology]

ZFP148 is a transcriptional repressor of cytolytic effector CD8⁺ T cell differentiation

Corresponding Author: Dr Zihai Li

Version 0:

Decision Letter:

11th Dec 2024

Dear Dr. Li,

Thank you for your message asking whether Nature Immunology would be interested in your manuscript. While the observations on the role of ZFP148 in regulating a cytotoxic effector program in exhausted T cells are interesting, it remains unclear if the manuscript addresses the question of what regulates the role of ZFP148 and the dynamics between effector-like and terminally exhausted T cells. We believe such insight would be required to increase the impact of the manuscript. As such, we are not able to determine the suitability of the manuscript, but you can feel free to submit the full manuscript for evaluation (we've provided a link below).

We want to ensure that the methods and statistics reporting in our papers are of the highest quality. To that end, we ask authors to fill out a reporting summary that collects information on experimental design and reagents. The summary can be found at <https://www.nature.com/documents/nr-reporting-summary.pdf>

The Editorial Policy Checklist can be found here: <https://www.nature.com/documents/nr-editorial-policy-checklist.pdf>

Please mention this correspondence in your cover letter when you submit, and thank you for your interest in Nature Immunology.

The complete manuscript can be submitted through the link below:

Link Redacted

Sincerely,

Ioana Staicu, Ph.D.
Senior Editor
Nature Immunology

Tel: 212-726-9207
Fax: 212-696-9752
www.nature.com/ni

Version 1:

Decision Letter:

25th Apr 2025

Dear Dr. Li,

Thank you for submitting your manuscript entitled "ZFP148 is a novel transcriptional checkpoint for effector CD8+ T cell differentiation", for consideration. We have carefully evaluated the work, but regret that we will not be sending it out for in-depth review. Although your results are interesting, the editors consider the current manuscript is better suited for to a more specialized immunology journal.

Although we very much regret that we cannot offer to publish your paper in Nature Immunology for editorial reasons, the editors at Nature Communications would send it out for formal peer review. Should you wish to have your paper considered by Nature Communications, please use the link to the manuscript transfer service in the footnote. It is not necessary to reformat your paper at this point. Please note that Nature Communications is a fully open access journal; all submissions and transfers received, if accepted, will be open access upon publication and subject to an article processing charge. Please see the open access pages (http://www.nature.com/ncomms/open_access/index.html) for information about article processing charges, open access funding, and advice and support.

As you may know, we decline a substantial proportion of manuscripts without sending them to referees, so that they may be sent elsewhere without delay. These editorial judgments are based on such considerations as the degree of advance provided, the breadth of potential interest to researchers and timeliness. Please be assured that this editorial decision does not represent a criticism of the quality of your work, nor are we questioning its value to others working in this area.

I am sorry that we cannot respond more positively on this occasion. I certainly appreciate your interest in Nature Immunology and wish you continued success in your investigations.

Sincerely,

Ioana Staicu, Ph.D.
Senior Editor
Nature Immunology

Tel: 212-726-9207
Fax: 212-696-9752
www.nature.com/ni

To transfer your manuscript to Nature Communications please use our manuscript transfer portal. You will not have to re-supply manuscript metadata and files, unless you wish to make modifications. For more information, please see our [manuscript transfer FAQ](http://www.nature.com/authors/author_resources/transfer_manuscripts.html?WT.mc_id=EMI_NPG_1511_AUTHORTRANSF&WT.ec_id=AUTHOR) page.

Version 2:

Decision Letter:

20th May 2025

Dear Dr. Li,

Your Article, "ZFP148 is a novel transcriptional checkpoint for effector CD8+ T cell differentiation" has now been seen by 3 referees. While the work is of potential interest, the reviewers have raised substantial concerns that must be addressed. As such, we cannot accept the current manuscript for publication, but would be interested in considering a revised version that addresses these serious concerns, as long as novelty is not compromised in the interim.

Please revise the manuscript to address all issues raised by the referees. We consider it is important to address the issues raised by referee #2 regarding the interpretation of the data and data presentation, as well as the mechanistic link between ZFP148 and KLF2. At resubmission, please include a point-by-point "Response to referees" detailing how you have addressed each referee comment (please specify the page and figure number where the new data can be found in the revised manuscript and please highlight the changes in the manuscript as well). This response will be sent back to the referees along with the revised manuscript.

In addition, please include a revised version of any required reporting checklist. It will be available to referees (and, potentially, statisticians) to aid in their evaluation if the manuscript goes back for peer review. A revised checklist is essential for re-review of the paper.

The Reporting Summary can be found here:
<https://www.nature.com/documents/nr-reporting-summary.pdf>

When submitting the revised version of your manuscript, please pay close attention to our [Digital Image Integrity Guidelines](https://www.nature.com/nature-portfolio/editorial-policies/image-integrity) and

to the following points below:

Extended Data figures and tables are online-only (appearing in the online PDF and full-text HTML version of the paper), peer-reviewed display items that provide essential background to the Article but are not included in the printed version of the paper due to space constraints or being of interest only to a few specialists. A maximum of ten Extended Data display items (figures and tables) is typically permitted. When re-submitting your manuscript, please ensure that any supplementary figures and tables that are more critical to the manuscript's conclusions are converted to Extended data to increase these data's visibility.

Link Redacted

We hope to receive a suitably revised manuscript within 6 months. If you cannot send it within this time, please let us know. We will be happy to consider your revision so long as nothing similar has been accepted for publication at Nature Immunology or published elsewhere.

Nature Immunology is committed to improving transparency in authorship. As part of our efforts in this direction, we are now requesting that all authors identified as 'corresponding author' on published papers create and link their Open Researcher and Contributor Identifier (ORCID) with their account on the Manuscript Tracking System (MTS), prior to acceptance. ORCID helps the scientific community achieve unambiguous attribution of all scholarly contributions. You can create and link your ORCID from the home page of the MTS by clicking on 'Modify my Springer Nature account'. For more information please visit www.springernature.com/orcid.

Thank you for the opportunity to review your work.

Sincerely,

Ioana Staicu, Ph.D.
Senior Editor
Nature Immunology

Tel: 212-726-9207
Fax: 212-696-9752
www.nature.com/ni

Reviewer comments:

Reviewer #1

(Remarks to the Author)

In the manuscript titled "ZFP148 is a novel transcriptional checkpoint for effector CD8+ T cell differentiation", the authors demonstrated ZFP148 as a novel negative regulator of effector CD8 T cell differentiation. They showed that ZFP148 KO had increased differentiation to the CX3CR1+ Teff subset and upregulated cytokines and granzyme B. ZFP148 KO also had reduced Tpro suggesting a role of ZFP148 in inhibiting the differentiation from Tpro to Teff. The authors further demonstrated that pro-Teff transcription factor Klf2 as a direct target of ZFP148. ZFP148 KO mice are more sensitive to aPD1 induced tumor control than wild type mice. The study is interesting and novel. Below are my suggestions.

1. Please provide cell numbers in Fig 1h and 6c.
2. Please provide fold changes in Fig 2e-h. Although they are statistically significant, some differences are difficult to assess.
3. How does ZFP148 deficiency affect the virus titer in mice infected with LCMV CI13?

4. They authors sorted CD44^{high} GP33-41⁺ CD8⁺ T cells for their scATAC+gene expression experiment. Yet, there is a naïve T cell population in Fig 3d.
5. ZFP148 KO increased the percentage of proliferating cells but not the number of tetramer⁺ T cells. Does ZFP148 KO increase cell death?
6. T_{pro} is the key driver of the response to aPD1. However, the authors showed that ZFP148 KO reduced T_{pro} but increased the sensitivity to aPD1. Some discussion would be highly appreciated.
7. Information of the anti-perforin antibody needs to be included.

Reviewer #2

(Remarks to the Author)

Persistent antigen exposure gives rise to dysfunctional “exhausted” CD8 T cell populations that exhibit loss of cytotoxicity and proliferative potential. These cells are derived from a stem-like pool of progenitor cells (TPEX), however, in instances of chronic infection or cancer, the efficient de novo generation of cytolytic effector populations from T_{pex} is outpaced by viral replication or tumor growth, respectively. Thus, there is great interest in determining how CD8 T cell responses may be safeguarded from exhaustion.

In this manuscript, Xiao and colleagues identify ZFP148 as a novel regulator of cytotoxic CD8 T cell differentiation. Although it is unclear how they identified Zfp148 as the main subject of this study, the major findings of this study are:

- 1) During acute and chronic infection, Zfp148 is expressed in antigen-specific CD8⁺ T cells. The authors show that there is marginally lower expression in cells responding to chronic LCMV infection compared to acute LCMV infection. Specifically, there was higher expression in acute effector and resting memory states compared to those responding to chronic LCMV infection as determined by flow and scRNA sequencing. Zfp148 is almost uniformly expressed in all subsets with a slightly higher level in Ly108⁺ TCF-1⁺ GZMB⁻ TPEX cells.
- 2) In CD8 T cell-specific Zfp148 cKO mice, higher frequencies of antigen-specific CD8⁺ T cells exhibit a CX3CR1⁺ effector state while Ly108⁺ TPEX and Ly108⁻ CX3CR1⁻ T_{EX} cells are reduced both in frequencies and numbers, confirmed by both flow and scRNA-seq. This indicates that absence of Zfp148 corresponds with greater effector development.
- 3) ATAC-seq results were consistent with RNA-seq, with more accessibility at loci encoding genes more highly expressed in CX3CR1⁺ effectors, such as Zeb2 and S1pr2, in Zfp148-KO CD8 T cells. Increased accessibility in the potential cis-acting elements containing T-bet consensus motifs in KO TPEX cells suggest a potential overlapping function with FLI-1 or BACH2 that restrict mobilization of TPEX into effectors.
- 4) As a potential mechanism, the authors proposed that Zfp148 represses KLF2 based on increased KLF2 expression in the proliferating subset accompanied by increased accessibility of KLF2 motifs in the same subsets.
- 5) Finally, the findings were extended to CD8⁺ T cells responding to tumors both in human and mouse, showing enhanced anti-tumor immunity with anti-PD1 in Zfp148-cko mice as well as correlation to pronounced T cell responses with colon and gastric cancer patients with Zfp148-low tumors.

Overall, the authors findings with phenotypes in mice and correlation to human patients are interesting and this is the first report showing the contribution of this transcription factors in CD8⁺ T cell response. Analyses of mouse phenotypes were conducted reasonably thoroughly and clinical data showing inverse correlation between Zfp148 and patients' prognosis. However, this reviewer sees a few major concerns with regards to the mechanisms, which must be clarified with additional experiments as follows:

- 1) Although the authors repeatedly stated that Zfp148 expression is the highest in TPEX or resting T cells, its change in expression levels is very small even though it is unnecessarily exaggerated in Fig 1a with an inappropriate Y-axis scaling. The protein levels look the same, and KO phenotypes in cultured T cells show statistically significant but biologically questionable changes in effector molecule expression, suggesting that most of the in vivo phenotypes resulted from increased frequencies in effector cells rather than increased effector function on a per-cell basis. These are not major issues as long as the authors revise the text to reflect real funding with unnecessary exaggeration. However, the findings the authors have now suggest that the presence of co-operating factors whose expression changes more dramatically between TPEX and effector cells may account for the differences. This has to be clearly solved. An obvious candidate is Zfp281, which is partially redundant with Zfp148 in T cells as demonstrated by Bosselut et al. but the authors conclusion that its high expression in TPEX suppress their differentiation may not be correct.
- 2) The weakest and potentially problematic part of the paper is their attempt to mechanically connected to KLF2. As stated above, it is very likely that the enrichment of Klf2 gene activity in Fig. 5a reflects changes in KLF2-hi population rather than higher KLF2 gene activity on a per cell basis in Zfp148-KO cells. The authors seem to assume that Zfp148 binds to the +10 kb region of the Klf2 locus to enhance its expression, but this has to be directly tested by ablation of the specific region rather than ablation of Klf2 by RNP electroporation. Since Klf2 ablation is sufficient to reduce CX3CR1⁺ cells in Fig 5i-k, the DKO phenotype in Fig. 5i is uninterpretable and the functional connection is very questionable.
- 3) In Fig.3f, the authors show predicted differentiation trajectories of CD8 T cells in clone 13 infection. While these types of analyses often show discrepancies from more biological readouts, how real are the results from this analysis with effector transit to proliferating population or naïve cells to TPEX? Are the naïve cells endogenous GP33-specific CD8 T cells for which such high frequencies are unrealistic?
- 4) Do Zfp148 cKO mice show enhanced viral control in LCMV-clone 13 infection or superior lysis activity ex vivo?
- 5) Does Zfp148 conditional deletion impact memory CD8 T cell (TCM, TEM, TRM) differentiation following acute infection?
- 6) In Fig. 6 (and also relevant to ICB responsiveness in patients), the data show enhanced response to anti-PD1 in the MC38 in mice. Given that there is more effector differentiation in the LCMV model without PD1 blockade, it is surprising that

there is no difference in anti-tumor immunity without anti-PD1 (closed circle versus closed square cohort). Since TPEX cells have been thought to be the major responding cells to anti-PD1 and TPEX cells are reduced in Zfp148-cKO mice, how the authors interpret this phenotype?

Reviewer #3

(Remarks to the Author)

The manuscript by Xiao et al. presents a comprehensive analysis of the transcription factor ZFP148 in the context of T cell exhaustion, utilizing both chronic LCMV infection and tumor models. The authors first examine ZFP148 expression using existing RNAseq data, validated via flow cytometry. By leveraging a CD8⁺ T cell-specific ZFP148 knockout model, they demonstrate enhanced effector differentiation, supported by flow cytometry and single-cell RNA sequencing. Integrative single-cell ATAC and RNA sequencing further elucidate the epigenetic landscape shaped by ZFP148, identifying KLF2 as a putative downstream target. This is functionally validated through CRISPR-RNP-mediated deletion of KLF2 and/or ZFP148. Finally, the authors show that ZFP148 deficiency improves responsiveness to immune checkpoint blockade in the MC38 tumor model and correlate these findings with human transcriptomic data.

Overall, this is a well-executed and conceptually interesting study that characterizes a previously underappreciated transcription factor and its role in restraining effector differentiation in exhausted T cells. The work is likely to be of interest to both researchers and clinician scientists. That said, a few issues should be addressed to further strengthen the manuscript and ensure clarity and rigor of the conclusions.

Major Comments:

- Further characterization in the LCMV model is warranted. The authors should extend their flow cytometric analysis of ZFP148 expression across different CD8⁺ T cell subsets throughout the course of chronic LCMV infection. This will help contextualize its dynamic regulation and functional relevance. Additionally, the data in Fig. 2 would benefit from further characterization to substantiate claims of enhanced effector differentiation and quality. This should include: cytokine production analyses following restim (e.g., IFN- γ , TNF, IL-2) and additional assessment of progenitor/effector markers such as CD62L, CD69, c-KIT and CD101. Critically, the authors need to extend these analyses to multiple time points, including later stages post-day 21. This would further help to identify if the ability to continuously sustain a T cell response is impaired. The authors should also perform functional assays, such as cytotoxicity (killing assays), to directly assess effector capabilities in the absence of ZFP148. The in vitro activation data (Fig. 2i-k) provide limited insight into the in vivo kinetics of differentiation. These could be replaced or supplemented with analyses of effector formation during the expansion phase of acute vs. chronic infection to more directly address the notion of "accelerated effector differentiation."
- The viral burden and infection control in CD8-specific ZFP148-deficient mice must be addressed. All LCMV data derive from E8i-Cre-driven ZFP148-deficient mice. It is essential that the authors report viral titers to exclude the possibility that observed transcriptional or phenotypic changes are secondary to altered viral control, which could question all the multiomic analyses.
- The authors report increased PD-1⁺GzmB⁺ TILs in ZFP148-deficient tumors (Fig. 6C), yet Fig. 6J suggests similar T cell numbers between genotypes. This discrepancy requires clarification. Furthermore, enhanced characterization of the anti-tumor response (e.g., tumor-infiltrating T cell function, exhaustion marker expression, cytokine production) would add depth to the tumor-related conclusions. The inclusion of an additional tumor model (e.g., B16, At3) would also enhance the robustness of the findings.
- Finally, additional mechanistic insights in the regulation of ZFP148 would be highly beneficial and significantly strengthen the impact of the study.

Minor comments/suggestions:

- The authors may consider removing Fig. 1G-I and the corresponding text (lines 147-159), as the functional analysis using knockout mice renders this overexpression experiment redundant and potentially confusing at this point in the manuscript.
- Fig. 2F, the use of a log scale would better visualize differences in subset frequencies.
- Line 201 should reference Extended Data Fig. 1a (not Fig. 1a).

Version 3:

Decision Letter:

5th Dec 2025

Dear Dr. Li,

Your Article, "ZFP148 is a novel transcriptional checkpoint for effector CD8⁺ T cell differentiation" has now been seen by 3 referees. Although we are interested in the possibility of publishing your study in Nature Immunology, the issues raised by the referees need to be addressed.

Please revise to address all issues raised by the referees. At resubmission, please include a "Response to referees"

detailing, point-by-point, how you addressed each referee comment (please specify the page number and the figures where the new data is found). If no action was taken to address a point, you must provide a compelling argument. This response will be sent back to the referees along with the revised manuscript.

Please include a revised version of any required reporting checklist. It will be available to referees to aid in their evaluation. The Reporting Summary can be found here:

<https://www.nature.com/documents/hr-reporting-summary.pdf>

When submitting the revised version of your manuscript, please pay close attention to our [href="https://www.nature.com/nature-portfolio/editorial-policies/image-integrity">Digital Image Integrity Guidelines. and to the following points below:](https://www.nature.com/nature-portfolio/editorial-policies/image-integrity)

Please note, Extended Data figures and tables are online-only (appearing in the online PDF and full-text HTML version of the paper), peer-reviewed display items that provide essential background to the Article but are not included in the printed version of the paper due to space constraints or being of interest only to a few specialists. A maximum of ten Extended Data display items (figures and tables) is typically permitted. When re-submitting your manuscript, please ensure that any supplementary figures and tables that are more critical to the manuscript's conclusions are converted to Extended data to increase these data's visibility.

Link Redacted

We hope to receive your revised manuscript within 4-6 weeks. If you cannot send it within this time, please let us know. We will be happy to consider your revision so long as nothing similar has been accepted for publication at Nature Immunology or published elsewhere.

Nature Immunology is committed to improving transparency in authorship. As part of our efforts in this direction, we are now requesting that all authors identified as 'corresponding author' on published papers create and link their Open Researcher and Contributor Identifier (ORCID) with their account on the Manuscript Tracking System (MTS), prior to acceptance. ORCID helps the scientific community achieve unambiguous attribution of all scholarly contributions. You can create and link your ORCID from the home page of the MTS by clicking on 'Modify my Springer Nature account'. For more information please visit <http://www.springernature.com/orcid>.

Sincerely,

Ioana Staicu, Ph.D.
Senior Editor
Nature Immunology

Tel: 212-726-9207
Fax: 212-696-9752
www.nature.com/ni

Reviewer comments:

Reviewer #1

(Remarks to the Author)

The authors have adequately addressed all the concerns raised in the initial review. The additional experiments have significantly strengthened the manuscript, providing convincing evidence for the conclusions.

Reviewer #2

(Remarks to the Author)

The authors have conducted additional experiments in response to concerns raised by this reviewer and others. While the authors responses were reasonable and have clarified some of the reviewers' comments, others remained unaddressed or the limitation of the current study is revealed. Overall, this reviewer feels would like to support publishing this work as long as the manuscript accurately reflect both conclusions supported by experimental data AND limitations without convincing experimental demonstration. Please see specifics for each of the originally raised concerns as follows:

(1) The authors have revised the texts accordingly for the 1st half the comments. Regarding the second half, the data suggest that Zfp148 and Zfp281 are not redundant in the described context. Given the limited dynamics of Zfp148 expression, the authors should describe that it is highly likely that there is unknown factor(s) that cooperate with Zfp148 in cell-type specific manners.

(2) Please discuss the possibility that the absence of Zfp148 expression alters the differentiation of KLF2-negative lineage instead of changing expression of Klf2. While many studies proposed a linear progression of effector/intermediate to terminally exhausted CD8+ T cells, it is still controversial and alternative models suggesting effector/intermediate and terminally exhausted lineages are independent have not been excluded. Given that Figure 7h data shows only a minor change in KLF2 reporter expression, this also would be consistent with the latter model. Please show endogenous expression of KLF2 and ZFP148 in EL4 as well as expression of reporter lacking the KLF2 binding site instead of lacking KLF2 protein. This marginal difference of reporter expression change may result from weak expression of KLF4. These should be easy experiments which can be done within a week or two.

(3) to (6) These concerns have been adequately addressed.

Reviewer #3

(Remarks to the Author)

The authors have done an excellent job addressing all comments. I only have a few minor comments left:

- After re-reading the study, I would still suggest to remove Figure 1F-h, as it's been subsequently addressed more accurately. Moreover, the in vitro data also do not really align with the in vivo data (Extended Fig 3i,j) and therefore do not necessarily strengthen the study.
- Fig 2C: Rather than gating and demonstrating %GzmB+ tetramer+ cells, the authors could gate on Tex cells and show the gzmB MFI. The outcome should be the same, but it would represent a more straightforward illustration.
- The authors are showing "Numbers of cells/million splenocytes" (Ext Fig. 3). The authors need to show total number of cells, not per splenocytes. Moreover, the authors need to adjust their axis labeling as existing axis (eg Ext Fig 4) might have a mistake as ~5 million tetramer+ cells/"million splenocytes" would indicate ~5x10e12 cells/LN.
- What is the rationale for referring to increases in Ki67 as "accelerated proliferation" (lines 171)? It rather represents an increased proliferation, which is not sustained over time as Ki67 expression was similar on day 30 (Fig. 2d). Similarly, what is the Ki67 expression of Tpex or terminal exhausted T cells? If Ki67 expression is decreased in KO-Tpex cells, then one could argue that ZFP148 restricts proliferation but rather differentiation as KO cells show enhanced proliferation and formation of CXC3CR1+ cells, which is in line with the authors own conclusion: "These findings suggest that ZFP148 deletion primarily limits the differentiation of cells into Tex without altering the dysfunctional phenotype on a per-cell basis." (Line 189)

Version 4:

Decision Letter:

Our ref: NI-A39198D

21st Dec 2025

Dear Dr. Li,

Thank you for submitting your revised manuscript "ZFP148 is a novel transcriptional checkpoint for effector CD8+ T cell differentiation" (NI-A39198D). We are happy to inform you that if you revise your manuscript appropriately according to our editorial requirements, your manuscript should be publishable in Nature Immunology.

I will now pre-edit the current version of your paper. We will also perform detailed checks on your paper and will send you a checklist detailing our editorial and formatting requirements. Because of the holiday schedule please expect these in about three-four weeks. Please do not upload the final materials and make any revisions until you receive this additional information from us.

While waiting for the pre-edit check, please deposit all omic and code data into public repositories so that the accession codes are readily available to be added in the revised manuscript. We cannot accept the paper without the codes.

In addition, all corresponding authors need to update and link their ORCID to their Nature account. We cannot accept the

paper without this information. We suggest that you look into this while waiting for the pre-edited manuscript. Should you have any query or comments about ORCID, please do not hesitate to contact our editorial assistant at immunology@us.nature.com.

If you had not uploaded a Word file for the current version of the manuscript, we will need one before beginning the editing process; please email that to immunology@us.nature.com at your earliest convenience.

Thank you again for your interest in Nature Immunology. Please do not hesitate to contact me if you have any questions.

Sincerely,

Ioana Staicu, Ph.D.
Senior Editor
Nature Immunology

Tel: 212-726-9207
Fax: 212-696-9752
www.nature.com/ni

RE: NI-A39198C

"ZFP148 is a novel transcriptional checkpoint for effector CD8⁺ T cell differentiation"

November 18, 2025

Dear Ioana,

We sincerely thank you for the positive assessment and for considering our manuscript for publication in *Nature Immunology*. We also deeply appreciate the three reviewers for their thorough evaluation and constructive feedback. We are encouraged by their shared recognition that our study is “*interesting and novel*” (Reviewer #1), “*interesting and reasonably thorough*” (Reviewer #2), and “*well-executed and conceptually interesting*” (Reviewer #3). The reviewers collectively raised six key points regarding our original submission: (1) How Zinc-Finger Protein 148 (ZFP148) deletion influences viral control and the cytolytic capacity of CD8⁺ T cells? (2) How ZFP148 regulates differentiation of antigen-specific CD8⁺ T cells kinetically? (3) Whether ZFP281 functions as a cooperative or redundant cofactor of ZFP148? (4) What upstream pathways regulate ZFP148 expression? (5) What is the specific genomic regulatory element through which ZFP148 controls *Klf2*? (6) How ZFP148 deficiency synergizes with anti-PD-1 therapy? We have comprehensively addressed each of these points through substantial new experimentation and analyses.

Specifically, we measured serum viral titers at multiple time points after LCMV Cl13 infection and conducted cytotoxicity assays to assess the killing capacity of ZFP148-deficient CD8⁺ T cells. We performed an expanded chronic LCMV time course to evaluate differentiation kinetics, complemented by acute LCMV infection to define the effects of ZFP148 deficiency on effector differentiation. To investigate cofactor interactions, we applied CRISPR deletion of ZFP148, ZFP281, or both, thereby delineating their cooperative or redundant roles. We also quantified the kinetics of ZFP148 protein expression throughout the full course of chronic LCMV infection and used *in vitro* stimulation assays to identify the upstream signaling pathways regulating its expression. Importantly, we identified the mechanism through which ZFP148 represses *Klf2* by deleting the putative ZFP148-binding element in the distal *Klf2* locus, followed by rigorous functional validation. Finally, we conducted detailed phenotyping of ZFP148-deficient tumor-infiltrating CD8⁺ T cells to define how ZFP148 loss enhances responsiveness to anti-PD-1 therapy.

In total, we extensively revised two main figures (**Figs. 1 and 5**), generated one new main figure (**Fig. 2**), and added two new extended data figures (**Extended Data Figs. 3 and 4**). **We demonstrated conclusively that Zinc-Finger Protein 148 (ZFP148) is a crucial first transcription repressor discovered that limits the cytolytic effector differentiation of CD8⁺ T cells via repressing multiple transcriptional factors including *Klf2*.**

To address each of the Reviewers' comments in detail, we have prepared the following point-by-point response.

Reviewer #1 (Remarks to the Author):

In the manuscript titled “ZFP148 is a novel transcriptional checkpoint for effector CD8⁺ T cell differentiation”, the authors demonstrated ZFP148 as a novel negative regulator of effector CD8 T cell differentiation. They showed that ZFP148 KO had increased differentiation to the CX3CR1⁺ Teff subset and upregulated cytokines and granzyme B. ZFP148 KO also had

reduced T_{pro} suggesting a role of ZFP148 in inhibiting the differentiation from T_{pro} to T_{eff}. The authors further demonstrated that pro-T_{eff} transcription factor Klf2 as a direct target of ZFP148. ZFP148 KO mice are more sensitive to aPD1 induced tumor control than wild type mice. The study is interesting and novel. Below are my suggestions.

Response: We greatly appreciate the valuable and positive feedback on our work.

1. Please provide cell numbers in Fig 1h and 6c.

Response: We thank the reviewer for this suggestion. We have now included absolute cell numbers corresponding to the data shown in **Fig. 1h**, presenting Granzyme B (GZMB)⁺ TCF1⁻ and IFN- γ ⁺ TNF- α ⁺ CD8⁺ T cells from control and ZFP148 KO groups (Revised manuscript page 55, **Extended Data Fig. 1i**; **Response Letter Fig. 1a**). Likewise, absolute numbers of PD-1⁺ GZMB⁺ CD8⁺ tumor-infiltrating lymphocytes (TILs) are also provided (**Response Letter Fig. 1b**). For your convenience, these updated data are provided below.

Response letter Figure 1. a, Numbers of GZMB⁺ TCF1⁻ and IFN- γ ⁺ TNF- α ⁺ P14 CD8⁺ T cells per ml of culture medium 2 days after CRISPR-mediated ZFP148 knockout. **b**, Numbers of PD-1⁺ GZMB⁺ CD8⁺ T cells per mm² of tumor area from day 16 MC38 tumors in control or ZFP148 KO mice. The p-values in **a** were determined using two-sided paired-sample t-tests. The p-values in **b** were determined using two-sided unpaired-sample t-test. *p-value \leq 0.05, ** p-value \leq 0.01, *** p-value \leq 0.001, and **** p-value \leq 0.0001.

2. Please provide fold changes in Fig 2e-h. Although they are statistically significant, some differences are difficult to assess.

Response: We agree with this valuable suggestion and have performed a more comprehensive analysis of control and ZFP148-deficient splenic antigen-specific CD8⁺ T cells across multiple time points (days 8, 16, 22, and 30) following LCMV Cl13 infection. Based on these data, we generated a new main figure (Page 43, **Fig. 2**) and a new extended data figure (page 58, **Extended Data Fig. 3**). Fold-change values have been added to the corresponding bar plots to facilitate direct comparison between groups. In summary, our extended analyses revealed that ZFP148 deletion accelerates effector differentiation and enhances cytolytic capacity of antigen-specific CD8⁺ T cells throughout chronic LCMV infection, without compromising their persistence. For your convenience, the key data are also provided below.

Specifically, we observed a marked increase in T_{eff} cells from day 8 to day 30 (Page 43 and 58, **Fig. 2b** and **Extended Data Fig. 3d**; **Response Letter Fig. 2b,d**), accompanied by a corresponding decrease in T_{pro} cells beginning as early as day 8 post-infection and a reduction in T_{exh} cells on day 22 (page 58, **Extended Data Fig. 3e-h**; **Response Letter Fig. 2e-h**). This pattern indicates accelerated effector differentiation that initiates during the early phase of

chronic LCMV infection. The enhanced differentiation was primarily cytolytic, as evidenced by a higher proportion of GZMB⁺ TCF1⁻ CD8⁺ T cells (Page 43, Fig. 2c; Response Letter Fig. 2c).

Response letter Figure 2. a, Schematic for the chronic viral infection experiment by infecting E81-cre⁻ *Zfp148*^{fl/fl} (control) or E81-cre⁺ *Zfp148*^{fl/fl} (ZFP148 KO) mice with LCMV CI13 for 8, 16, 22, and 30 days. b, Left, scatter plots showing flow cytometric analysis of CX3CR1 vs. Ly108 expression in control and ZFP148 KO CD44^{high} GP₃₃₋₄₁ tetramer⁺ CD8⁺ T cells; right, bar plots showing relative frequency of subpopulations gated on left. c, Left, scatter plots showing flow cytometric analysis of GZMB vs. TCF-1 expression in control and ZFP148 KO CD44^{high} GP₃₃₋₄₁ tetramer⁺ CD8⁺ T cells; right, bar plots showing relative frequency of subpopulations as gated on the left. e, Bar plot showing number of CX3CR1⁺ Ly108⁻ T_{eff} subset in spleens from experimental groups described in a. f and g, Bar plots showing frequency (among CD44^{high} GP₃₃₋₄₁ tetramer⁺ CD8⁺ T cells, f) or number (g) of CX3CR1⁻ Ly108⁻ T_{pro} subset in spleens from experimental groups described in a. h and i, Bar plots showing frequency (among CD44^{high} GP₃₃₋₄₁ tetramer⁺ CD8⁺ T cells, h) or number (i) of CX3CR1⁻ Ly108⁻ T_{exh} subset in spleens from experimental groups described in a. The p-values in b-h were determined using multiple independent-sample t-tests. *p-value ≤ 0.05, ** p-value ≤ 0.01, *** p-value ≤ 0.001, and **** p-value ≤ 0.0001.

3. How does ZFP148 deficiency affect the virus titer in mice infected with LCMV CI13?

Response: We thank the reviewer for this question. To determine whether ZFP148 deficiency alters viral replication, we performed plaque assays to measure serum viral titers in control and CD8-specific ZFP148-deficient mice infected with LCMV CI13 for 8, 21, and 28 days. No significant differences in viral titers were observed between the two groups (Page 58, **Extended Data Fig. 3r; Response Letter Fig. 3**).

Viral control during chronic LCMV infection also depends on B cells, T follicular helper cells, and antibody responses (PMID: 27430722, 25680276, 35022243, 29196449). Consistent with many CD8-restricted conditional gene deletions that show minimal impact on systemic viral load under chronic infection (PMID: 29246443, 41145844, 27599295, 32374402), ZFP148 deletion in CD8⁺ T cells did not measurably affect serum viral titers in this setting.

Response letter Figure 3. The titers of virus in serum from control or ZFP148 KO mice infected with LCMV CI13 for 8, 21, and 28 days, determined by plaque assay. The p-values were determined using multiple independent-sample t-tests. ns, not significant.

4. They authors sorted CD44^{high} GP33-41⁺ CD8⁺ T cells for their scATAC+gene expression experiment. Yet, there is a naïve T cell population in Fig 3d.

Response: We appreciate this thoughtful feedback. Upon re-examining our single-cell data, we confirmed that this “naïve-like” cluster did not represent true naïve cells. Instead, these cells correspond to an early progenitor subset within the antigen-specific CD8⁺ T cell compartment. Using the expression profiles of key lineage-defining genes (*Cx3cr1*, *Gzmb*, *Pdcd1*, *Slamf6*, *Tcf7*, *Sell*, *Havcr2*, *Cd244a*, *Cd44*), we re-annotated this population as T_{pro1} rather than T_{naïve}, for the following reasons:

1. Post-sorting analysis showed >91% purity of CD44^{high} GP₃₃₋₄₁ tetramer⁺ CD8⁺ T cells and the residual tetramer⁻ CD8⁺ T cells were likewise CD44⁺ (Page 62, **Extended Data Fig. 5a**).
2. These cells expressed the activation marker *Cd44*, low levels of *Pdcd1*, and high levels of stemness- and memory-associated genes (*Sell*, *Tcf7*, and *Slamf6*), but lacked effector (*Cx3cr1*, *Gzmb*) and exhaustion (*Havcr2*, *Cd244a*) markers (Page 45, **Fig. 3b; Response Letter Fig. 4a, red circle**).

- They exhibited enrichment for the “progenitor exhausted” gene signature but not for “effector” or “terminal exhausted” signatures described by Miller *et al.*, *Nat. Immunol.*, 2019 (Page 45, **Fig. 3c**; **Response Letter Fig. 4b**, red circle).
- They closely resembled the previously reported CD62L⁺ T_{pro}/T_{pex} subset (PMID: 35978192, 40954251), which can give rise to CD62L⁻ T_{pro} cells during progressive differentiation.
- Consistent with our scRNA-seq data, this CD62L⁺ T_{pro}/T_{pex} subset was also detected by flow cytometry in samples collected 8–30 days post-LCMV CI13 infection (**Response Letter Fig. 4c**).

Based on these findings, we concluded that this cluster represents an early progenitor population (T_{pro1}), developmentally upstream of the previously defined T_{pro} (now renamed T_{pro2}) subset. Notably, although the total T_{pro} population decreased (Page 58, **Extended Data Fig. 3e,f**; **Response Letter Fig. 4d**), the ZFP148-deficient group showed an increased frequency of T_{pro1} cells within the total T_{pro} compartment at later infection time points (**Response Letter Fig. 4c**). Furthermore, frequency of CD62L⁺ subset was elevated not only in the T_{pro} subset but across all CD44^{High} GP₃₃₋₄₁ tetramer⁺ CD8⁺ T cells, accompanied by a concordant reduction in the frequency of CD69⁺ subset (Page 58, **Extended Data Fig. 3o**; **Response Letter Fig. 4e**). This pattern is consistent with enhanced KLF2 activity in ZFP148-deficient mice, as KLF2 promotes CD62L expression while repressing CD69 in T cells (PMID: 17548599, 16855590, 39946463, 40954251).

Collectively, these results clarify that the apparent “naïve” cluster reflects an early progenitor subset within the antigen-experienced CD44^{High} GP₃₃₋₄₁ compartment, rather than true naïve CD8⁺ T cells.

Response letter Figure 4. a, mRNA expression of selected genes projected onto the weighted nearest-neighbor (WNN) UMAP. **b**, Enrichment of gene signatures from a chronic LCMV dataset (Miller *et al.*, *Nat. Immunol.* 2019) projected onto the same UMAP. **c**, Flow-cytometric analysis of Ly108 versus CD62L in CX3CR1⁻ Ly108⁺ T_{pro} cells from control or ZFP148 KO mice at day 22 post-LCMV Cl13 infection (left) and corresponding quantification (right). **d**, Frequencies (left) and numbers (right) of CD44^{High} GP₃₃₋₄₁ tetramer⁺ CD8⁺ T cells in spleens of the indicated groups. **e**, Frequencies of CD62L⁺ and CD69⁺ subpopulations among total CD44^{High} GP₃₃₋₄₁ tetramer⁺ CD8⁺ T cells. The p-values in **c**, **d**, and **e** were determined using multiple independent-sample t-tests. ns, not significant, *p-value ≤ 0.05, ** p-value ≤ 0.01, *** p-value ≤ 0.001, and **** p-value ≤ 0.0001.

5. ZFP148 KO increased the percentage of proliferating cells but not the number of tetramer+ T cells. Does ZFP148 KO increase cell death?

Response: We thank the reviewer for this insightful question. To directly address this point, we performed additional experiments measuring apoptosis and cell death in antigen-specific CD8⁺ T cells following chronic LCMV infection. A modest increase in apoptotic (Annexin V⁺ PI⁻) cells was detected in the ZFP148 KO group at day 8 post-infection, whereas no significant differences were observed at later time points (days 16, 22, and 30) (Page 58, **Extended Data Fig. 3l,m; Response Letter Fig. 5a,b**, left). Minimal differences were also seen in the proportion of dead cells (Annexin V⁺ PI⁺) between the two groups (**Response Letter Fig. 5b**, right).

Since ZFP148 transcriptionally represses *Klf2*, and KLF2 expression has been shown to promote the circulation of CD8⁺ T cell (PMID: 34677611), we next assessed the frequency and absolute number of CD44^{High} GP₃₃₋₄₁ Tetramer⁺ CD8⁺ T cells in the blood. Both parameters were increased in ZFP148 KO mice (Page 58, **Extended Data Fig. 3n; Response Letter Fig. 5c**), suggesting potentially enhanced egress of ZFP148-deficient CD8⁺ T cells from lymphoid organs into the circulation. Consistent with this, ZFP148 KO cells exhibited elevated CD62L but reduced CD69 expression as mentioned in our response to comment #4, indicative of diminished tissue retention and a greater propensity to recirculate between the blood and lymphoid tissues (PMID: 11018170, 15032576, 17429841, 31139190, 32396847, 24162775). This trafficking pattern also mirrors the behavior of CX3CR1⁺ Ly108⁻ effector-like CD8⁺ T cells, which are highly circulating and intravascular (PMID: 11018170, 15032576, 17429841, 31139190, 31810882, 32034098, 40954251). Consequently, the total number of gp33-specific CD8⁺ T cells in the spleen remained largely unchanged (day 8, 16, and 30) or even decreased (day 22) (Page 58, **Extended Data Fig. 3c; Response Letter Fig. 5d**).

Response letter Figure 5. **a**, Flow-cytometric analysis of Annexin V versus propidium iodide (PI) in CD44^{High} GP₃₃₋₄₁ tetramer⁺ CD8⁺ T cells from control or ZFP148 KO mice at day 8 post-LCMV CI13 infection. **b**, Frequencies of Annexin V⁺ PI⁻ and Annexin V⁺ PI⁺ subsets from control and ZFP148 KO mice at days 8, 16, 22, and 30 post-infection. **c**, Frequencies (among total CD8⁺ T cells) and numbers (per 10⁶ splenocytes) of CD44^{High} GP₃₃₋₄₁ tetramer⁺ CD8⁺ T cells in peripheral blood from control or ZFP148 KO mice at days 0, 8, 16, and 21 post-infection. Fold change in **b** and **c** calculated as mean(Zfp148 KO) / mean(control). The p-values in **b** and **d** were determined using multiple independent-sample t-tests. The p-values in **c** were determined using two-sided unpaired-sample t-test. ns, not significant, *p-value ≤ 0.05, ** p-value ≤ 0.01, *** p-value ≤ 0.001, and **** p-value ≤ 0.0001.

6. T_{pro} is the key driver of the response to aPD1. However, the authors showed that ZFP148 KO reduced T_{pro} but increased the sensitivity to aPD1. Some discussion would be highly appreciated.

Response: We appreciate this insightful comment. We interpret the enhanced sensitivity to anti-PD-1 immunotherapy in ZFP148-deficient mice as resulting from two complementary mechanisms: (1) preserved frequency of progenitor-like (T_{pro}/T_{pex}) CD8⁺ tumor-infiltrating lymphocytes (CD8⁺ TILs) despite ZFP148 deletion, contrasting with their reduction in chronic LCMV infection; and (2) augmented differentiation of these T_{pro} cells into cytolytic effectors (T_{eff}) upon PD-1 blockade.

As expected, ZFP148 deficiency reduced the T_{pro} frequency in chronic LCMV (Page 58, **Extended Data Fig. 3e,f**), whereas in MC38 tumors it remained largely unchanged (significant only at day 10; Page 66, **Extended Data Fig. 7c**). Instead, ZFP148-deficient CD8⁺ TILs showed a modest but consistent increase in GZMB⁺ TCF1⁻ T_{eff} cells (Page 66, **Extended Data Fig. 7b**)—smaller than in LCMV (fold-change ZFP148 KO/control: D10 ×1.17, D18 ×1.18 vs. up to ×2.1 in LCMV; Page 43, **Fig. 2c**). These CD8⁺ TILs also expressed higher levels of PD-1, TIM-3, LAG-3, TIGIT, and CD39 (Page 66, **Extended Data Fig. 7d**), indicating coexisting cytolytic activation and exhaustion.

Differences between the two models likely reflect their distinct antigenic and immunological contexts. Chronic LCMV provides a uniform, high antigen load (PMID: 19433785, 27455951), whereas tumors feature low, heterogeneous antigen presentation due to MHC-I downregulation (PMID: 29107330, 27433843, 29070816). Since induction of ZFP148 expression depends on TCR signal strength (Page 41, **Fig. 1d**), these differences likely modulate its expression and regulatory role. Even in chronic infection, ZFP148 deletion did not improve viral clearance (Page 58, **Extended Data Fig. 3r**), suggesting that additional immune components—such as B cells, T follicular helper cells, and antibody responses (PMID: 27430722, 25680276, 35022243, 29196449)—contribute to viral control.

Before PD-1 blockade, ZFP148 deficiency modestly enhanced CD8⁺ TIL cytolytic activity but not tumor control, likely due to limited magnitude and immune suppression. Anti-PD-1 treatment increased GZMB and perforin expression (Page 51, **Fig. 6j,k**), Ki-67⁺ proliferating cells (Page 51, **Fig. 6l**), and CD8⁺ TIL infiltration (Page 66, **Extended Data Fig. 7e,f**), consistent with known anti-PD-1 ICB mechanisms (PMID: 22186141, 16382236, 12218188, 28397821, 28446615, 39121847). In ZFP148-deficient mice, PD-1 blockade further amplified T_{eff} differentiation and upregulated ICOS, CD27, and CD25 (Page 51, **Fig. 6i, bottom**; Page 66, **Extended Data Fig. 7i,j**), indicating strengthened IL-2 signaling, survival, and effector function (PMID: 18178815, 14615582, 19955658, 22781761, 20096608, 23352221, 20732639). Collectively, these data demonstrate that ZFP148 deficiency primes CD8⁺ TILs with greater cytolytic potential, resulting in a more robust anti-PD-1 response despite modest T_{pro} reduction.

To illustrate these points, we added additional contents in the last paragraph of the discussion section (**line 599-607**).

7. Information of the anti-perforin antibody needs to be included.

Response: We thank the reviewer for the suggestion. The information for the anti-perforin antibody (Vendor: BioLegend; Cat#: 154310; Clone: S16009A; Fluorophore: FITC) has been added in the Methods section of the revised manuscript.

Reviewer #2 (Remarks to the Author):

Persistent antigen exposure gives rise to dysfunctional “exhausted” CD8 T cell populations that exhibit loss of cytotoxicity and proliferative potential. These cells are derived from a stem-like pool of progenitor cells (TPEX), however, in instances of chronic infection or cancer, the efficient de novo generation of cytolytic effector populations from T_{pex} is outpaced by viral replication or tumor growth, respectively. Thus, there is great interest in determining how CD8 T cell responses may be safeguarded from exhaustion.

In this manuscript, Xiao and colleagues identify ZFP148 as a novel regulator of cytotoxic CD8 T cell differentiation. Although it is unclear how they identified Zfp148 as the main subject of this study, the major findings of this study are:

- 1) During acute and chronic infection, Zfp148 is expressed in antigen-specific CD8⁺ T cells. The authors show that there is marginally lower expression in cells responding to chronic LCMV infection compared to acute LCMV infection. Specifically, there was higher expression in acute effector and resting memory states compared to those responding to chronic LCMV infection as determined by flow and scRNA sequencing. Zfp148 is almost uniformly expressed in all subsets with a slightly higher level in Ly108⁺ TCF-1⁺ GZMB⁻ TPEX cells.
- 2) In CD8 T cell-specific Zfp148 cKO mice, higher frequencies of antigen-specific CD8⁺ T cells exhibit a CX3CR1⁺ effector state while Ly108⁺ TPEX and Ly108⁻ CX3CR1⁻ TEX cells are

reduced both in frequencies and numbers, confirmed by both flow and scRNA-seq. This indicates that absence of Zfp148 corresponds with greater effector development.

3) ATAC-seq results were consistent with RNA-seq, with more accessibility at loci encoding genes more highly expressed in CX3CR1⁺ effectors, such as Zeb2 and S1pr2, in Zfp148-KO CD8 T cells. Increased accessibility in the potential cis-acting elements containing T-bet consensus motifs in KO TPEX cells suggest a potential overlapping function with FLI-1 or BACH2 that restrict mobilization of TPEX into effectors.

4) As a potential mechanism, the authors proposed that Zfp148 represses KLF2 based on increased KLF2 expression in the proliferating subset accompanied by increased accessibility of KLF2 motifs in the same subsets.

5) Finally, the findings were extended to CD8⁺ T cells responding to tumors both in human and mouse, showing enhanced anti-tumor immunity with anti-PD1 in Zfp148-cko mice as well as correlation to pronounced T cell responses with colon and gastric cancer patients with Zfp148-low tumors.

Overall, the authors findings with phenotypes in mice and correlation to human patients are interesting and this is the first report showing the contribution of this transcription factors in CD8⁺ T cell response. Analyses of mouse phenotypes were conducted reasonably thoroughly and clinical data showing inverse correlation between Zfp148 and patients' prognosis.

Response: We are grateful for the reviewer's constructive feedback and comprehensive summary of our study. We apologize for the lack of clarity regarding how ZFP148 was initially identified. In the Introduction of the original manuscript, we stated that "ZFP148 was previously found through computational inference as a potential transcriptional regulator in CD8⁺ T cells during both chronic viral infection and cancer (PMID: 35420889, 31606264). However, its role in regulating the differentiation and functionality of CD8⁺ T cells in these disease settings remains completely unknown." To improve clarity for readers, we have now moved this statement to the beginning of the Results section (**line 109-110**).

However, this reviewer sees a few major concerns with regards to the mechanisms, which must be clarified with additional experiments as follows:

1) Although the authors repeatedly stated that Zfp148 expression is the highest in TPEX or resting T cells, its change in expression levels is very small even though it is unnecessarily exaggerated in Fig 1a with an inappropriate Y-axis scaling. The protein levels look the same, and KO phenotypes in cultured T cells show statistically significant but biologically questionable changes in effector molecule expression, suggesting that most of the in vivo phenotypes resulted from increased frequencies in effector cells rather than increased effector function on a per-cell basis. These are not major issues as long as the authors revise the text to reflect real findings with unnecessary exaggeration. However, the findings the authors have now suggest that the presence of co-operating factors whose expression changes more dramatically between TPEX and effector cells may account for the differences. This has to be clearly solved. An obvious candidate is Zfp281, which is partially redundant with Zfp148 in T cells as demonstrated by Bosselut et al. but the authors conclusion that its high expression in TPEX suppress their differentiation may not be correct.

Response: We thank the reviewer for this constructive feedback. The Y-axis in **Fig. 1a** (Page 41) has been rescaled, and we agree that the magnitude of change in *Zfp148* mRNA is modest between acute and chronic LCMV infection. Our intention was not to present this panel as a large fold-change, but rather to show that *Zfp148* transcription is modestly but consistently lower in virus-specific CD8⁺ T cells during chronic (Clone 13) infection compared with the acute Armstrong model across all time points.

We next assessed ZFP148 protein expression, which showed a more substantial dynamic pattern during chronic infection. Compared with naïve CD44⁻ CD62L⁺ CD8⁺ T cells, gp33-specific CD8⁺ T cells showed an ~1.3-fold increase in ZFP148 protein during the effector phase (days 0–8) of Clone 13 infection (Page 41, **Fig. 1b**; **Response Letter Fig. 6b**), suggesting an early role for ZFP148 during priming. Likewise, naïve CD8⁺ T cells rapidly upregulated ZFP148 protein upon in vitro TCR stimulation, in a manner dependent primarily on anti-CD3 dose and only minimally on anti-CD28 costimulation (Page 41, **Fig. 1c,d**; **Response Letter Fig. 6c,d**). Inhibition of the calcium–calcineurin–NFAT pathway with cyclosporin A blocked this induction in a dose-dependent manner (Page 55, **Extended Data Fig. 1b**; **Response Letter Fig. 6e**), confirming the involvement of canonical TCR–NFAT signaling.

Following the effector phase of chronic infection, ZFP148 protein levels then declined by ~50% between days 8 and 16 and remained stable through day 30 (Page 41, **Fig. 1b**; **Response Letter Fig. 6b**), indicating that ZFP148 protein levels decrease as CD8⁺ T cells differentiate over time.

As suggested, we have revised the text to more accurately reflect these findings. Specifically, we changed “*Zfp148* expression was markedly reduced...” to “*Zfp148* expression was reduced...” (**line 113**). We also softened “marked expansion of cytolytic effector subsets” to “expansion of cytolytic effector subsets” to avoid overinterpretation (**line 139**).

We agree with the reviewer on the importance of examining ZFP281, a partially redundant zinc-finger transcription factor. We found that ZFP281 does not exert an equivalent regulatory role to ZFP148 in effector differentiation during chronic LCMV infection. On day 21 post-infection, fold changes of ZFP281 protein between T_{pro} and T_{eff} (×1.12) or T_{exh} (×1.05) subsets were smaller than those of ZFP148 (T_{pro}/T_{eff} ×1.19; T_{pro}/T_{exh} ×1.20) (**Response Letter Fig. 6f**). To assess its kinetics, we quantified ZFP281 peptides by mass spectrometry across antigen-specific CD8⁺ T-cell subsets from both acute and chronic LCMV infections (PMID: 41034580). ZFP281 expression trended higher in progenitor-like (T_{pro}) and memory-like (MPEC) subsets but did not reach statistical significance (**Response Letter Fig. 6g**).

To test potential redundancy, we performed CRISPR-RNP-mediated knockout of ZFP148, ZFP281, or both in P14 CD8⁺ T cells, followed by adoptive transfer into C57BL/6J recipients and LCMV CI13 infection (**Response Letter Fig. 6h**). On day 21 post-infection, transferred cells from spleens were analyzed by flow cytometry. Knockout efficiencies were reduced in the double-KO condition (ZFP148: 78.5% vs. 56.3%; ZFP281: 72.6% vs. 26.4%), likely due to technical limitations or partial redundancy (**Response Letter Fig. 6i,j**). Neither ZFP281 single KO nor double KO resulted in enhanced effector differentiation or increased GZMB⁺ TCF1⁻ or GZMA⁺ GZMB⁺ T_{eff} frequencies (**Response Letter Fig. 6k**). Collectively, these findings indicate that ZFP148 plays a dominant role over ZFP281 in restricting the differentiation of antigen-specific CD8⁺ T cells into cytolytic effectors during chronic LCMV infection. For your convenience, these data are also provided below.

Response letter Figure 6. **a**, mRNA expression of *Zfp148* in adoptively transferred P14 CD8⁺ T cells from C57BL/6 mice infected with LCMV Armstrong or CI13 for 0, 8, 15, and 30 days, re-analyzed from Giles *et al.*, *Nat. Immunol.* 2022. **b**, Mean fluorescence intensity (MFI) of ZFP148 in CD44^{High} GP₃₃₋₄₁ tetramer⁺ CD8⁺ T cells from spleens of LCMV CI13-infected mice at days 8, 16, 22, and 30. **c**, Left, ZFP148 protein expression in naïve CD8⁺ T cells stimulated with 5 $\mu\text{g ml}^{-1}$ anti-CD3 with or without 2 $\mu\text{g ml}^{-1}$ anti-CD28 for 0–48 h; right, corresponding MFI over time. **d**,

Left, ZFP148 protein expression in naïve CD8⁺ T cells stimulated with increasing concentrations of anti-CD3 (plus 2 µg ml⁻¹ anti-CD28) for 48 h; right, MFI versus anti-CD3 concentration. **e**, Left, ZFP148 expression in naïve CD8⁺ T cells stimulated with anti-CD3 + anti-CD28 for 48 h in the presence of increasing cyclosporin A (CsA) concentrations; right, MFI versus CsA concentration. **f**, MFI of ZFP148 (left) or ZFP281 (right) in CD44^{high} PD-1^{high} CD8⁺ T cells from spleens at day 21 post-LCMV CI13 infection. **g**, Proteomic quantification of ZFP281 in antigen-specific CD8⁺ T-cell subsets from LCMV Armstrong- or CI13-infected mice at days 8 and 30 (PMID: 41034580). **h**, Schematic of CRISPR knockout experiment in which activated P14 CD8⁺ T cells received control or *Zfp148*, *Zfp281*, or double-targeting sgRNAs, followed by adoptive transfer and LCMV CI13 infection. **i**, ZFP148 (left) and ZFP281 (right) expression in edited P14 CD8⁺ T cells described in **h**. **j**, Corresponding MFI of ZFP148 (left) and ZFP281 (right). **k**, Frequencies of indicated subpopulations among transferred P14 CD8⁺ T cells from spleens at day 21 post-infection. The p-values in **b**, **f**, **g**, and **k** were determined using Tukey's Honestly Significant Difference (HSD) test. The p-values in **c** were determined using a linear mixed effects model. *p-value ≤ 0.05, ** p-value ≤ 0.01, *** p-value ≤ 0.001, and **** p-value ≤ 0.0001.

2) The weakest and potentially problematic part of the paper is their attempt to mechanically connect to KLF2. As stated above, it is very likely that the enrichment of *Klf2* gene activity in Fig. 5a reflects changes in KLF2-hi population rather than higher KLF2 gene activity on a per cell basis in *Zfp148*-KO cells. The authors seem to assume that *Zfp148* binds to the +10 kb region of the *Klf2* locus to enhance its expression, but this has to be directly tested by ablation of the specific region rather than ablation of *Klf2* by RNP electroporation. Since *Klf2* ablation is sufficient to reduce CX3CR1⁺ cells in Fig 5i-k, the DKO phenotype in Fig. 5i is uninterpretable and the functional connection is very questionable.

Response: We appreciate this insightful comment and have addressed it accordingly. Using CRISPR–RNP–mediated genomic element deletion and a luciferase reporter assay, we directly validated that ZFP148 represses *Klf2* transcription through a distal regulatory element bound by ZFP148, thereby establishing a direct mechanistic connection between these two factors. More details are described below.

As the reviewer noted, *Klf2* mRNA expression was highest in the T_{eff} subset (Page 49, **Fig. 5b**; **Response Letter Fig. 7a**). Importantly, we also detected higher *Klf2* mRNA levels across all antigen-specific CD8⁺ T-cell subsets in the absence of ZFP148 (Page 49, **Fig. 5b**; **Response Letter Fig. 7a**), indicating that *Klf2* is upregulated along the entire CD8⁺ T cell differentiation trajectory, including in the T_{pro} populations. Given the well-established role of KLF2 in promoting effector differentiation (PMID: 39946463, 38012417, 40954251), we hypothesized that KLF2 acts as a functional driver rather than a passive correlate of the augmented cytolytic differentiation observed in ZFP148-deficient CD8⁺ T cells.

To directly test this, we employed a CRISPR–RNP approach to delete the ZFP148-bound distal regulatory element within the *Klf2* locus in P14 CD8⁺ T cells. Edited or control cells were adoptively transferred into C57BL/6J recipients followed by LCMV CI13 infection (**Response Letter Fig. 7b**). Genomic editing was confirmed by gel electrophoresis (**Response Letter Fig. 7c**) and validated by Inference of CRISPR Edits (ICE) analysis of sanger sequencing data (https://ice.editco.bio/#/analyze/results/hwcfnd1ar12xhbf/Sample_122_59) (**Response Letter Fig. 7d**). Following distal regulatory element deletion, we observed a significant increase in both *Klf2* mRNA and KLF2–EGFP fusion protein compared to non-edited controls (Page 49, **Fig. 5j,k**; **Response Letter Fig. 7e,f**). On day 21 post-infection, flow cytometric analysis revealed that distal regulatory element-deleted cells exhibited a higher frequency of CX3CR1⁺ Ly108⁻ and GZMB⁺ TCF1⁻ T_{eff} cells relative to cells receiving non-targeting sgRNA (Page 49, **Fig. 5l**; **Response Letter Fig. 7g**), closely recapitulating the phenotype of ZFP148-deficient CD8⁺ T

cells. These results demonstrate that the distal regulatory element directly mediates ZFP148-dependent repression of *Klf2*.

As a complementary approach, we cloned the putative ZFP148-bound distal regulatory element, identified by CUT&Tag-seq (Page 49, **Fig. 5g**), upstream of the SV40 minimal promoter in a pGL3 luciferase vector (designated pGL3-*Klf2*-distal). Dual-luciferase reporter assays were then performed in control and ZFP148-deficient EL4 cells. Notably, deletion of ZFP148 led to a significant increase in luciferase activity (Page 49, **Fig. 5m**; **Response Letter Fig. 7h**), confirming that ZFP148 functions as a transcriptional repressor through this distal regulatory element of the *Klf2* locus. For your convenience, these key data are also provided below.

Response letter Figure 7. a, Violin plots showing *Klf2* mRNA expression in control or ZFP148 KO cells by subset. **b**, Schematic of the CRISPR knockout experiment in which activated P14 CD8⁺ T cells were transduced with non-targeting control sgRNA or sgRNAs targeting *Zfp148* or the distal regulatory element, followed by adoptive transfer and LCMV C13 infection. **c**, Agarose gel electrophoresis of PCR products spanning the distal regulatory region from CRISPR-edited

P14 CD8⁺ T cells using three primer sets. **d**, Indel efficiency, deletion efficiency, and deletion size at the distal regulatory site determined by Inference of CRISPR Edits (ICE) analysis of Sanger sequencing data (see: https://ice.editco.bio/#/analyze/results/hwcfnr1ar12xhbf/Sample_122_59). **e**, Relative *Klf2* mRNA levels in P14 CD8⁺ T cells receiving control, distal element-targeting, or *Klf2*-targeting sgRNAs. **f**, KLF2-EGFP fusion protein expression in CRISPR-edited P14 CD8⁺ T cells as in **e**. **g**, Frequencies of CX3CR1⁺ Ly108⁻ and GZMB⁺ TCF1⁻ subsets among transferred P14 CD8⁺ T cells from spleens 21 days post-infection. **h**, Relative luciferase activity in control or ZFP148 KO EL4 cells transfected with the pGL3 reporter vector containing the distal regulatory element, normalized to controls. The p-values in **a** were determined using Wilcoxon rank sum test. The p-values in **e** were determined using Tukey's HSD test. The p-values in **g** and **h** were determined using two-sided independent-sample t-test. *p-value ≤ 0.05, ** p-value ≤ 0.01, *** p-value ≤ 0.001, and **** p-value ≤ 0.0001.

3) In Fig.3f, the authors show predicted differentiation trajectories of CD8 T cells in clone 13 infection. While these types of analyses often show discrepancies from more biological readouts, how real are the results from this analysis with effector transit to proliferating population or naïve cells to TPEX? Are the naïve cells endogenous GP33-specific CD8 T cells for which such high frequencies are unrealistic?

Response: We appreciate this helpful comment. The population initially annotated as “T_{naïve}” in **Fig. 3d** does not represent true naïve cells but rather a subset of antigen-experienced cells that exhibit early progenitor-like features. Using the expression profiles of key lineage-defining genes (*Cx3cr1*, *Gzmb*, *Pdcd1*, *Slamf6*, *Tcf7*, *Sell*, *Havcr2*, *Cd244a*, *Cd44*), we re-annotated this population as T_{pro1} rather than T_{naïve}, for the following reasons:

1. Post-sorting analysis showed >91% purity of CD44^{high} GP₃₃₋₄₁ tetramer⁺ CD8⁺ T cells and the residual tetramer⁻ CD8⁺ T cells were likewise CD44⁺ (Page 62, **Extended Data Fig. 5a**).
2. These cells express the activation marker *Cd44*, low levels of *Pdcd1*, and high levels of stemness- and memory-associated genes (*Sell*, *Tcf7*, and *Slamf6*), but lacked effector (*Cx3cr1*, *Gzmb*) and exhaustion (*Havcr2*, *Cd244a*) markers (Page 45, **Fig. 3b**; **Response Letter Fig. 8a, red circle**).
3. They exhibit enrichment for the “progenitor exhausted” gene signature but not for “effector” or “terminal exhausted” signatures described by Miller *et al.*, *Nat. Immunol.*, 2019 (Page 45, **Fig. 3c**; **Response Letter Fig. 8b, red circle**).
4. They closely resemble the previously reported CD62L⁺ T_{pro}/T_{pex} subset (PMID: 35978192, 40954251), which can give rise to CD62L⁻ T_{pro} cells during progressive differentiation.
5. Consistent with our scRNA-seq data, this CD62L⁺ T_{pro}/T_{pex} subset was also detected by flow cytometry in samples collected 8–30 days post-LCMV CI13 infection (**Response Letter Fig. 8c**).

Based on these findings, we concluded that this cluster represents an early progenitor population (T_{pro1}), developmentally upstream of the previously defined T_{pro} (now renamed T_{pro2}) subset. Notably, although the total T_{pro} population decreased (Page 58, **Extended Data Fig. 3e,f**; **Response Letter Fig. 8d**), the ZFP148-deficient group showed an increased frequency of T_{pro1} cells within the total T_{pro} compartment at later infection time points (**Response Letter Fig. 8c**). Furthermore, frequency of CD62L⁺ subset was elevated not only in the T_{pro} subset but across all CD44^{high} GP₃₃₋₄₁ tetramer⁺ CD8⁺ T cells, accompanied by a concordant reduction in the frequency of CD69⁺ subset (Page 58, **Extended Data Fig. 3o**; **Response Letter Fig. 8e**). This pattern is consistent with enhanced KLF2 activity in ZFP148-deficient mice, as KLF2

promotes CD62L expression while repressing CD69 in T cells (PMID: 17548599, 16855590, 39946463, 40954251).

Regarding the “proliferating” cluster, these cells expressed *Mki67* along with effector genes (*Cx3cr1*, *Gzmb*) and the “effector” gene signature, suggesting they represent a proliferative subset within the effector pool rather than a distinct lineage (Page 45, **Fig. 3c**; **Response Letter Fig. 8a,b, blue circle**). However, as the single-cell RNA-seq dataset captures a single snapshot (day 21 post-infection) rather than a full temporal trajectory, we acknowledge that pseudotime reconstruction provides inferential rather than definitive evidence of lineage transitions.

Response letter Figure 8. **a**, mRNA expression of selected genes projected onto the weighted nearest-neighbor (WNN) UMAP. **b**, Enrichment of gene signatures from a chronic LCMV dataset (Miller et al., Nat. Immunol. 2019) projected onto the same UMAP. **c**, Flow-cytometric analysis of Ly108 versus CD62L in CX3CR1⁻ Ly108⁺ T_{pro} cells from control or ZFP148 KO mice at day 22 post-*LCMV* CI13 infection (left) and corresponding quantification (right). **d**, Frequencies (left) and numbers (right) of CD44^{high} GP₃₃₋₄₁ tetramer⁺ CD8⁺ T cells in spleens of the indicated groups. **e**, Frequencies of CD62L⁺ and CD69⁺ subpopulations among total CD44^{high} GP₃₃₋₄₁ tetramer⁺ CD8⁺ T cells. The p-values in **c**, **d**, and **e** were determined using multiple independent-sample t-tests. ns, not significant, *p-value ≤ 0.05, ** p-value ≤ 0.01, *** p-value ≤ 0.001, and **** p-value ≤ 0.0001.

4) Do Zfp148 cKO mice show enhanced viral control in LCMV-clone 13 infection or superior lysis activity ex vivo?

Response: We thank the reviewer for this question. To determine whether ZFP148 deletion enhances viral control, we performed plaque assays to quantify serum viral titers from control

and ZFP148-deficient mice infected with LCMV CI13 for 8, 21, and 30 days. No significant difference in viral load was observed between the two groups (Page 58, **Extended Data Fig. 3r; Response Letter Fig. 9a**). Given that viral clearance during chronic LCMV infection also depends on B cell, T follicular helper cells, and antibody responses (PMID: 27430722, 25680276, 35022243, 29196449), changes in CD8⁺ T cell effector function alone may not necessarily translate into measurable differences in systemic viremia. This is also consistent with many CD8-restricted conditional gene deletions that show minimal impact on systemic viral load under chronic infection (PMID: 29246443, 41145844, 27599295, 32374402).

To assess cytolytic function directly, we performed complementary *ex vivo* and *in vitro* killing assays. gp33- and gp276-specific CD8⁺ T cells were FACS-sorted from spleens of control and ZFP148 KO mice on day 21 post-infection and co-cultured with B16-GP target cells that overexpress the LCMV glycoprotein (GP). Annexin V–based live-cell imaging (Incucyte) revealed that ZFP148-deficient GP-specific CD8⁺ T cells induced significantly greater apoptosis in target cells compared with controls (Page 43, **Fig. 2g,h; Response Letter Fig. 9b,c**). Similarly, when control or ZFP148 KO P14 CD8⁺ T cells were co-cultured with B16-GP targets, the knockout cells exhibited consistently superior cytolytic capacity (Page 43, **Fig. 2i; Response Letter Fig. 9d,e**).

Together, these results demonstrate that ZFP148-deficient antigen-specific CD8⁺ T cells possess enhanced lytic activity *ex vivo* and *in vitro*, although this increased cytotoxicity did not translate into improved viral clearance *in vivo*.

Response letter Figure 9. **a**, Viral titers in serum from control or ZFP148 KO mice infected with LCMV CI13 for 8, 21, and 28 days, determined by plaque assay. **b**, Schematic of the cytotoxicity assay in which CD44^{high} GP_{33–41} and GP_{276–286} tetramer⁺ CD8⁺ T cells, sorted from spleens of control or ZFP148 KO mice 22 days post-infection, were co-cultured with B16-GP target cells. **c**, Kinetics of Annexin V⁺ signal in B16-GP target cells co-cultured with control or ZFP148 KO antigen-specific CD8⁺ T cells as in **b**. **d**, Schematic of the cytotoxicity assay using control or ZFP148 KO P14 CD8⁺ T cells co-cultured with B16-GP cells. **e**, Kinetics of Annexin V⁺ signal in B16-GP target cells co-cultured with control or ZFP148 KO P14 CD8⁺ T cells. The p-values in **a** were determined using multiple independent-sample t-tests. The p-values in **c** and **e** were

determined using a linear mixed effects model. ns, not significant, *p-value \leq 0.05, ** p-value \leq 0.01, *** p-value \leq 0.001, and **** p-value \leq 0.0001.

5) Does Zfp148 conditional deletion impact memory CD8 T cell (TCM, TEM, TRM) differentiation following acute infection?

Response: We appreciate this insightful question. Yes, conditional deletion of ZFP148 impairs the generation and maintenance of memory CD8⁺ T cells, resulting in reduced overall memory cellularity, preferential differentiation into central memory (TCM) over effector memory (TEM) cells, and diminished formation of tissue-resident memory (TRM) cells. Details and key data are also provided below.

To address this, we infected control and ZFP148 KO mice with LCMV Armstrong to induce acute viral infection (Page 60, **Extended Data Fig. 4a; Response Letter Fig. 10a**). During the effector phase (day 9 p.i.), ZFP148-deficient gp33-specific CD8⁺ T cells exhibited a skewed differentiation toward KLRG1⁺ CD127⁻ short-lived effector cells (SLECs) with a concomitant reduction in KLRG1⁻ CD127⁺ memory precursors (MPECs) in both spleens and inguinal lymph nodes (iLNs) (Page 60, **Extended Data Fig. 4c–e; Response Letter Fig. 10b–d**). Within the MPEC population, the absence of ZFP148 promoted differentiation toward CD62L⁺ central memory (TCM) rather than CD62L⁻ effector memory (TEM) cells, as shown by increased TCM and decreased TEM frequencies in iLNs (**Response Letter Fig. 10e,f**).

Response letter Figure 10. a, Schematic of the acute viral infection experiment in which control or ZFP148 KO mice were infected with LCMV Armstrong for 9 days. **b**, Flow-cytometric analysis of KLRG1 versus CD127 expression in CD44^{high} GP₃₃₋₄₁ tetramer⁺ CD8⁺ T cells from inguinal lymph nodes (iLNs; left) and spleens (right). **c**, Frequencies (left) and numbers (right) of KLRG1⁺ CD127⁻ subsets among CD44^{high} GP₃₃₋₄₁ tetramer⁺ CD8⁺ T cells in iLNs and spleens. **d**, Frequencies (left) and numbers (right) of KLRG1⁻ CD127⁺ subsets among CD44^{high} GP₃₃₋₄₁ tetramer⁺ CD8⁺ T cells in iLNs and spleens. **e**, Frequencies of CD62L⁻ effector memory (TEM; left) and CD62L⁺ central memory (TCM; right) subsets among KLRG1⁻ CD127⁺ MPECs in iLNs and spleens. **f**, Numbers of CD62L⁻ TEM (left) and CD62L⁺ TCM (right) subsets in iLNs and spleens from groups described in a. The p-values in c-f were determined using multiple independent-sample t tests. *p-value ≤ 0.05, ** p-value ≤ 0.01, *** p-value ≤ 0.001, and **** p-value ≤ 0.0001.

By the memory phase (day 37 p.i.), ZFP148 KO mice displayed reduced numbers of total CD44^{High} GP₃₃₋₄₁ tetramer⁺ CD8⁺ T cells across multiple tissues (small intestine (Intraepithelial Lymphocyte, IEL), liver, and spleen); **Response Letter Fig. 11a**), consistent with impaired memory maintenance. Within this reduced pool, significantly decreased frequencies of MPECs were observed in the IEL and iLNs while decreased number in the IEL and spleens (**Response Letter Fig. 11b**). Further, though the frequencies of both TCM and TEM did not change, their absolute numbers significantly decreased in the spleens from the ZFP148 KO group (**Response Letter Fig. 11c,d**), indicating that loss of ZFP148 compromises the maintenance of both subsets. Moreover, tissue-resident memory CD8⁺ T cells (TRMs) were diminished, as reflected by lower frequencies and numbers of CD69⁺ CD103⁺ TRMs in IEL and CD69⁺ CD103⁻ TRMs in iLNs, lungs, and spleens (**Response Letter Fig. 11e,f**).

In summary, ZFP148 is required for the proper establishment and long-term maintenance of the memory CD8⁺ T cell compartment. Its deletion results in reduced total memory T cell cellularity, preferential bias toward TCM over TEM differentiation, and defective formation of TRM populations in both lymphoid and non-lymphoid tissues.

Response letter Figure 11. **a**, Frequencies (left) and numbers (right) of CD44^{High} GP₃₃₋₄₁ tetramer⁺ CD8⁺ T cells in blood, small intestine (intraepithelial lymphocytes, IEL), inguinal lymph nodes (iLNs), liver, lung, and spleen from control or ZFP148 KO mice 37 days after LCMV Armstrong infection. **b**, Frequencies (left) and numbers (right) of KLRG1⁻ CD127⁺ subsets among CD44^{High} GP₃₃₋₄₁ tetramer⁺ CD8⁺ T cells in tissues described in **a**. **c**, Frequencies of CD62L⁻ effector memory (TEM; left) and CD62L⁺ central memory (TCM; right) subsets among KLRG1⁻ CD127⁺ MPECs in tissues described in **a**. **d**, Numbers of CD62L⁻ TEM (left) and CD62L⁺ TCM (right) subsets in tissues described in **a**. **e**, Frequencies (left) and numbers (right) of CD69⁺ CD103⁺ subsets among CD44^{High} GP₃₃₋₄₁ tetramer⁺ CD8⁺ T cells in tissues described in **a**. **f**, Frequencies (left) and numbers (right) of CD69⁺ CD103⁻ subsets among CD44^{High} GP₃₃₋₄₁ tetramer⁺ CD8⁺ T cells in tissues described in **a**. The p-values in **a-f** were determined using

multiple independent-sample t tests. *p-value \leq 0.05, ** p-value \leq 0.01, *** p-value \leq 0.001, and **** p-value \leq 0.0001.

6) In Fig. 6 (and also relevant to ICB responsiveness in patients), the data show enhanced response to anti-PD1 in the MC38 in mice. Given that there is more effector differentiation in the LCMV model without PD1 blockade, it is surprising that there is no difference in anti-tumor immunity without anti-PD1 (closed circle versus closed square cohort). Since TPEX cells have been thought to be the major responding cells to anti-PD1 and TPEX cells are reduced in Zfp148-cKO mice, how the authors interpret this phenotype?

Response: We thank the reviewer for this point. We interpret the enhanced sensitivity to anti-PD-1 immunotherapy in ZFP148-deficient mice as resulting from two complementary mechanisms: (1) preserved frequency of progenitor-like ($T_{\text{pro}}/T_{\text{pex}}$) $CD8^+$ tumor-infiltrating lymphocytes ($CD8^+$ TILs) despite ZFP148 deletion, contrasting with their reduction in chronic LCMV infection; and (2) augmented differentiation of these T_{pro} cells into cytolytic effectors (T_{eff}) upon PD-1 blockade.

To clarify these points, we analyzed $CD8^+$ T cell responses in both LCMV CI13 and MC38 models. As expected, ZFP148 deficiency reduced the T_{pro} frequency in chronic LCMV (Page 58, **Extended Data Fig. 3e,f**), whereas in MC38 tumors it remained largely unchanged (significant only at day 10; Page 66, **Extended Data Fig. 7c**). Instead, ZFP148-deficient $CD8^+$ TILs displayed a modest but consistent increase in $GZMB^+ TCF1^- T_{\text{eff}}$ cells (Page 66, **Extended Data Fig. 7b**), although to a smaller extent than in LCMV (fold-change ZFP148 KO/control: D10 $\times 1.17$, D18 $\times 1.18$ vs up to $\times 2.1$ in LCMV; Page 43, **Fig. 2c**). ZFP148-deficient $CD8^+$ TILs also expressed higher levels of inhibitory receptors (PD-1, TIM-3, LAG-3, TIGIT, and CD39; Page 66, **Extended Data Fig. 7d**), suggesting concurrent enhancement of cytolytic activity and exhaustion.

Differences between tumor and chronic infection likely reflect distinct antigenic and immunological contexts. Chronic LCMV infection features uniform, high antigen load (PMID: 19433785, 27455951), whereas tumor antigens are heterogeneous and often poorly presented due to MHC-I downregulation (PMID: 29107330, 27433843, 29070816). Since induction of ZFP148 expression depends on TCR signal strength as mentioned in the response to comment #1 (Page 41, **Fig. 1d**; **Response Letter Fig. 6d**), these differences likely modulate its expression and function. Even in chronic infection, ZFP148 deletion did not improve viral clearance (Page 58, **Extended Data Fig. 3r**), indicating contributions from other immune components such as B cells, T follicular helper cells, and antibody responses (PMID: 27430722, 25680276, 35022243, 29196449). Together, these results suggest that ZFP148 loss alone is insufficient to enhance baseline tumor immunity.

Before PD-1 blockade, ZFP148 deficiency increased the cytolytic potential of $CD8^+$ TILs and slightly raised effector frequency, but these changes were too small to improve tumor control amid persistent immune suppression. Anti-PD-1 treatment elevated GZMB and perforin expression (Page 51, **Fig. 6j,k**), Ki-67⁺ proliferating cells (Page 51, **Fig. 6l**), and $CD8^+$ TIL infiltration (Page 66, **Extended Data Fig. 7e,f**), consistent with known anti-PD-1 ICB mechanisms (PMID: 39121847). In ZFP148-deficient mice, PD-1 blockade further amplified T_{eff} differentiation and upregulated ICOS, CD27, and CD25—markers of enhanced IL-2 signaling, survival, and effector function (PMID: 10542150, 27387997, 34193916, 11070168, 22057290, 22861099, 20601952, 21419664) (Page 51, **Fig. 6i, bottom**; Page 66, **Extended Data Fig. 7i**,

j). Collectively, these findings demonstrate a synergistic interaction between ZFP148 loss and PD-1 blockade in driving superior CD8⁺ TIL effector responses and anti-tumor activity.

Reviewer #3 (Remarks to the Author):

The manuscript by Xiao et al. presents a comprehensive analysis of the transcription factor ZFP148 in the context of T cell exhaustion, utilizing both chronic LCMV infection and tumor models. The authors first examine ZFP148 expression using existing RNAseq data, validated via flow cytometry. By leveraging a CD8⁺ T cell-specific ZFP148 knockout model, they demonstrate enhanced effector differentiation, supported by flow cytometry and single-cell RNA sequencing. Integrative single-cell ATAC and RNA sequencing further elucidate the epigenetic landscape shaped by ZFP148, identifying KLF2 as a putative downstream target. This is functionally validated through CRISPR-RNP-mediated deletion of KLF2 and/or ZFP148. Finally, the authors show that ZFP148 deficiency improves responsiveness to immune checkpoint blockade in the MC38 tumor model and correlate these findings with human transcriptomic data.

Overall, this is a well-executed and conceptually interesting study that characterizes a previously underappreciated transcription factor and its role in restraining effector differentiation in exhausted T cells. The work is likely to be of interest to both researchers and clinician scientists. That said, a few issues should be addressed to further strengthen the manuscript and ensure clarity and rigor of the conclusions.

Response: We are deeply grateful for the reviewer's insightful and supportive comments on our work.

Major Comments:

- Further characterization in the LCMV model is warranted. The authors should extend their flow cytometric analysis of ZFP148 expression across different CD8⁺ T cell subsets throughout the course of chronic LCMV infection. This will help contextualize its dynamic regulation and functional relevance. Additionally, the data in Fig. 2 would benefit from further characterization to substantiate claims of enhanced effector differentiation and quality. This should include: cytokine production analyses following restim (e.g., IFN- γ , TNF, IL-2) and additional assessment of progenitor/effector markers such as CD62L, CD69, c-KIT and CD101. Critically, the authors need to extend these analyses to multiple time points, including later stages post-day 21. This would further help to identify if the ability to continuously sustain a T cell response is impaired. The authors should also perform functional assays, such as cytotoxicity (killing assays), to directly assess effector capabilities in the absence of ZFP148. The in vitro activation data (Fig. 2i-k) provide limited insight into the in vivo kinetics of differentiation. These could be replaced or supplemented with analyses of effector formation during the expansion phase of acute vs. chronic infection to more directly address the notion of "accelerated effector differentiation."

Response: We thank the reviewer for these constructive suggestions. To address these points, we extended our longitudinal flow-cytometric analysis of antigen-specific CD8⁺ T cells from control and ZFP148-deficient mice at 8, 16, 22, and 30 days after LCMV CI13 infection. These results are now included in the revised **Fig. 2** (Page 43) and **Extended Data Fig. 3** (Page 58). In summary, our expanded analyses revealed that ZFP148 deletion accelerates effector

differentiation and enhances cytolytic capacity of antigen-specific CD8⁺ T cells throughout chronic LCMV infection, without compromising their persistence.

Specifically, we observed a marked decrease in the frequency of T_{pro} cells beginning as early as day 8 post-infection (Page 58, **Extended Data Fig. 3e,f**), accompanied by a corresponding increase in T_{eff} cells from day 8 to day 30 (Page 43, **Fig. 2b**). This pattern indicates accelerated effector differentiation that initiates during the early phase of chronic LCMV infection. Despite this shift, overall gp33-specific CD8⁺ T cell numbers remained stable through day 30 (except for a transient decrease on day 22) (Page 58, **Extended Data Fig. 3b,c**), demonstrating that the response was sustained. The enhanced differentiation was primarily cytolytic, as evidenced by a higher proportion of GZMB⁺ TCF1⁻ CD8⁺ T cells (Page 43, **Fig. 2c**), rather than increased pro-inflammatory cytokines (IFN- γ , TNF- α , and IL-2) production (Page 58, **Extended Data Fig. 3i,j**).

To functionally validate these findings, we assessed cytolytic activity *ex vivo*. gp33- and gp276-specific CD8⁺ T cells were FACS-sorted from spleens of control and ZFP148 KO mice on day 21 post-LCMV CI13 infection and co-cultured with B16-GP target cells expressing the LCMV glycoprotein (GP). Live-cell Annexin V assays (Incucyte) revealed significantly higher apoptosis in B16-GP cells (Annexin V⁺) induced by ZFP148-deficient CD8⁺ T cells compared with controls (Page 43, **Fig. 2g,h**; **Response Letter Fig. 12a,b**). Likewise, ZFP148 KO P14 CD8⁺ T cells displayed superior killing capacity when co-cultured with B16-GP targets *in vitro* (Page 43, **Fig. 2i**; **Response Letter Fig. 12c,d**).

We next examined differentiation and exhaustion-associated surface markers. ZFP148-deficient CD8⁺ T cells exhibited increased frequencies of CD62L⁺ and decreased CD69⁺ cells at later stages (days 22 and 30) (Page 58, **Extended Data Fig. 3o**), consistent with enhanced KLF2 activity, as KLF2 promotes CD62L while repressing CD69 expression in CD8⁺ T cells (PMID: 17548599, 16855590, 39946463, 40954251). TCF1⁺ progenitor populations (TIM-3⁻ TCF1⁺ or GZMB⁻ TCF1⁺ cells) were reduced in ZFP148 KO samples (**Response Letter Fig. 12e**), further supporting a bias toward effector differentiation. The expression of c-KIT—a marker progressively lost during T cell maturation and associated with less-differentiated, apoptosis-prone cells (PMID: 17659849, 23073628, 12415312, 30930902)—was modestly lower in ZFP148-deficient antigen-specific CD8⁺ T cells on days 8 and 22 (**Response Letter Fig. 12f**), consistent with a more differentiated phenotype.

For exhaustion-associated molecules, we assessed PD-1, TIM-3, LAG-3, CTLA-4, CD39, and CD101 (Page 43, **Fig. 2f**). On day 22, PD-1 and TIM-3 were elevated, whereas LAG-3 was reduced, in ZFP148-deficient gp33-specific CD8⁺ T cells (Page 43, **Fig. 2f**, top row) and within the T_{exh} subset (Page 58, **Extended Data Fig. 3p**, top row). By day 30, almost no differences in their expression were detected (Page 43, **Fig. 2f**, bottom row; Page 58, **Extended Data Fig. 3p**, bottom row). Collectively, these findings suggest that ZFP148 primarily regulates differentiation fate rather than the degree of exhaustion in the chronic LCMV setting.

Finally, to parallel these analyses in an acute infection model, we infected control and ZFP148 KO mice with LCMV Armstrong. On day 9 post-infection, gp33-specific CD8⁺ T cells from spleen and inguinal lymph nodes (iLNs) showed a significant increase in KLRG1⁺ CD127⁻ short-lived effector cells (SLECs) and a corresponding decrease in KLRG1⁻ CD127⁺ memory precursors (MPECs) in ZFP148 KO mice (Page 60, **Extended Data Fig. 4c–e**). These cells also expressed higher levels of GZMA and GZMB (Page 60, **Extended Data Fig. 4f**), whereas IFN- γ ⁺ TNF- α ⁺ cytokine-producing cells were slightly reduced or unchanged (Page 60, **Extended Data Fig. 4g**).

Together, these data indicate that ZFP148 deficiency accelerates effector differentiation and enhances cytolytic function in both chronic and acute LCMV infection settings, while preserving the overall magnitude and longevity of the CD8⁺ T cell response.

Response letter Figure 12. **a**, Schematic of the cytotoxicity assay in which CD44^{high} GP₃₃₋₄₁ and GP₂₇₆₋₂₈₆ tetramer⁺ CD8⁺ T cells, sorted from spleens of control or ZFP148 KO mice 22 days post-LCMV CI13 infection, were co-cultured with B16-GP target cells. **b**, Kinetics of Annexin V⁺ signal in B16-GP cells co-cultured with control or ZFP148 KO antigen-specific CD8⁺ T cells as in **a**. **c**, Schematic of the cytotoxicity assay using control or ZFP148 KO P14 CD8⁺ T cells co-cultured with B16-GP cells. **d**, Kinetics of Annexin V⁺ signal in B16-GP cells co-cultured with control or ZFP148 KO P14 CD8⁺ T cells. **e**, Frequencies of TIM-3⁻ TCF1⁺ (left) and GZMB⁻ TCF1⁺ (right) subsets among CD44^{high} GP₃₃₋₄₁ tetramer⁺ CD8⁺ T cells from spleens of control or ZFP148 KO mice infected with LCMV CI13 for 8, 16, 22, and 30 days. **f**, Mean fluorescence intensity (MFI) of c-Kit in CD44^{high} GP₃₃₋₄₁ tetramer⁺ CD8⁺ T cells from the groups described in **e**. The p-values in **b** and **d** were determined using a linear mixed effects model. The p-values in **e** and **f** were determined using multiple independent-sample t-tests. ns, not significant, *p-value ≤ 0.05, ** p-value ≤ 0.01, *** p-value ≤ 0.001, and **** p-value ≤ 0.0001.

- The viral burden and infection control in CD8-specific ZFP148-deficient mice must be addressed. All LCMV data derive from E8i-Cre–driven ZFP148-deficient mice. It is essential that the authors report viral titers to exclude the possibility that observed transcriptional or phenotypic changes are secondary to altered viral control, which could question all the multiomic analyses.

Response: We thank the reviewer for the insight. To directly assess viral burden, we performed plaque assays to quantify serum viral titers from control and CD8-specific ZFP148-deficient mice infected with LCMV CI13 for 8, 21, and 28 days. No significant difference in viral load was observed between the two groups (Page 58, **Extended Data Fig. 3r**; **Response Letter Fig. 13**).

Since viral control during chronic LCMV infection also involves B cells, T follicular helper cells, and antibody responses (PMID: 27430722, 25680276, 35022243, 29196449), changes in CD8⁺ T cell effector function alone may not necessarily translate into measurable differences in systemic viremia. This is also consistent with many CD8-restricted conditional gene deletions that show minimal impact on systemic viral load under chronic infection (PMID: 29246443, 41145844, 27599295, 32374402).

Response letter Figure 13. The titers of virus in serum from control or ZFP148 KO mice infected with LCMV CI13 for 8, 21, and 28 days, determined by plaque assay. The p-values were determined using multiple independent-sample t-tests. ns, not significant.

- The authors report increased PD-1⁺Gzmb⁺ TILs in ZFP148-deficient tumors (Fig. 6C), yet Fig. 6J suggests similar T cell numbers between genotypes. This discrepancy requires clarification. Furthermore, enhanced characterization of the anti-tumor response (e.g., tumor-infiltrating T cell function, exhaustion marker expression, cytokine production) would add depth to the tumor-related conclusions. The inclusion of an additional tumor model (e.g., B16, At3) would also enhance the robustness of the findings.

Response: We appreciate this valuable observation. **Fig. 6j** is intended to show the synergistic effect between ZFP148 deletion and anti-PD-1 ICB in promoting the cytolytic effector function of CD8⁺ TILs, indicated by highest frequency of CX3CR1⁺ GZMB⁺ and Perforin⁺ GZMB⁺ subpopulations in the combination group. If comparing only within isotype-treated groups, ZFP148-deficient mice still showed enhanced effector function, we can still observe enhanced effector function in the ZFP148 KO mice showcased by higher frequency of CX3CR1⁺ GZMB⁺ and Perforin⁺ GZMB⁺ subpopulations (Page 51, **Fig. 6j,k**, closed circle versus closed square cohort; **Response Letter Fig. 14a**).

We further performed comprehensive flow-cytometric analyses to characterize the anti-tumor CD8⁺ T cell response. On days 10–18 post-MC38 tumor inoculation, both the frequency and total number of CD8⁺ TILs were comparable between genotypes (**Response Letter Fig. 14b,c**). Higher frequency of GZMB⁺ TCF1⁺ T_{eff} cells was observed in the ZFP148-deficiency group (Page 66, **Extended Data Fig. 7b**; **Response Letter Fig. 14d**). However, there was minimal difference in the frequency of T_{pro} cells among total CD8⁺ T cells infiltrating MC38 tumors between control and ZFP148 KO groups (only significant on day 10) (Page 66, **Extended Data**

Fig. 7c; Response Letter Fig. 14e). Functionally, ZFP148-deficient CD8⁺ TILs produced similar amounts of IFN- γ and TNF- α , reduced IL-2, but elevated levels of cytolytic molecules including GZMB and Perforin (**Response Letter Fig. 14f**). Regarding exhaustion markers, PD-1, TIM-3, LAG-3, TIGIT, and CD39 were all upregulated in ZFP148-deficient TILs relative to controls (Page 51 and 66, **Fig. 6b and Extended Data Fig. 7d; Response Letter Fig. 14g**). This pattern differs from the chronic LCMV setting, where ZFP148 deletion enhanced cytolytic differentiation without concurrent upregulation of inhibitory receptors (please refer to our response to comment #1). Together, these findings indicate that ZFP148 deficiency promotes cytolytic effector differentiation of CD8⁺ TILs, albeit with features of T cell exhaustion within the tumor microenvironment.

As the reviewer suggested, we further tested an additional tumor model, E0771 breast carcinoma. ZFP148-deficient mice showed delayed tumor progression compared with control counterparts (**Response Letter Fig. 14h**), suggesting augmented anti-tumor capacity of ZFP148 KO CD8⁺ T cells. For your convenience, the key data are also provided below.

Response letter Figure 14. **a**, Frequencies of CX3CR1⁺ GZMB⁺ (left) and Perforin⁺ GZMB⁺ (right) subsets among total CD8⁺ T cells infiltrating day 13 MC38 tumors in control or ZFP148 KO mice treated with isotype control antibodies. **b**, Frequencies of CD8⁺ T cells among total live CD45⁺ immune cells infiltrating day 10, 14, 16, and 18 MC38 tumors in control or ZFP148 KO mice. **c**, Numbers of CD8⁺ tumor-infiltrating lymphocytes (TILs) normalized to tumor weight (days 10 and 18, left) or tumor area (days 14 and 16, right). **d**, Frequencies of GZMB⁺ TCF1⁻ subsets, and **e**, frequencies of GZMB⁻ TCF1⁺ subsets among total CD8⁺ T cells infiltrating day 10, 14, 16, and 18 MC38 tumors in control or ZFP148 KO mice. **f**, Frequencies of IFN- γ ⁺ TNF- α ⁺, IL-2⁺, GZMB⁺, and Perforin⁺ subsets among total CD8⁺ TILs from the groups described in **a** following *ex vivo* restimulation. **g**, Mean fluorescence intensity (MFI) of individual effector proteins in CD8⁺ T cells infiltrating day 16 MC38 tumors in control or ZFP148 KO mice. **h**, Growth curves of E0771 tumors subcutaneously injected into control or ZFP148 KO mice; data represent mean tumor area (mm²) \pm s.e.m., $n = 6-10$ mice per group. The p-values in **a**, **f**, and **g** were determined using two-sided independent-sample t-test. The p-values in **b**, **c** and **d** were determined using multiple independent-sample t tests. The p-values in **h** were determined using a linear mixed effects model. *p-value ≤ 0.05 , ** p-value ≤ 0.01 , *** p-value ≤ 0.001 , and **** p-value ≤ 0.0001 .

- Finally, additional mechanistic insights in the regulation of ZFP148 would be highly beneficial and significantly strengthen the impact of the study.

Response: This is an excellent suggestion and has been implemented. To further clarify the regulation of ZFP148, we examined its expression dynamics and upstream signaling in CD8⁺ T cells. Flow cytometry revealed that ZFP148 protein was upregulated in CD44^{High} GP₃₃₋₄₁ tetramer⁺ CD8⁺ T cells during the effector phase (days 0–8), declined by ~50% between days 8–16, and stabilized through day 30 (Page 41, **Fig. 1b**; **Response Letter Fig. 15a**). A similar trend was observed across T_{pro}, T_{eff}, and T_{exh} subsets, with the highest expression in T_{pro} cells (**Response Letter Fig. 15b**), suggesting a key role for ZFP148 during T cell priming and early differentiation.

In vitro, ZFP148 expression was rapidly induced in naïve splenic CD8⁺ T cells stimulated with anti-CD3 and anti-CD28 antibodies, but was primarily dependent on TCR signaling (anti-CD3) rather than costimulation (anti-CD28) in a dose-dependent manner (Page 41, **Fig. 1c,d**; **Response Letter Fig. 15c,d**). Inhibition of the calcium–calcineurin–NFAT pathway using increasing concentrations of cyclosporin A (CsA) progressively reduced ZFP148 upregulation under anti-CD3 stimulation (Page 55, **Extended Data Fig. 1b**; **Response Letter Fig. 15e**), confirming that calcium–calcineurin–NFAT signaling is required for ZFP148 induction by TCR activation.

In acute LCMV infection, ZFP148 levels were comparable among naïve (CD44⁻CD62L⁺), total antigen-specific (CD44^{High} GP₃₃₋₄₁ tetramer⁺), short-lived effector (KLRG1⁺CD127⁻), and memory precursor (KLRG1⁻CD127⁺) CD8⁺ T cells at day 9 post-infection (**Response Letter Fig. 15f**). In the tumor setting, ZFP148 expression was highest in CX3CR1⁻ TCF1⁺ progenitor-like cells compared with CX3CR1⁺ TCF1⁻ effector and CX3CR1⁻ TCF1⁻ exhausted subsets (Page 55, **Extended Data Fig. 1e**; **Response Letter Fig. 15g**), consistent with the chronic LCMV pattern.

In summary, ZFP148 is induced through TCR-dependent calcium–calcineurin–NFAT signaling and is transiently upregulated during early CD8⁺ T cell differentiation. Its expression peaks during the effector phase and declines as cells further differentiate or undergo prolonged antigen stimulation, underscoring a dynamic mode of ZFP148 regulation during chronic stimulation. For your convenience, these updated data are also provided below.

Response letter Figure 15. **a**, Mean fluorescence intensity (MFI) of ZFP148 in CD44^{High} GP₃₃₋₄₁ tetramer⁺ CD8⁺ T cells from spleens of C57BL/6 mice infected with LCMV Cl13 for 8, 16, 22, and 30 days. **b**, MFI of ZFP148 in T_{pro}, T_{eff}, and T_{exh} subsets among CD44^{High} GP₃₃₋₄₁ tetramer⁺ CD8⁺ T cells from groups described in **a**. **c**, Left, ZFP148 protein expression in naïve CD8⁺ T cells stimulated with 5 µg ml⁻¹ anti-CD3 with or without 2 µg ml⁻¹ anti-CD28 for 0, 10, 24, and 48 h; right, corresponding MFI over time. **d**, Left, ZFP148 protein expression in naïve CD8⁺ T cells stimulated with increasing concentrations of anti-CD3 (plus 2 µg ml⁻¹ anti-CD28) for 48 h; right, MFI versus anti-CD3 concentration. **e**, Left, ZFP148 protein expression in naïve CD8⁺ T cells stimulated with anti-CD3 and anti-CD28 for 48 h in the presence of increasing concentrations of cyclosporin A (CsA); right, MFI versus CsA concentration. **f**, MFI of ZFP148 in subpopulations of CD44^{High} GP₃₃₋₄₁ tetramer⁺ CD8⁺ T cells from spleens of C57BL/6 mice infected with LCMV Armstrong for 9 days. **g**, MFI of ZFP148 in subpopulations of CD8⁺ T cells infiltrating day 14 MC38 tumors. The p-values in **a**, **b**, **f** and **g** were determined using Tukey's Honestly Significant Difference (HSD) test. The p-values in **c** were determined using a linear mixed effects model. *p-value ≤ 0.05, ** p-value ≤ 0.01, *** p-value ≤ 0.001, and **** p-value ≤ 0.0001.

Minor comments/suggestions:

- The authors may consider removing Fig. 1G-I and the corresponding text (lines 147-159), as

the functional analysis using knockout mice renders this overexpression experiment redundant and potentially confusing at this point in the manuscript.

Response: We thank the reviewer for this helpful suggestion. We apologize for the lack of clarity in the original submission. In **Fig. 1g-i** in the original submission (Page 41, **Fig. 1f-h** in the revised manuscript), we used CRISPR–RNP–mediated deletion of ZFP148 in activated P14 CD8⁺ T cells and compared the production of effector molecules between control and ZFP148-deficient conditions. Acute ZFP148 deletion promoted the expansion of cytolytic effector subsets expressing GZMB, GZMA, perforin, and IFN- γ (Page 41, **Fig. 1g,h**; Page 55, **Extended Data Fig. 1i-k**). These findings indicate that ZFP148 restrains effector differentiation of activated CD8⁺ T cells *in vitro*, complementing the *in vivo* knockout data.

- Fig. 2F, the use of a log scale would better visualize differences in subset frequencies.

Response: We have updated **Extended Data Fig. 3c, 3d, 3f, and 3h** (Page 58) using a log₁₀ scale to improve visualization of differences in cell numbers across groups. For your convenience, the key data are also provided below.

Response letter Figure 16. **a**, Number of CD44^{High} GP₃₃₋₄₁ tetramer⁺ CD8⁺ T cells in the spleens from control or ZFP148 KO mice infected with LCMV CI13 for 8, 16, 22, and 30 days. **b**, Numbers of CX3CR1⁺ Ly108⁻ T_{eff}, CX3CR1⁻ Ly108⁺ T_{pro}, or CX3CR1⁻ Ly108⁻ T_{exh} subsets in spleens from experimental groups described in **a**. The p-values in **a** and **b** were determined using multiple independent-sample t-tests. *p-value ≤ 0.05, ** p-value ≤ 0.01, *** p-value ≤ 0.001, and **** p-value ≤ 0.0001.

- Line 201 should reference Extended Data Fig. 1a (not Fig. 1a).

Response: We thank the reviewer for this helpful suggestion. Upon revisiting line 201 in the original submission, we confirmed that this section describes the *in vitro* exhaustion assay using CD8⁺ T cells from control and ZFP148 KO mice, which should indeed reference **Extended Data Fig. 1c** rather than **Fig. 1c**. However, as noted in comment #1, this dataset has now been replaced with new results generated from the acute LCMV infection model (Page 60, **new Extended Data Fig. 4**). We have also carefully reviewed the entire manuscript to ensure that all figure references are accurate.

1 RE: NI-A39198D

2 "ZFP148 is a novel transcriptional checkpoint for effector CD8⁺ T cell differentiation"

3 In this revised version of the manuscript, we have added several new datasets and made
4 targeted revisions in response to the referees' comments. Newly generated data include: (i)
5 KLF2 gene signature enrichment in every subset of CD44^{High} GP₃₃₋₄₁ tetramer⁺ CD8⁺ T cells
6 (**revised manuscript page 60, new data, Extended Data Fig. 6e**); (ii) flow cytometric
7 validation of endogenous ZFP148 and KLF2 protein expression in EL4 cells and confirmation of
8 efficient ZFP148 deletion following CRISPR knockout (**page 60, new data, Extended Data Fig.**
9 **6m,n**); (iii) revised quantification of absolute numbers of antigen-specific CD8⁺ T cells and
10 subsets with corrected axis labeling (**page 54, new data, Extended Data Fig. 3c,d,f,h; page**
11 **56, new data, Extended Data Fig. 4b,d,e**); and (iv) additional analysis of Ki67 expression in
12 progenitor and terminally exhausted CD8⁺ T cell subsets across multiple time points (**page 54,**
13 **new data, Extended Data Fig. 3l**). In addition, we have removed **existing data Fig. 1f-h** and
14 **Extended Data Fig. 1h-k** in response to Reviewer #3's first comment (**page 39 and page 52,**
15 **respectively**). All other referenced figures represent existing data, unless explicitly noted.

16

17 Reviewer #1 (Remarks to the Author):

18

19 The authors have adequately addressed all the concerns raised in the initial review. The
20 additional experiments have significantly strengthened the manuscript, providing convincing
21 evidence for the conclusions.

22 **Response:** We greatly appreciate the valuable and positive feedback on our work.

23

24

25 Reviewer #2 (Remarks to the Author):

26

27 The authors have conducted additional experiments in response to concerns raised by this
28 reviewer and others. While the authors responses were reasonable and have clarified some of
29 the reviewers' comments, others remained unaddressed or the limitation of the current study is
30 revealed. Overall, this reviewer feels would like to support publishing this work as long as the
31 manuscript accurately reflect both conclusions supported by experimental data AND limitations
32 without convincing experimental demonstration. Please see specifics for each of the originally
33 raised concerns as follows:

34

35 (1) The authors have revised the texts accordingly for the 1st half the comments. Regarding the
36 second half, the data suggest that Zfp148 and Zfp281 are not redundant in the described
37 context. Given the limited dynamics of Zfp148 expression, the authors should describe that it is
38 highly likely that there is unknown factor(s) that cooperate with Zfp148 in cell-type specific
39 manners.

40 **Response:** We thank the reviewer for this suggestion and have revised the Discussion
41 accordingly. The manuscript now notes that, "given its broad regulatory capacity and restricted
42 expression dynamics across CD8⁺ T cell subsets, ZFP148 is highly likely to function in
43 cooperation with additional, yet-to-be-defined cofactors in a cell type-specific manner" (**revised**
44 **manuscript, line 496-499**).

45

46 (2) Please discuss the possibility that the absence of Zfp148 expression alters the differentiation
47 of KLF2-negative lineage instead of changing expression of Klf2. While many studies proposed
48 a linear progression of effector/intermediate to terminally exhausted CD8⁺ T cells, it is still

49 controversial and alternative models suggesting effector/intermediate and terminally exhausted
50 lineages are independent have not been excluded. Given that Figure 7h data shows only a
51 minor change in KLF2 reporter expression, this also would be consistent with the latter model.
52 Please show endogenous expression of KLF2 and ZFP148 in EL4 as well as expression of
53 reporter lacking the KLF2 binding site instead of lacking KLF2 protein. This marginal difference
54 of reporter expression change may result from weak expression of KLF4. These should be easy
55 experiments which can be done within a week or two.

56 (3) to (6) These concerns have been adequately addressed.

57 **Response:** We appreciate this thoughtful feedback and have now incorporated additional
58 discussion on the possibility that ZFP148 contributes to the maintenance or formation of KLF2-
59 negative lineages (progenitor or terminally exhausted populations), which are diminished in the
60 ZFP148-deficient setting. We have also performed new experiments to measure the expression
61 of endogenous ZFP148 and KLF2 in EL4 cells. ZFP148 was expressed at higher levels than in
62 activated CD8⁺ T cells and was efficiently deleted by CRISPR KO, supporting EL4 as a suitable
63 system for testing ZFP148-dependent repression (**revised manuscript, page 60, new data,**
64 **Extended Data Fig. 6m,n; Response Letter Fig. 1a,b**). KLF2 was also expressed at higher
65 level in EL4 cells (**page 60, new data, Extended Data Fig. 6m; Response Letter Fig. 1a**).
66 Since the reporter specifically measures ZFP148-mediated repression, endogenous KLF2 levels
67 should not influence the assay.

68 **Response Letter Figure 1. a**, ZFP148 and KLF2 protein expression in EL4 cells and activated CD8⁺ T cells
69 (stimulated with 3µg/ml anti-CD3 and 1µg/ml anti-CD28 for 3 days). **b**, ZFP148 protein expression in control
70 or ZFP148 KO EL4 cells.
71

72 Regarding the reviewer's request to examine "expression of reporter lacking the KLF2 binding
73 site instead of lacking KLF2 protein", we interpret this comment as referring to a reporter lacking
74 the ZFP148 binding site rather than deletion of ZFP148 protein, and we apologize for any
75 ambiguity in the original wording. We agree that motif-level dissection of ZFP148 binding within
76 this element would provide additional insight on alternative mechanism(s) and represent an
77 important direction for future studies beyond the scope of the present work.

78 In the current study, we have provided the following key evidence supporting a direct
79 transcriptional repressive role of ZFP148 on *Klf2*.

- 80 1. ZFP148 loss resulted in a coordinated elevation of *Klf2* mRNA, KLF2 motif accessibility,
81 and KLF2-regulated gene signatures across all CD8⁺ T cell subsets—including
82 progenitors—**revealing enhanced KLF2 transcriptional activity from the early**
83 **stages of differentiation (page 46, existing data, Fig. 5b,e and page 60, new data,**
84 **Extended Data Fig. 6e; Response Letter Fig. 2a)**

85
86
87
88
89
90
91
92

2. CUT&Tag-seq revealed robust ZFP148 binding at a distal regulatory element ~10.9 kb downstream of *Klf2*, which became more accessible in ZFP148-deficient cells (page 46, existing data, Fig. 5g; Response Letter Fig. 2b).
3. ZFP148 KO increased CX3CR1⁺ Ly108⁻ effector cells, whereas KLF2 deletion—alone or combined with ZFP148 deletion—abolished this subset, demonstrating that KLF2 is required for the effector-skewing phenotype of ZFP148 loss (page 46, existing data, Fig. 5h,i; Response Letter Fig. 2c).

93
94
95
96
97
98
99
100

4. Most critically, CRISPR deletion of the ZFP148-bound distal element increased KLF2 expression and expanded CX3CR1⁺ Ly108⁻ and GZMB⁺ TCF1⁻ effector cells, phenocopying ZFP148 KO despite intact ZFP148 protein (page 46, existing data, Fig. 5c,d,j–l; Response Letter Fig. 2d-f).
5. A *Klf2* distal-element luciferase reporter further showed higher activity in ZFP148-deficient EL4 cells, supporting ZFP148-dependent trans-repression (page 46, existing data, Fig. 5m; Response Letter Fig. 2g).

Response Letter Figure 2. **a**, *Klf2* mRNA expression, KLF2 motif accessibility, and enrichment of KLF2 gene signature in control or ZFP148 KO cells. **b**, Top, chromatin accessibility of the *Klf2* locus; Bottom, ZFP148 and IgG peaks measured by CUT&Tag-seq. **c**, CX3CR1 versus Ly108 expression in CRISPR-edited P14 CD8⁺ T cells 21 days post-LCMV CI13 infection. Right, frequency of subpopulations. **d**, Relative mRNA expression of *Klf2*. **e**, KLF2-EGFP fusion protein expression. **f**, Frequency of CX3CR1⁺ Ly108⁻ or GZMB⁺ TCF1⁻ subpopulations among transferred P14 CD8⁺ T cells 21 days post-LCMV CI13 infection. **g**, Relative luciferase activity in control or ZFP148-deficient EL4 cells transfected with luciferase reporter containing the *Klf2* distal element.

Taken in their entirety, our data support a model in which **ZFP148 represses *Klf2* through a defined distal cis-regulatory element and thereby restrains KLF2-driven effector differentiation (line 367-368)**. Nevertheless, we agree with the reviewer that further experimentation will be required, and that additional mechanisms may coexist to drive effector differentiation independently of KLF2. As such, we have added the following discussion: “Future studies are needed to define the exact ZFP148-bound motifs and thereby elucidate its precise regulatory mechanism. Our findings also raise the possibility that ZFP148 is required for the formation or maintenance of KLF2-negative lineages, including progenitor and terminally exhausted subsets. When ZFP148 is absent, these lineages are diminished, thereby shifting the differentiation landscape toward KLF2-positive effector populations. This model warrants further investigation using lineage-tracing approaches and temporally controlled deletion of ZFP148 during defined differentiation stages.” (revised manuscript, line 490-496)

Reviewer #3 (Remarks to the Author):

The authors have done an excellent job addressing all comments. I only have a few minor comments left:

- After re-reading the study, I would still suggest to remove Figure 1F-h, as it's been subsequently addressed more accurately. Moreover, the in vitro data also do not really align with the in vivo data (Extended Fig 3i,j) and therefore do not necessarily strengthen the study.

132 **Response:** We thank the reviewer for this helpful suggestion and have removed the figure
133 panels from this CRISPR KO experiment (page 39, existing data, Fig. 1f-h and page 52,
134 existing data, Extended Data Fig. 1h-k).

135
136 - Fig 2C: Rather than gating and demonstrating %GzmB+ tetramer+ cells, the authors could
137 gate on Tex cells and show the gzmB MFI. The outcome should be the same, but it would
138 represent a more straightforward illustration.

139 **Response:** We agree with this valuable suggestion and have substituted existing data Fig. 3c
140 with density plots showing normalized GZMB expression in control versus ZFP148 KO CD44^{high}
141 GP₃₃₋₄₁ tetramer⁺ CD8⁺ T cells and bar plot showing GZMB MFI (page 40, new data, Fig. 3c;
142 Response Letter Fig. 3).

143
144 **Response Letter Figure 3.** Left, GZMB protein expression in CD44^{high} GP₃₃₋₄₁ tetramer⁺ CD8⁺ T cells from
145 control or ZFP148 KO mice days 8, 16, 22, and 30 post-LCMV CI13 infection. Right, GZMB MFI. The p-
146 values were determined using Tukey's HSD test. *p-value ≤ 0.05, ** p-value ≤ 0.01, *** p-value ≤ 0.001, and
147 **** p-value ≤ 0.0001.

148
149 - The authors are showing “Numbers of cells/million splenocytes” (Ext Fig. 3). The authors need
150 to show total number of cells, not per splenocytes. Moreover, the authors need to adjust their
151 axis labeling as existing axis (eg Ext Fig 4) might have a mistake as ~5 million tetramer+
152 cells/”million splenocytes” would indicate ~5x10e12 cells/LN.

153 **Response:** We thank the reviewer for this helpful suggestion and apologized for the mistake
154 regarding axis labeling. We have now replaced with new data Extended Data Fig. 3c, d, f, h
155 (page 54) and new data Extended Data Fig. 4b, d, e (page 56) accordingly, which are also
156 shown below.

157

158

159 **Response Letter Figure 4. a**, Number of CD44^{High} GP₃₃₋₄₁ tetramer⁺ CD8⁺ T cells in the spleens from

160 control or ZFP148 KO mice infected with LCMV Cl13. **b**, Numbers of CX3CR1⁺ Ly108⁻ T_{eff}, CX3CR1⁻ Ly108⁺

161 T_{pro}, or CX3CR1⁻ Ly108⁻ T_{exh} subsets in spleens from experimental groups described in **a**. **c**, Number of

162 CD44^{High} GP₃₃₋₄₁ tetramer⁺ CD8⁺ T cells on day 9 post-LCMV Armstrong infection. **d**, Number of KLRG1⁺

163 CD127⁻ or KLRG1⁻ CD127⁺ subset on day 9 post-LCMV Armstrong infection. The p-values in **a-d** were

164 determined using multiple independent-sample t-tests. *p-value ≤ 0.05, ** p-value ≤ 0.01, *** p-value ≤

0.001, and **** p-value ≤ 0.0001.

165

166 - What is the rationale for referring to increases in Ki67 as “accelerated proliferation” (lines
 167 171)? It rather represents an increased proliferation, which is not sustained over time as Ki67
 168 expression was similar on day 30 (Fig. 2d). Similarly, what is the Ki67 expression of T_{pex} or
 169 terminal exhausted T cells? If Ki67 expression is decreased in KO-T_{pex} cells, then one could
 170 argue that ZFP148 restricts proliferation but rather differentiation as KO cells show enhanced
 171 proliferation and formation of CX3CR1⁺ cells, which is in line with the authors own conclusion:
 172 “These findings suggest that ZFP148 deletion primarily limits the differentiation of cells into T_{ex}
 173 without altering the dysfunctional phenotype on a per-cell basis.” (Line 189)

174 **Response:** We thank the reviewer for this insightful comment. We agree that Ki67 reflects the
 175 proportion of actively cycling cells and therefore indicates increased proliferation rather than
 176 accelerated proliferation. We have revised the text accordingly (**revised manuscript, line 155**).

177 To address the reviewer’s question regarding Ki67 expression in T_{pex}/T_{pro} and terminally
 178 exhausted subsets, we quantified Ki67 in T_{pex}/T_{pro} and T_{exh} populations across multiple
 179 timepoints. We observed significantly higher Ki67 expression in ZFP148 KO T_{pex}/T_{pro} cells on
 180 days 16 and 22 post-infection (higher trend on days 8 and 30), whereas Ki67 levels in T_{exh} cells
 181 remained comparable between groups (**page 54, new data, Extended Data Fig. 3I; Response**
 182 **Letter Fig. 5a**). Thus, ZFP148 primarily restrains proliferation in less-exhausted antigen-specific
 183 CD8⁺ T cells at early stages of chronic LCMV infection.

184 Importantly, although ZFP148-deficient T_{pex}/T_{pro} cells exhibit increased proliferation, their
 185 frequency and absolute number in the spleen are reduced (**page 54, existing data, Extended**
 186 **Data Fig. 3e,f; Response Letter Fig. 5b**), indicating that they concurrently undergo preferential
 187 differentiation into downstream lineages—predominantly CX3CR1⁺ effector cells. These data

188 therefore do not support a model in which ZFP148 only restricts differentiation but instead
 189 demonstrate that ZFP148 deficiency enhances early proliferative activity and promotes
 190 differentiation toward the effector lineage over the terminal exhausted lineage.

191
 192 **Response Letter Figure 5.** a, Ki67 protein MFI in T_{pro} or T_{exh} subsets of CD44^{high} GP₃₃₋₄₁ tetramer⁺ CD8⁺ T
 193 cells post-LCMV CI13 infection. b, frequency or number of CX3CR1⁻ Ly108⁺ T_{pro} subset in spleens. The p-
 194 values in a and b were determined using Tukey's HSD test. *p-value ≤ 0.05, ** p-value ≤ 0.01, *** p-value ≤
 195 0.001, and **** p-value ≤ 0.0001.

196
 197
 198